JCB | Journal of Cell Biology

# EVL and MIM/MTSS1 regulate actin cytoskeletal remodeling to promote dendritic filopodia in neurons

Sara S. Parker[1], Kenneth Tran Ly[1], Adam D. Grant[2], Jillian Sweetland[1], Ashley M. Wang[1], James D. Parker[1], Mackenzie R. Roman[3], Kathylynn Saboda[4], Denise J. Roe[4], Megha Padi[2,5], Charles W. Wolgemuth[4,5,6,7], Paul Langlais[3], and Ghassan Mouneimne[1,2]

Dendritic spines are the postsynaptic compartment of a neuronal synapse and are critical for synaptic connectivity and plasticity. A developmental precursor to dendritic spines, dendritic filopodia (DF), facilitate synapse formation by sampling the environment for suitable axon partners during neurodevelopment and learning. Despite the significance of the actin cytoskeleton in driving these dynamic protrusions, the actin elongation factors involved are not well characterized. We identified the Ena/VASP protein EVL as uniquely required for the morphogenesis and dynamics of DF. Using a combination of genetic and optogenetic manipulations, we demonstrated that EVL promotes protrusive motility through membrane-direct actin polymerization at DF tips. EVL forms a complex at nascent protrusions and DF tips with MIM/MTSS1, an I-BAR protein important for the initiation of DF. We proposed a model in which EVL cooperates with MIM to coalesce and elongate branched actin filaments, establishing the dynamic lamellipodia-like architecture of DF.

## Introduction

The neuronal synapse is the communication interface between neurons. Aberrant synaptic structure and connectivity is implicated in neurodevelopmental disorders, including intellectual disability, schizophrenia, and autism spectrum disorder (ASD; De Rubeis et al., 2014; Gilman et al., 2011; Fromer et al., 2014). Dendritic filopodia (DF) are actin-rich synaptic precursors that provide opportunities for new synaptic connections, and are abundant during neonatal neurodevelopment and activity-dependent plasticity (Zuo et al., 2005; Portera-Cailliau et al., 2003; Ziv and Smith, 1996). During synaptogenesis, these dynamic protrusions emanate from the dendritic arbor, sampling axon partners to establish new connections. If the axo-dendritic pairing is favored, DF are stabilized and can mature into dendritic spines—the postsynaptic compartment of excitatory synapses. Recent works suggest the protrusion dynamics of DF influence nascent synapse formation and the capability to remodel into a spine (Sanchez-Arias et al., 2020; Kayser et al., 2008; Carlson et al., 2011). Importantly, across several neurodevelopmental conditions, an exuberance of DF, morphologically immature synapses, and altered actin dynamics is observed in patients, as well as in mouse and in vitro models of Fragile X

syndrome, ASD, and schizophrenia (Cruz-Martín et al., 2010; Isshiki et al., 2014; Griesi-Oliveira et al., 2018; Jia et al., 2014; Sudarov et al., 2013). These conditions are often associated with mutations and variants in actin-associated proteins, suggesting a convergent etiological mechanism of dysregulation in actin dynamics, and highlighting the need for exquisitely tight control of the actin cytoskeleton for appropriate neural connectivity (Fromer et al., 2014; Gilman et al., 2011; Yan et al., 2016). As such, uncovering the actin regulators involved in the initiation and dynamics of DF informs not only the molecular basis of neuroplasticity, but furthers our understanding of the pathophysiology of neurodevelopmental disorders.

The organization of actin in DF includes Arp2/3-mediated branched actin, filaments of mixed polarity, and non-muscle myosin-II; this cytoskeletal architecture is distinct from the parallel linear filaments actin observed in conventional cell filopodia, which are bundled by fascin and are devoid of myosin-II (Korobova and Svitkina, 2010; Hotulainen et al., 2009). Extensive works have established Arp2/3 as required for the initiation of DF and spine morphogenesis, and that its loss or dysregulation is associated with behavioral deficits in mice (Hotulainen et al.,

[1]Department of Cellular and Molecular Medicine, College of Medicine, University of Arizona, Tucson, AZ, USA; [2]Cancer Biology Program, University of Arizona Cancer Center, Tucson, AZ, USA; [3]Division of Endocrinology, Department of Medicine, College of Medicine, University of Arizona, Tucson, AZ, USA; [4]University of Arizona Cancer Center and Mel and Enid Zuckerman College of Public Health, University of Arizona, Tucson, AZ, USA; [5]Department of Molecular and Cellular Biology, College of Science, University of Arizona, Tucson, AZ, USA; [6]Department of Physics, College of Science, University of Arizona, Tucson, AZ, USA; [7]Johns Hopkins Physical Sciences-Oncology Center, Johns Hopkins University, Baltimore, MD, USA.

Correspondence to Ghassan Mouneimne: gmouneimne@arizona.edu; Sara S. Parker: saraparker@arizona.edu.

2009; Spence et al., 2016; Kim et al., 2013). Arp2/3 nucleates actin branching from pre-existing filaments, creating free barbed ends for polymerization by actin elongation factors (Chesarone and Goode, 2009), including formins and Ena/VASP family proteins. These proteins associate processively with growing actin filaments to facilitate the addition of profilin: G-actin complexes (Chesarone and Goode, 2009). Although critically important for actin remodeling, the specific actin elongation factors contributing to DF are not fully characterized. Notably, despite their essential functions in regulating neuronal morphogenesis and axonal growth cone filopodia (Lebrand et al., 2004; Kwiatkowski et al., 2007; Dent et al., 2007; Menon et al., 2015), and in contrast to the described role for formins in DF (Kawabata Galbraith et al., 2018; Spence et al., 2016; Hotulainen et al., 2009), whether the Ena/VASP family plays a role in DF dynamics is largely unknown.

In this study, we determined that EVL is the dominant Ena/VASP paralog expressed during early synaptogenesis in cortical neurons and is required for DF morphogenesis and protrusive motility. EVL enriches to the tips of DF in an EVH1-dependent manner and enhances their dynamics by promoting actin polymerization. Loss of EVL results in a failure of DF elongation and dynamics, leaving small lamellipodia-like protrusions that are myosin-II- and Arp2/3-dependent. Further, by acutely localizing EVL through optogenetic approaches, we demonstrated that EVL is both necessary and sufficient for DF motility. Using an unbiased proteomics approach, we identified a complex between EVL and the Inverse BAR (I-BAR) protein MIM/MTSS1, which interact at nascent protrusions and DF tips to promote DF initiation and motility, respectively. Our findings support a model where collaboration of MIM and Arp2/3 at the initial site of protrusion provides a "hot spot" of dynamic actin, which is coalesced and elongated by the enrichment of EVL to protrusion tips, giving rise to a canonical DF.

## Results

To investigate actin remodeling in DF, primary cortical neuron cultures were imaged at in vitro day 11 (D11), which immediately precedes a developmental period of robust synaptogenesis (Fig. S1, A and B). Using the non-perturbing postsynaptic density fluorescent reporter PSD95-FingR (Gross et al., 2013) in living neurons, we confirmed that nascent connections formed through DF can mature into stable synapses over time (Fig. S1 B). Immunofluorescence labeling or expression of the Arp2/3 complex subunit Arp3 or the myosin-II regulatory light chain (MRLC) confirmed that the majority of DF at D11 are fascin-negative and myosin-II/Arp2/3-positive structures (Fig. S1, C–E). In contrast, growth cone filopodia and filopodia originating from the soma were strongly labeled for fascin (Fig. S1 C). Arp2/3 was observed along the length and at the tips of DF, while MRLC enriched at the base (Fig. S1, D and E). This cytoskeletal architecture agrees with prior characterizations of DF composition (Korobova and Svitkina, 2010; Hotulainen et al., 2009).

DF engage in many dynamic behaviors, including initiation, protrusion, and retraction. We disentangled these behaviors by extracting several metrics from tip tracking data. Absolute tip

displacement in time represents the net dynamics arising from all behaviors. We established a "substantiative motility" threshold of 0.0128 µm/s (one pixel displacement per 5 s interval), in order to define DF or durations of time as motile or non-motile. Protrusion and retraction events, indicative of actin remodeling, are captured by the change in DF length between successive timepoints; rate is derived from the median of all instantaneous changes in length exceeding ± 0.0128 µm/s. Due to the rapid motility of DF, we used Total Internal Reflection Fluorescence Microscopy (TIRFM) to maximize acquisition rate while minimizing phototoxicity.

### EVL is the predominant Ena/VASP paralog regulating DF

To determine involvement of the Ena/VASP family proteins—MENA, VASP, and EVL—in early synaptogenesis, we examined the effects of suppressing their activity. We used peptides containing the FPPPP (FP4) repeat sequence from *Listeria monocytogenes* ActA protein (Niebuhr et al., 1997), which binds the EVH1 domains of MENA, VASP, and EVL. Since the primary mode of activation of Ena/VASP proteins is through recruitment to their sites of action, sequestering them at mitochondria using FP4 fused to a mitochondrial targeting sequence (FP4-MITO) suppresses their activity (Bear et al., 2000). Additionally, we engineered an acute induction system by cloning these constructs into a doxycycline-inducible lentiviral expression vector. This allowed us to minimize harmful effects caused by long-term suppression of Ena/VASP (Kwiatkowski et al., 2007; Dent et al., 2007). After 12 h of doxycycline induction of mCherry-FP4-MITO expression, DF exhibited reduced overall dynamics and altered morphology, compared to the negative control mCherry-APPPP(AP4)-MITO (Fig. 1 A). We examined the effect of FP4-MITO induction on DF tip motility, and found that average speed was substantially reduced compared to AP4-MITO (Fig. 1 B). These data revealed that a greater proportion of DF from FP4-MITO-expressing neurons are, on average, "non-motile" (52.1%) during the duration of imaging, compared to AP4-MITO (29.8%; Fig. 1 C). Further, FP4-MITO significantly reduced the length of DF (Fig. 1 D). These data suggest that Ena/VASP proteins influence DF morphology and motility.

We examined the expression of MENA, EVL, and VASP in primary cortical neuron cultures throughout development by RT-qPCR. *Enah* and *Evl* expression was high during neuronal morphogenesis (D7) and during periods of early synaptogenesis (D11, D14) compared to *Vasp* (Fig. 1 E and Fig. S2 G). Western blot corroborated the expression pattern at the protein level (Fig. 1 F). Importantly, primary neuron cultures inherently include a mixture of glia and neurons; due to this unavoidable contamination by glial cells, the individual contribution of each cell type to total mRNA or protein levels the individual contribution of each cell type to total mRNA or protein levels cannot be assessed. To overcome this, we examined expression in cortical neurons specifically, by analyzing publicly available mouse single-cell RNA-seq datasets from The Allen Brain Map data portal (Allen Cell Types Database [2015]). Glutamatergic cortical neurons from adult mouse (Fig. S2, A, B, and E) and adult human (Fig. S2. C, D, and F) had strong expression of *Enah* and *Evl* and

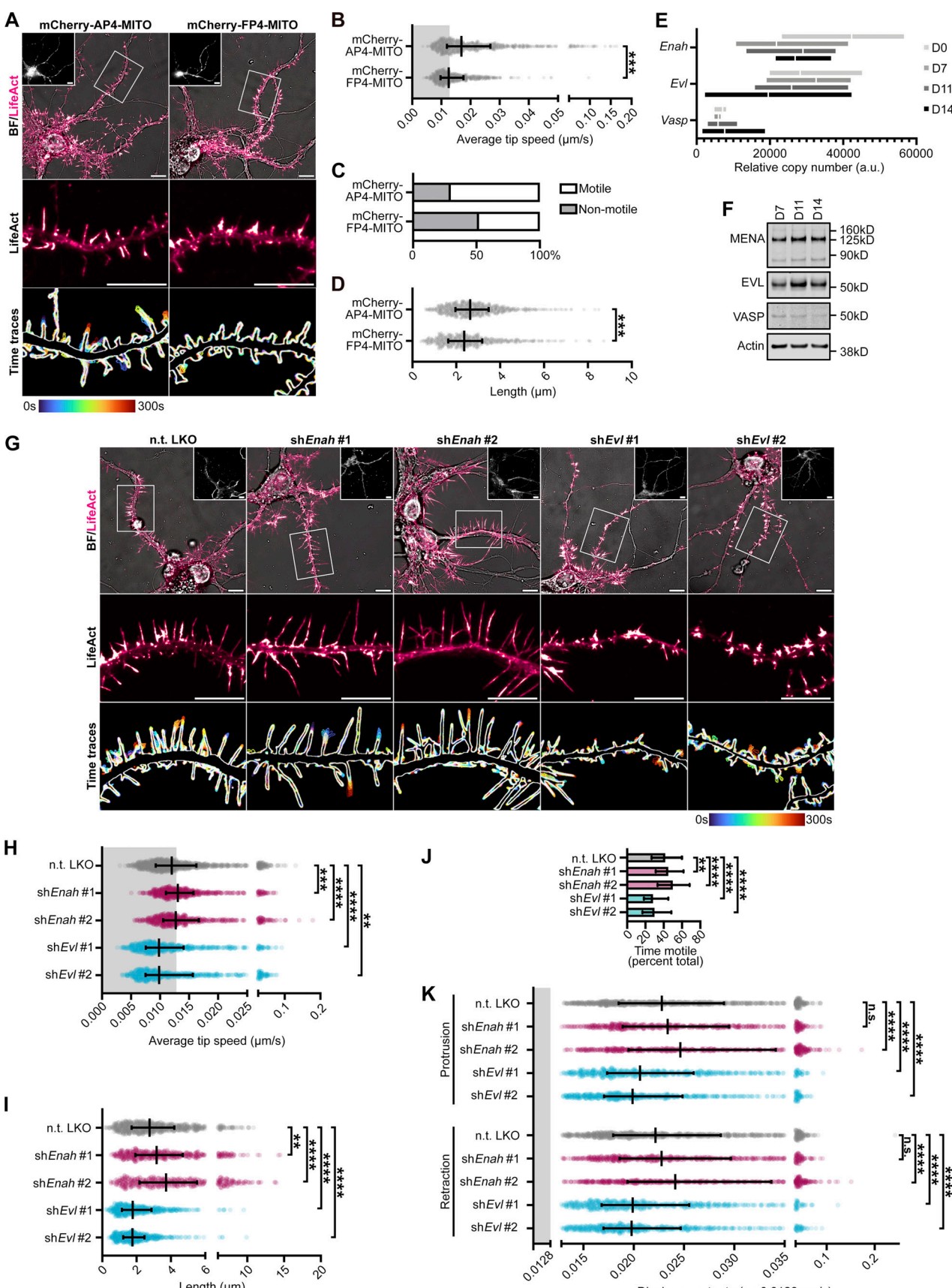

Figure 1. **EVL is the predominant Ena/VASP paralog regulating dendritic filopodia. (A)** Live primary mouse cortical neurons at day in vitro 11 (D11) expressing EGFP-LifeAct and doxycycline-inducible mCherry-AP4-MITO (left column) or mCherry-FP4-MITO (right column), imaged 12 h after doxycycline

induction. Top: Full cell image. Inset: mCherry-AP4/FP4-MITO expression. Middle: Indicated segment of dendrite presented with an intensity-coded LUT. Bottom: Maximum intensity projection of temporally color-coded binary mask outline, illustrating DF dynamics during imaging (5 s interval, 5 min duration). **(B)** Scatterplot of average speed of DF tips, calculated as the average absolute tip displacement between successive timepoints. Gray shaded region indicates average speed less than 0.0128 μm/s (non-motile DFs). Median ± interquartile range (IQR). Mixed-effects model was used for statistical comparisons; $n$ = 731–917 total DF from 4–7 neurons per biological replicate, $N$ = 3 replicates. **(C)** Bar graph of percent of total DF population with average tip speeds greater than 0.0128 μm/s (motile) or less than 0.0128 μm/s (non-motile). Mann-Whitney test; $n$ = 731–917 total DF from 4–7 neurons per biological replicate, $N$ = 3 biological replicates. **(D)** Scatterplot of average length reached during the duration of imaging. Median ± IQR. Mixed-effects model; $n$ = 731–917 total DF from 4–7 neurons per biological replicate, $N$ = 3 biological replicates. **(E)** RT-qPCR of RNA samples from primary mouse cortical neuron cultures at indicated days in vitro. CTs were normalized to the average of three housekeeping genes, and relative copy numbers were generated using $2^{-\Delta CT} \times 10^6$. Floating bars span the minimum and maximum data points, central line denotes mean. $N$ = 3 biological replicates. **(F)** Representative western blot of protein lysates from primary neuron cultures at indicated days in vitro, probed with antibodies targeting MENA, EVL, and VASP. **(G)** Live D11 neurons expressing EGFP-LifeAct and pLKO-shRNA-TurboRFP targeting *Enah*, *Evl*, or non-targeting (n.t.) as indicated. Cells were transduced with shRNA lentiviral particles on D7. Top: Full cell image. Inset: TurboRFP expression identifying shRNA-positive neurons. Middle: Indicated segment of dendrite. Bottom: Maximum intensity projection of temporally color-coded binary mask outline (5 s interval, 5 min duration). **(H)** Scatterplot of average speed of DF tips. Gray shaded region indicates average speed <0.0128 μm/s (non-motile). Median ± IQR. Mixed-effects model; $n$ = 630–1013 total DF from 4–5 neurons per biological replicate, $N$ = 3–5 biological replicates. **(I)** Scatterplot of average length reached during duration of imaging. Median ± IQR. Mixed-effects model; $n$ = 630–1,013 total DF from 4–5 neurons per biological replicate, $N$ = 3–5 biological replicates. **(J)** Bar graph of percent time motile (percent of time per DF in which instantaneous speed was greater than 0.0128 μm/s). Median ± IQR. Kruskal-Wallis test corrected for multiple comparisons; $n$ = 630–1,013 total DF from 4–5 neurons per biological replicate, $N$ = 3–5 biological replicates. **(K)** Scatterplot of median protrusion and retraction rates of DFs (the median of values when instantaneous change in length was greater than ±0.0128 μm/s [motile]). Median ± IQR. Mixed-effects model; $n$ = 630–1,013 total DF from 4–5 neurons per biological replicate, $N$ = 3–5 biological replicates. *$P$ <0.05, **$P$ <0.01, ***$P$ <0.001, ****$P$ <0.0001, n.s. is not significant. Scale bars = 10 μm. See also Fig. S2 and Video 1. Source data are available for this figure: SourceData F1.

comparatively low levels of *Vasp*. Thus, we prioritized examining MENA and EVL function in DF.

To determine the influence of MENA and EVL on DF dynamics, we utilized paralog-specific shRNA to knockdown expression (Fig. S2, H and I). At D11, sh*Enah* DF retained a normal morphology, with significantly higher tip speed and length, compared to non-targeting (n.t.) shRNA control neurons. In contrast, sh*Evl* neurons exhibited a stubby, flare-like DF morphology with profoundly suppressed dynamics (Fig. 1, G and I and Video 1). In addition, sh*Enah* DF spent more of their time engaging in substantive motility, while sh*Evl* DF exhibited significantly less time motile (Fig. 1 J). To explore this effect on motility, we examined the rate of protrusion and retraction as distinct events in control and knockdown cells. sh*Evl* DF had a slower rate of both protrusions and retractions compared with n.t. controls, while sh*Enah* DF overall trended toward faster rates (Fig. 1 K). Together, these data suggest differential roles for MENA and EVL in DF dynamics, and implicate EVL as a crucial regulator of DF morphology and motility.

**Tip enrichment of EVL precedes DF protrusion**
To investigate the function of MENA and EVL in regulating DF dynamics, we overexpressed EGFP-tagged constructs in cortical neuron cultures (Fig. S3 A). EGFP-MENA and EGFP-EVL both enriched at the tips of protrusions (Fig. 2 A and Fig. S3 B), and exhibited a dose-dependent spectrum of phenotypes. Strong expression (signal-to-noise ratio [SNR] > 1.5) of MENA had a high frequency of fan-like protrusions, while high-expressing EVL neurons extended broad lamellipodia-like protrusions from the dendrites (Fig. S3 B, arrowheads). These extreme phenotypes suggest unique functions for MENA and EVL in neuronal morphogenesis, and prompted us to restrict all subsequent quantification to low-expressing neurons (SNR < 1.5). We categorized DF morphology as normal, forked, multi-forked (>2 forks), flaring (lamellipodia-like protrusions), and complex forked-flaring morphology (Fig. S3 C). We found that MENA

overexpressors increased the proportion of multi-forked DF, while EVL promoting flaring. These findings further underscore the differential, non-overlapping function of MENA and EVL, and suggest that each paralog promotes distinct filopodia-like structures in neurons.

Expression of MENA and EVL each enhanced overall DF dynamics, tip speed, and protrusion rate, and MENA-expressing neurons exhibited slightly elongated filopodia (Fig. 2, A–E). EGFP-MENA and EGFP-EVL were readily visible at DF tips (Fig. 2 A, Fig. S3 B, and Video 2). This tip enrichment was dynamic, with EVL exhibiting a higher variance than MENA in mean intensity levels at a tip ROI (Fig. 3, A and B), indicating that EVL tip localization is more dynamic than MENA. Examination of the relationship between tip enrichment and motility by kymography revealed that while EGFP alone remained uniformly cytosolic during DF protrusion and retraction, EGFP-EVL consistently exhibited distinct and persistent enrichment at DF tips starting before a protrusion event. In contrast, EGFP-MENA was enriched at the tip during protrusion in only a subset of DF (Fig. 3 C).

To determine whether tip enrichment of MENA or EVL is correlated with DF protrusive behavior, we utilized the filopodia analysis program, Filopodyan (Urbančič et al., 2017). Filopodyan cross-correlates tip fluorescence and motility in time, determines the time offset at which correlation is highest, and identifies the top-correlating subcluster (TCS) of DFs in each condition. This enables examination of DF motility within the top-correlating subcluster compared to DFs with poor fluorescence-to-motility correlation (non-TCS; Fig. S4, A–C). Cross-correlation function (CCF) revealed that the top-correlating subcluster of EGFP-EVL DFs exhibited highest correlation at offsets of –10 and –5 s, indicating that EVL tip enrichment precedes motility. In contrast, cross-correlation in EGFP-MENA DFs peaked at 0 s and +5 s offset, suggesting that these DFs exhibited tip enrichment with or following motility (Fig. 3, D–F). We next compared protrusive motility in top-correlating versus

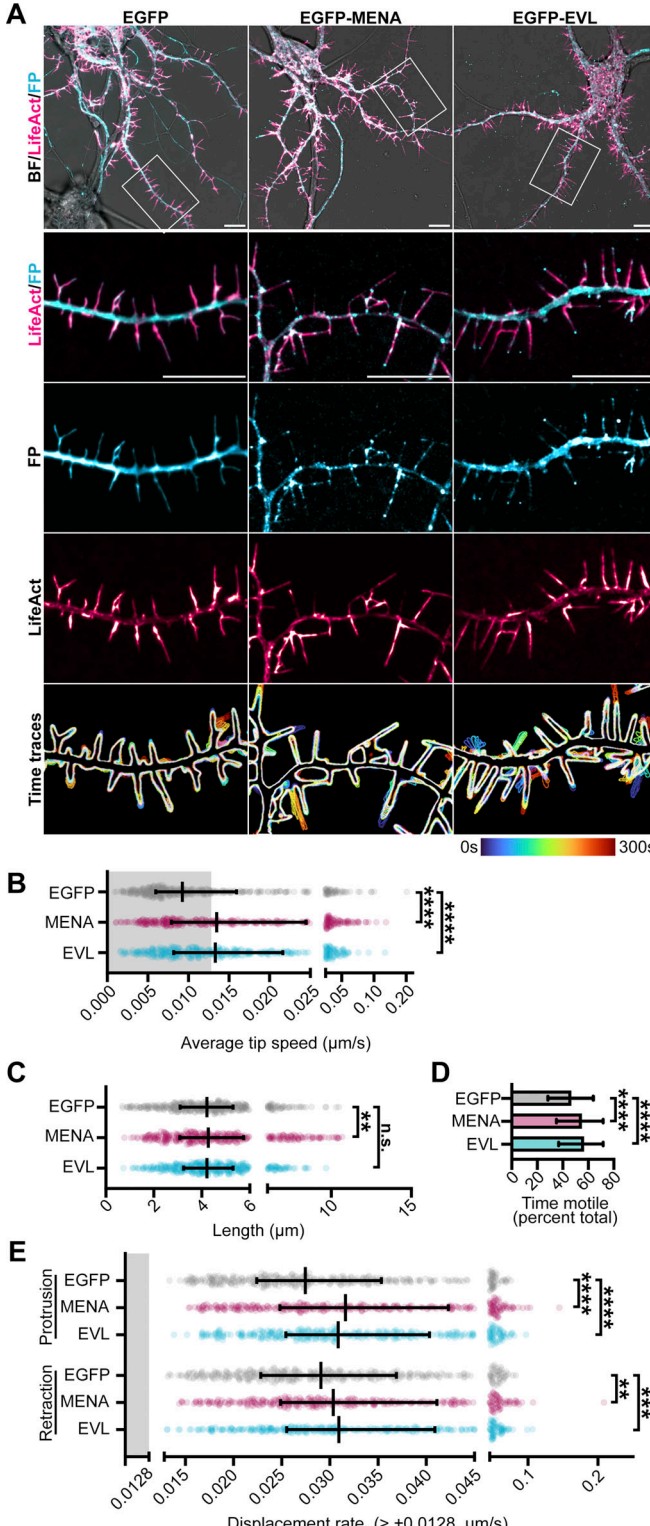

**Figure 2. MENA and EVL overexpression enhance DF motility. (A)** Live primary mouse cortical neurons at day in vitro 11 (D11) expressing mRuby2-LifeAct and EGFP (left column), EGFP-MENA (middle column), or EGFP-EVL (right column). Row 1: Full cell image. Row 2: Indicated segment of dendrite. Rows 3 and 4: Individual fluorescent channels presented with intensity-coded LUTs. Row 5: Maximum intensity projection of temporally color-coded binary mask outline, illustrating DF dynamics during imaging (5 s interval, 5 min duration). **(B)** Scatterplot of average speed of DF tips, calculated as the average absolute tip displacement between successive timepoints. Gray shaded

non-top-correlating subcluster DFs. Tip enrichment was associated with increased rates of protrusion in EGFP-EVL top-correlating subcluster DF, while EGFP-MENA top-correlating subcluster DF spent more time engaging in protrusive events (Fig. 3, G and H). Importantly, EGFP-EVL's observed top-correlating subcluster showed significantly higher cross-correlation than randomized datasets, while EGFP-MENA did not (Fig. S4, D and E). Together, these data suggest that although MENA overexpression may promote DF dynamics, EVL tip enrichment uniquely predicts protrusion events, providing further evidence for a distinct function for EVL in DF motility.

### EVL is required for DF morphology and motility

To investigate the direct involvement of EVL in DF dynamics, we examined cortical neurons derived from EVL knockout mice (EVL KO; Kwiatkowski et al., 2007), and compared them to wild-type neurons throughout a developmental time course in vitro. EVL knockout neurons underwent normal overall morphogenesis, and developed neurites, axons, and branching comparable to wild-type neurons (Fig. S5, A–E). Importantly, no compensatory upregulation of MENA or VASP was observed in EVL knockout neurons (Fig. S5, F and G). EVL knockout neurons manifested increasingly pronounced defects in DF morphology and dynamics over time, and by D9, the majority of dendrites exhibited very short filopodia or small lamellipodia-like flares (Fig. 4 A and Video 3), with maximal reductions in speed, length, and protrusion rate by D11 (Fig. 4, B and C and Fig. S5 H). This suggests that EVL is the dominant protein regulating DF dynamics after D9, or that sustained loss of EVL during DF morphogenesis drives these phenotypes.

Given the profound morphological defects observed in DF from loss of EVL, we next examined the impact on synaptogenesis. PSD95-FingR was delivered through lentivirus to high-density cultured cortical neurons from wild-type and EVL knockout mice, enabling live observation of dendritic spine formation and dynamics. By D14, synapses were abundant in wild-type neuron cultures. Wild-type neurons overexpressing EGFP-EVL had reduced dendritic spine density; in contrast, EVL knockout neurons showed a marked increase in spine density (Fig. 4, D and E). Next, we examined the kinetics of synapse formation and their stability in live cultured neurons. Nascent

region indicates average speed less than 0.0128 μm/s (non-motile DFs). Median ± interquartile range (IQR). Mixed-effects model; *n* = 368–389 total DF from 4–6 neurons per biological replicate, *N* = 3 biological replicates. **(C)** Scatterplot of average length reached during the duration of imaging. Median ± IQR. Mixed-effects model; *n* = 368–389 total DF from 4–6 neurons per biological replicate, *N* = 3 biological replicates. **(D)** Bar graph of percent time motile (percent of time per DF in which instantaneous speed was greater than 0.0128 μm/s). Median ± IQR. Kruskal-Wallis test corrected for multiple comparisons; *n* = 368–389 total DF from 4–6 neurons per biological replicate, *N* = 3 biological replicates. **(E)** Scatterplot of median protrusion and retraction rates of DFs (the median of values when instantaneous change in length was greater than ±0.0128 μm/s [motile]). Median ± IQR. Mixed-effects model; *n* = 368–389 total DF from 4–6 neurons per biological replicate, *N* = 3 biological replicates. *P < 0.05, **P < 0.01, ***P < 0.001, ****P < 0.0001, n.s. is not significant. Scale bars = 10 μm. See also Fig. S3 and Video 2.

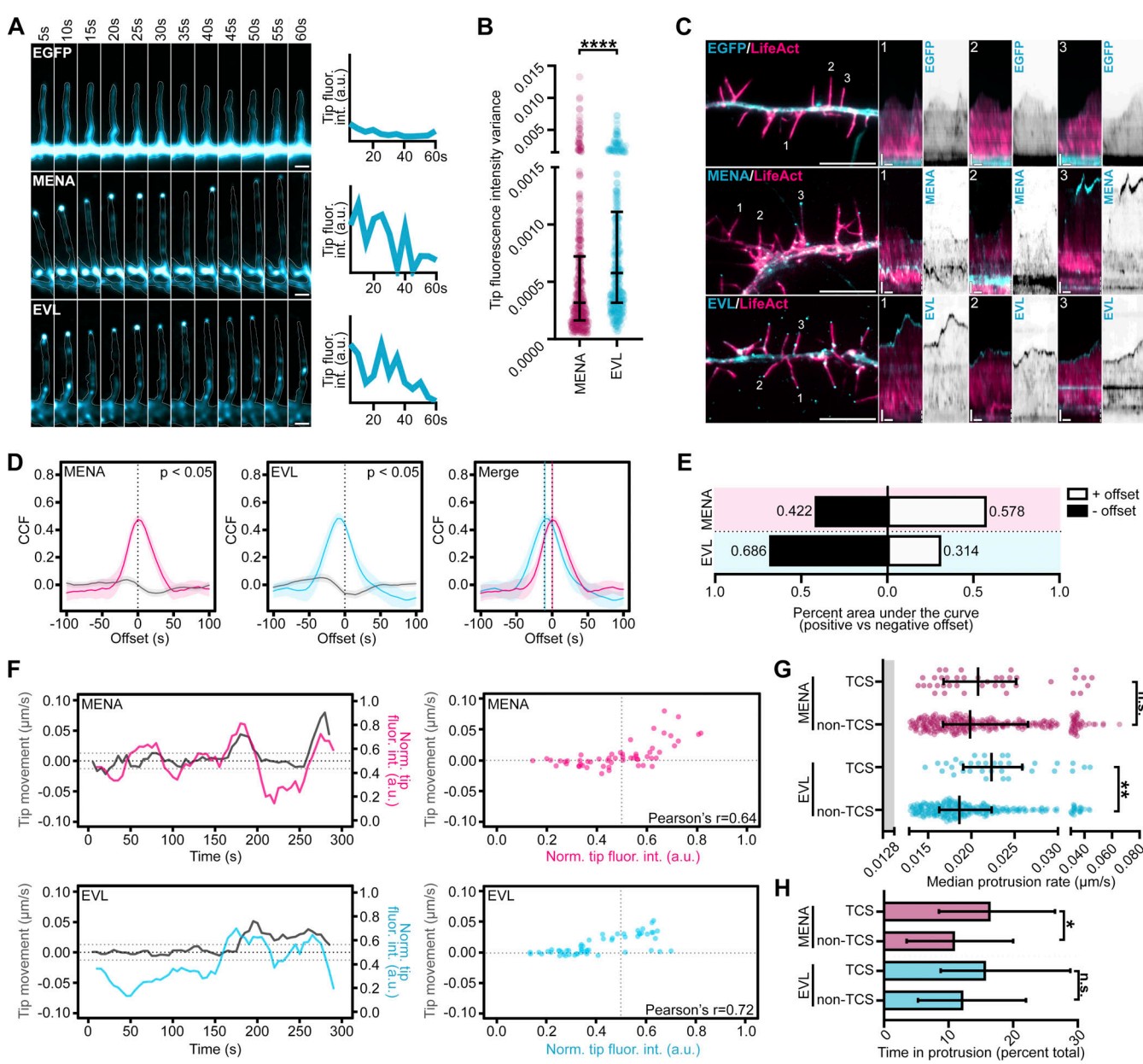

Figure 3. **Tip enrichment of EVL precedes DF protrusion. (A)** Filmstrip of live primary mouse cortical neurons at day in vitro 11 (D11) expressing EGFP (top row), EGFP-MENA (middle row), or EGFP-EVL (bottom row) in a representative DF showing localization dynamics presented with an intensity-coded LUT. Line scan of normalized fluorescence intensity over time at tracked tip (right; 5 s interval, 1 min duration). Scale bar = 1 μm. **(B)** Scatterplot of tip fluorescence variance at DF tips normalized to local background demonstrates range of tip enrichment per DF and hence magnitude of on-off dynamics. Median ± interquartile range (IQR). Mixed-effects model; n = 368–389 total DF from 4–6 neurons per biological replicate, N = 3 biological replicates. **(C)** Live D11 neurons expressing mRuby2-LifeAct and EGFP (top row), EGFP-MENA (middle row), or EGFP-EVL (bottom row). Segment of dendrite (left); scale bar = 10 μm. Numbers indicate DF position analyzed by kymograph (right), highlighting protein localization dynamics during DF motility (5 s interval, 5 min duration). Vertical scale bar = 1 μm, horizontal scale bar = 1 min, dashed line indicates dendrite. **(D)** Line plots of cross-correlation function (CCF) of normalized tip fluorescence intensity and tip motility as a function of time offset for top-correlating subcluster (TCS; color lines) versus non-top-correlating subcluster (non-TCS; gray lines) determined in Fig. S3 B, of DFs from D11 neurons expressing EGFP, EGFP-MENA, or EGFP-EVL. Peak cross-correlation at negative offset values indicates that fluorescence enrichment precedes motility, while peak cross-correlation at positive offset values indicates fluorescence enrichment follows motility. Mean ±95% CI. Peak cross-correlation GFP = 0.43 at offset 0.413 at offset −5s; MENA = 0.475 at offset 0, 0.464 at offset +5 s; EVL =, 0.482 at offset −10 s, 0.480 at offset −5 s. Statistical significance determined by peak cross-correlation >2/√(n-|offset|). **(E)** Percent of area under the curve at positive and negative offset values for indicated conditions. **(F)** Line plots (left) and scatterplots (right) for one representative DF from the top-correlating subcluster in D11 neurons from indicated conditions. Left: Line plot of tip motility (gray, left y-axis) and normalized tip fluorescence (color, right y-axis) during imaging (5 s interval, 5 min duration). Right: Scatterplot of normalized tip fluorescence and tip motility at individual timepoints demonstrating strength of relationship by Pearson's correlation test. **(G)** Scatterplot of median protrusion rates for top-correlating subcluster versus non-top-correlating subcluster DFs from indicated conditions. Protrusion rate is the median of values when instantaneous change in length was greater than +0.0128 μm/s (motile, protruding). Median ± interquartile range (IQR). Kruskal-Wallis test corrected for multiple comparisons. TCS n = 42 (MENA) and 39 (EVL); non-TCS n = 320 (MENA) and 321 (EVL) total DF from 4–6 neurons per biological replicate, N = 3 biological replicates. **(H)** Bar graph of percent of time in protrusion during the duration of imaging for top-correlating

subcluster versus non-top-correlating subcluster DFs from indicated conditions. Percent time in protrusion is calculated as the percent of time per DF in which positive change in length between successive timepoints was greater than 0.0128 µm/s. Median ± IQR. Kruskal-Wallis test corrected for multiple comparisons. TCS $n$ = 42 (MENA) and 39 (EVL); non-TCS $n$ = 320 (MENA) and 321 (EVL) total DF from 4–6 neurons per biological replicate, $N$ = 3 biological replicates. *P < 0.05, **P < 0.01, ***P < 0.001, ****P < 0.0001, n.s. is not significant. See also Fig. S4.

synapses were observed being formed by DF and turned over in wild-type neurons, as well as instances of DF extending from existing synapses and forming new connections (Fig. 4 F and Video 4). In contrast, actin-rich dendritic protrusions from EVL knockout neurons formed comparatively stable synapses with suppressed remodeling and dynamics. To quantify the dynamics of dendritic spines, we tracked PSD95 foci in time. In knockout neurons, PSD95 foci were longer lived than in wild-type neurons, and exhibited reduced displacement (Fig. 4, G and H). Collectively, these data suggest that EVL-mediated DF are not required for synapse formation, however, their presence and dynamic behaviors promotes synaptic remodeling.

**The EVH1 domain is required for EVL localization and activity**
To determine the mechanism of EVL's regulation of DF dynamics, we reconstituted either wild-type EVL, or mutants of EVL, in knockout neurons (Fig. 5, A and B). Re-introduction of wild-type EVL fully rescued DF motility, length, and tip localization (Fig. 5, C–F and Video 5). We investigated two categories of EVL mutants: domain deletions that influence actin polymerization, and deletions that impact EVL localization. EVL polymerizes actin through direct interactions with profilin (PFN), G-actin, and F-actin (Chereau and Dominguez, 2006). Reconstitution of a mutant lacking the PFN-binding site (ΔPFN) showed tip localization and slow DF elongation, while a G-actin binding mutant (ΔGAB) fully rescued of tip enrichment, motility, and partially restored length. In contrast, loss of both profilin- and G-actin-binding (ΔActinPoly) or both the G- and F-actin-binding domains (ΔGAB-FAB) resulted in DF with dynamic tip localization, but low motility (Fig. 5, C–F and Video 5).

Targeted localization of Ena/VASP proteins is critical to their function. The EVH1 domain interacts with proline-rich residues on binding partners, and the proline-rich domain (PRD) contains motifs that bind SH3 domains in addition to profilin (Lambrechts et al., 2000; Reinhard et al., 1995). Deletion of the EVH1 domain (ΔEVH1) failed to rescue DF, while expression of the EVH1 domain alone showed a diffuse tip enrichment and partial restoration of dynamics but not length. Expression of the PRD alone slightly rescued motility and altered DF morphology. Expression of the EVH2 domain alone, which contains the GAB and FAB domains, and a coiled-coiled region for Ena/VASP tetramerization, promoted the formation of large lamellipodia-like flares and diffuse localization throughout the protrusion (Fig. 5, C–F and Video 5). Together, these results indicate that the EVH1 domain is required for EVL's enrichment to DF tips, while actin polymerization, primarily mediated by the profilin-binding region, is required for promoting DF length and motility.

**EVL tip localization is necessary and sufficient for DF motility**
Since localized recruitment is the central mechanism by which Ena/VASP proteins are activated, we employed an optogenetic approach, iLID (improved light-induced dimer; Guntas et al., 2015; Zimmerman et al., 2016), to examine the effects of acute recruitment of EVL on DF dynamics. iLID is a LOV2-based technology, in which each half of a heterodimerizing protein pair (SsrA-SspB) is fused to (1) a protein of interest, or (2) the dark state-obscured C-terminus of the photosensitive protein AsLOV2 along with a plasma membrane targeting sequence (CAAX) or mitochondrial targeting sequence (MITO). During exposure to 488 nm light, LOV2 undergoes a conformational change, removing steric occlusion of SsrA to permit its dimerization with SspB, and thereby inducibly and reversibly recruiting the protein of interest to the plasma membrane or to the mitochondria (Fig. 6 A). We generated EVL-iLID and expressed it together with CAAX-iLID or MITO-iLID to inducibly and reversibly localize EVL towards or away from sites of action, respectively, during photostimulation (Fig. 6 A). In glial cells (used for design validation), plasma membrane recruitment of EVL-iLID increased leading edge and filopodial tip enrichment of EVL, and enhanced lamellipodial and filopodial protrusion. In contrast, photorecruitment of EVL-iLID to mitochondria resulted in reduced membrane localization, and suppressed membrane dynamics (Fig. 6 B and Fig. S6 B). Importantly, iLID does not alter the activity of EVL; high EVL-iLID expression exhibits similar effects to EVL overexpression even in the absence of photostimulation. To minimize this caveat, we examined low expressers of EVL-iLID (SNR < 1.5) reconstituted in EVL knockout neurons.

With CAAX-iLID, photostimulation rapidly recruited EVL-iLID to the plasma membrane and to DF tips, concurrent with DF elongation and increased motility; these activities subsided after termination of light exposure (Fig. 6, C, H, and I and Video 6). When co-expressed with MITO-iLID, EVL-iLID was pulled away from DF tips to mitochondria in the dendrite during photostimulation, resulting in reduced motility that recovered partially during the observation period following light withdrawal (Fig. 6, D, H, and I and Video 6). These data demonstrate that EVL localization at the tips of DF is both necessary and sufficient for DF motility. Further, we used the iLID system to acutely drive the ΔEVH1 mutant to the membrane. Expression of mEmerald-ΔEVH1 in EVL knockout neurons did not enhance DF dynamics or promote tip enrichment (Fig. 5, C–F). However, upon photostimulation, ΔEVH1-iLID became tip-enriched, and substantially increased DF motility, which subsided after termination of light exposure (Fig. 6, E, H, and I and Video 6). Conversely, the ΔActinPoly-iLID mutant, which lacks the ability to polymerize actin, exhibited membrane enrichment but no increase in DF motility during light exposure, while ΔPFN-iLID, which polymerizes actin weakly, showed enrichment with a modest but non-significant increase in motility (Fig. 6, G–I and Video 6). These data show that although the EVH1 domain is required for EVL localization and DF motility, engineered

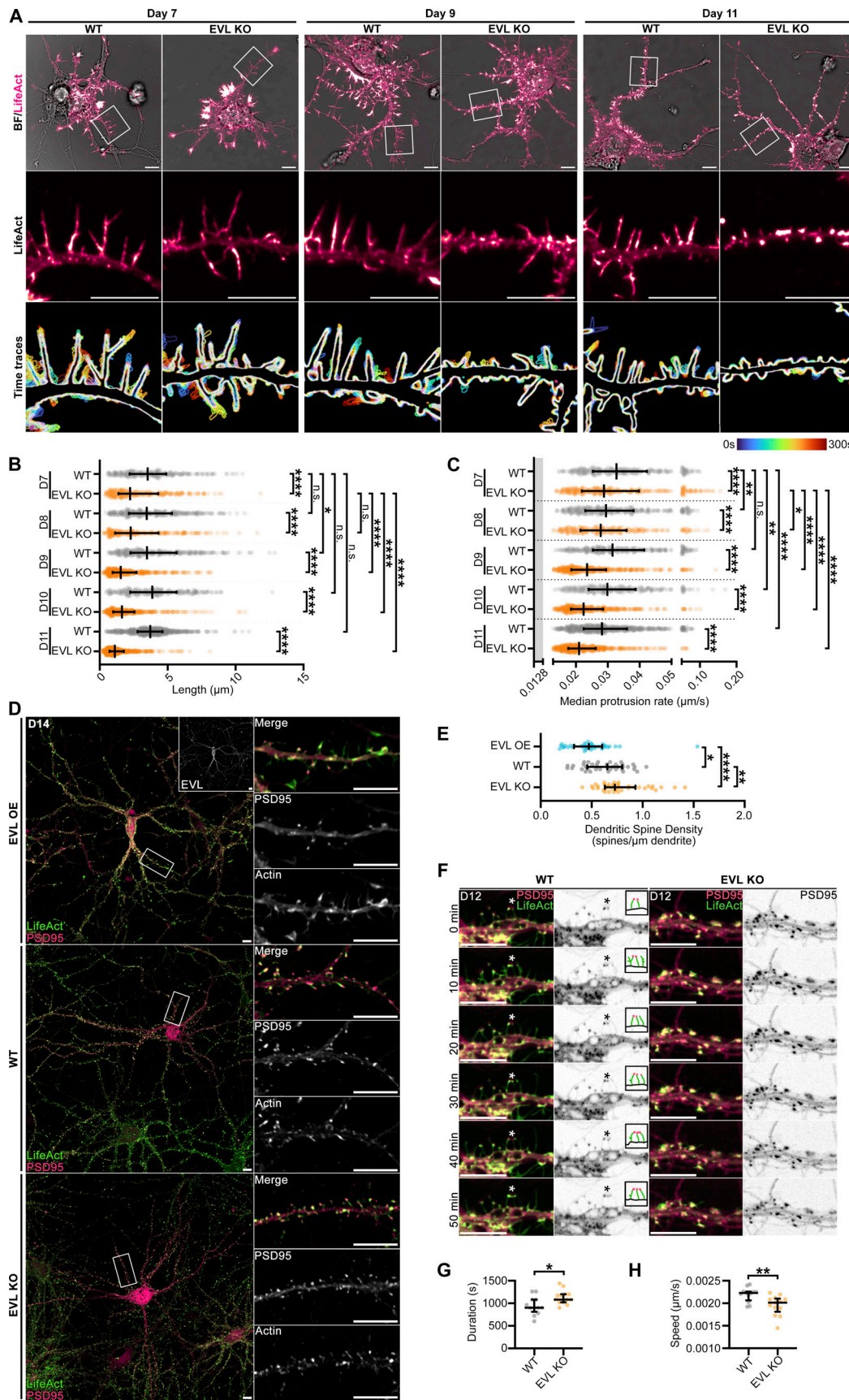

Figure 4. **EVL is required for DF morphogenesis and influences dendritic spine plasticity. (A)** Live primary cortical neurons derived from wild-type (WT) or EVL knockout (KO) mice at indicated days in vitro expressing mRuby2-LifeAct. Top: Full cell image. Middle: Indicated segment of dendrite presented with an

intensity-coded LUT. Bottom: Maximum intensity projection of temporally color-coded binary mask outline, illustrating DF dynamics during imaging (5 s interval, 5 min duration). Scale bars = 10 μm. **(B)** Scatterplot of average length reached during the duration of imaging for indicated conditions. Median ± interquartile range (IQR). Mixed-effects model; $n = 280–641$ total DF from 4–6 neurons per biological replicate, $N = 3$ biological replicates. **(C)** Scatterplot of median protrusion rates of DFs (the median of values when instantaneous change in length was greater than +0.0128 μm/s (motile), protruding). Median ± IQR. Mixed-effects model; $n = 280–641$ total DF from 4–6 neurons per biological replicate, $N = 3$ biological replicates. **(D)** High-density cultures of wild-type, EVL knockout, or wild-type overexpressing EVL neurons at day in vitro 14 (D14) expressing EGFP-LifeAct and synapse reporter PSD95-FingR-mRuby2. Left: Full cell image. Inset: EVL expression. Right: Indicated segment of dendrite. Scale bars = 10 μm. **(E)** Scatterplot of average dendritic spine density per neuron of indicated conditions. Median ± IQR. One-way ANOVA corrected with Holm-Sidak multiple comparisons; $n = 38$ total neurons for each condition, $N = 3$ biological replicates. **(F)** Filmstrip of segment of dendrite from live cortical neurons derived from wild-type or EVL knockout mice at D14 expressing EGFP-LifeAct and PSD95-FingR-mRuby2 showing synapse formation and dynamics. Inset: Illustration of region indicated by asterisks. Scale bars = 10 μm. **(G)** Scatterplot of average duration of synapse persistence per neuron as quantified by PSD95-positive foci tracked in time. Median ± IQR. Unpaired $t$ test; $n = 12$ (wild-type) and 13 (knockout) total neurons; $N = 3$ biological replicates. **(H)** Scatterplot of average speed per neuron of PSD95-positive foci tracked in time. Median ± IQR. Unpaired $t$ test; $n = 12$ (wild-type) and 13 (knockout) total neurons; $N = 3$ biological replicates. *$P < 0.05$, **$P < 0.01$, ***$P < 0.001$, ****$P < 0.0001$, n.s. is not significant. See also Fig. S5, Video 3, and Video 4.

### The I-BAR protein MIM/MTSS1 is a binding partner of EVL

To further investigate the mechanism by which EVL regulates DF, we took an unbiased quantitative proteomics approach to identify putative protein partners of EVL that co-regulate DF motility. We performed affinity purification mass spectrometry (AP-MS) using endogenous EVL as bait in lysates from cultured cortical neurons. Spectrum counts of EVL immunoprecipitates versus IgG negative control immunoprecipitates uncovered a high-confidence interaction with MIM/MTSS1, as well as bona fide Ena/VASP binding partners including profilin (Fig. S7 A). MIM is an Inverse BAR (I-BAR) domain protein, which binds and deforms the plasma membrane at sites of $PIP_2$ enrichment to promote outward membrane curvature (Saarikangas et al., 2009; Mattila et al., 2007). In addition, MIM has been shown to indirectly promote Arp2/3-dependent assembly of branched actin networks (Lin et al., 2005). Together, these activities promote the generation of a proto-protrusion, a critical first step in DF initiation in cortical neurons (Saarikangas et al., 2015). Therefore, we prioritized studying MIM as an EVL binding partner.

We confirmed EVL:MIM interaction by performing the reciprocal co-immunoprecipitation with tagged proteins in HEK293T, demonstrating that EVL robustly co-immunoprecipitates with MIM (Fig. 7 A). To identify the domains mediating interaction, we expressed MIM with full-length EVL, ΔEVH1 mutant, or a mutant lacking the proline-rich domain (ΔPRD). Although MIM interacted with full-length EVL and ΔPRD, the interaction was lost with ΔEVH1 (Fig. 7 B). The EVH1 domain of Ena/VASP proteins binds target proteins through interaction with the proline-rich consensus sequence [FWYL]PXΦP ("FP4 motif"; Hwang et al., 2022; Ball et al., 2000). A ProSite scan revealed an EVH1-recognition motif $_{634}$LPSPP$_{638}$ at the C-terminus of MIM (Sigrist et al., 2013). To determine if this motif contributes to EVL:MIM binding, we mutagenized LPSPP (WT) to AGGGG (AG4) to eliminate EVH1 recognition (Ball et al., 2000) in ΔIBAR-MIM to prevent dimerization with endogenous MIM. In contrast to ΔIBAR-MIM-WT, the EVH1 domain of EVL was unable to co-immunoprecipitate ΔIBAR-MIM-AG4 (Fig. S7 B), suggesting that the canonical EVH1:FP4 motif interaction is important for the EVL:MIM protein complex.

### MIM cooperates with EVL to promote DF initiation and motility

In neurons, expression of MIM was robust throughout in vitro development and was not affected by EVL knockout (Fig. S7 C). To investigate the relationship between MIM and EVL in regulating DF, we expressed MIM alone, EVL alone, or both MIM and EVL in wild-type cortical neurons and examined DF motility. When expressed alone, MIM enriched at DF tips during protrusive motility (Fig. 7 C) and increased absolute tip dynamics, yet MIM expression was not sufficient to significantly increase protrusion rate (Fig. 7, F and G). When MIM and EVL are co-expressed, they strongly co-localized at DF tips (Fig. 7, D and E) and dramatically enhanced all aspects of motility compared to control and EVL-expressing neurons (Fig. 7, F and G and Video 7). Interestingly, MIM expression significantly decreased DF length, a phenotype that was ameliorated by EVL co-expression (Fig. 7 H). These findings suggest that the relationship between MIM and EVL is cooperative, and co-expression manifests additive phenotypes compared to expression of MIM or EVL alone.

When expressed in EVL knockout neurons, MIM robustly localizes to the tips of short DFs and protrusions; these sites of enrichment fail to elongate or exhibit substantial protrusive motility (Fig. S7, E–I). Intriguingly, while MIM and wild-type EVL co-expression rescues DF morphology and motility in knockout neurons, co-expression of MIM and ΔEVH1 resulted in a moderate yet significant intermediate phenotype, with diffuse ΔEVH1 localized proximally to the MIM-enriched DF tip (Fig. S7, E–I). These observations underscore the necessity of the EVH1 domain for EVL's tip localization, and suggests that ΔEVH1 EVL, which is able to polymerize actin, can partially restore EVL activity when MIM is abundant.

To determine if MIM is competent to promote dynamics at the tip, we generated MIM-iLID to acutely manipulate MIM localization. Light-stimulated recruitment of MIM-iLID to the membrane failed to promote initiation or DF motility in wild-type neurons (data not shown). This may be due to MIM's requirement of upstream regulation by $PIP_2$ for its activity, or saturation of suitable sites (Mattila et al., 2007). In contrast, optogenetic sequestration of MIM-iLID at mitochondria reduced lamellipodia dynamics in glial cells, and resulted in modest but significant reductions in DF motility in wild-type neurons (Fig. S7, J–L). This suggests that tip-localized MIM facilitates actin polymerization.

membrane recruitment using CAAX-iLID rescues knockout phenotypes.

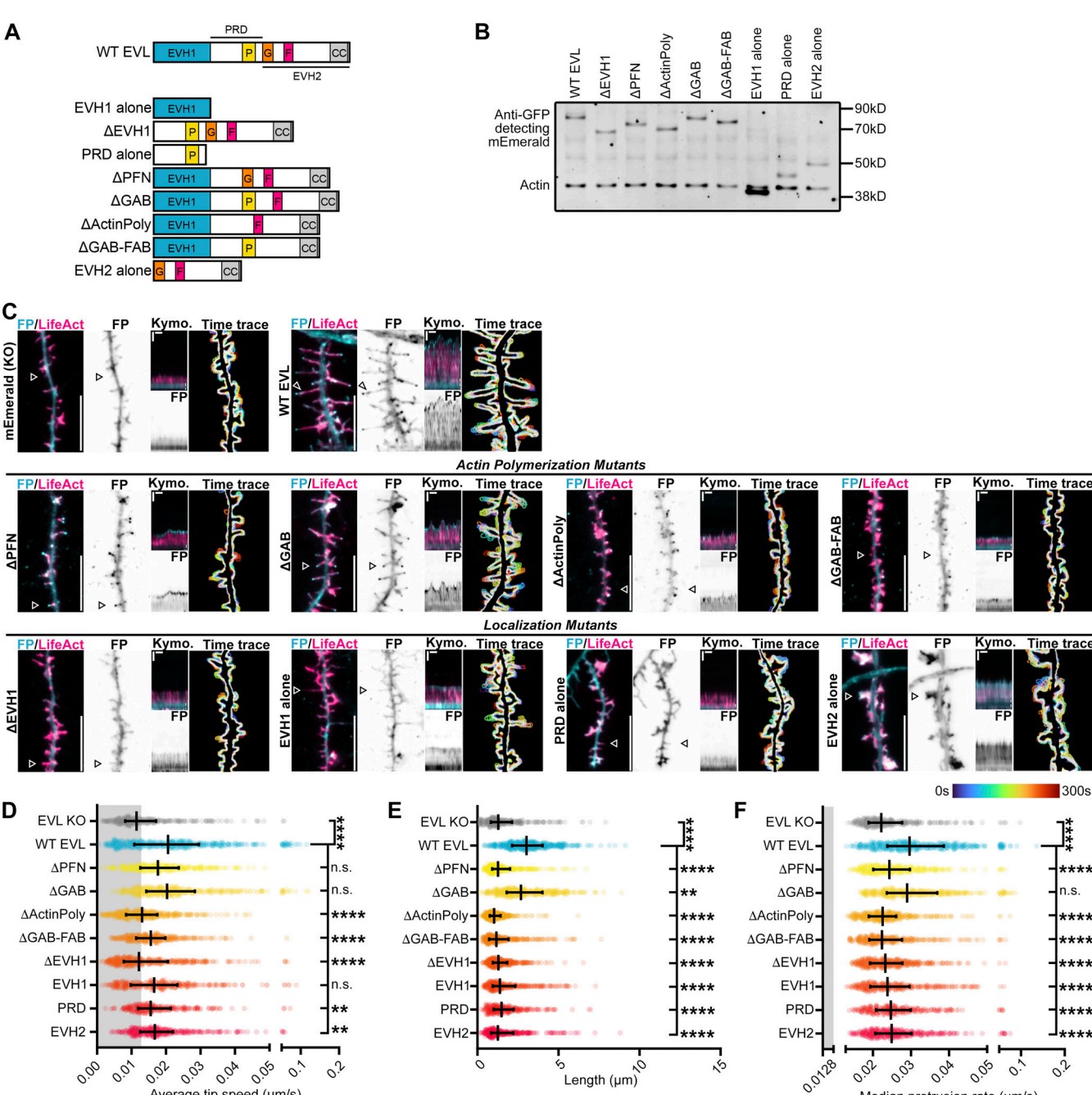

Figure 5. **EVL regulates DF morphogenesis and motility through membrane-targeted actin polymerization. (A)** Schematic of protein domains of wild-type EVL and EVL mutants used in this study. EVH1 (blue) = Enabled/VASP homology-1, P (yellow) = profilin binding region, G (orange) = G-actin binding region, F (red) = F-actin binding region, CC (gray) = coiled-coil domain, PRD = proline-rich region, EVH2 = Enabled/VASP homology-2. **(B)** Western blot of protein lysates from D11 cortical neurons derived from EVL knockout (KO) mice expressing mEmerald-EVL or various mutants of EVL as indicated. **(C)** Segment of dendrite from live primary cortical neurons derived from EVL knockout mice at indicated day in vitro 11 (D11) expressing mRuby2-LifeAct and mEmerald alone, or mEmerald-tagged wild-type EVL or mutants of EVL as indicated. From left to right, Panel 1: Merge of LifeAct and mEmerald-tagged proteins. Scale bar = 10 µm. Panel 2: mEmerald fluorescence intensity presented with an inverted LUT. Panel 3: Kymograph of DF position indicated by arrowhead (5 s interval, 5 min duration). Vertical scale bar = 1 µm, horizontal scale bar = 1 min, dashed line indicates dendrite. Panel 4: Maximum intensity projection of temporally color-coded binary mask outline (5 s interval, 5 min duration). **(D)** Scatterplot of average speed of DF tips, calculated as the average absolute tip displacement between successive timepoints. Gray shaded region indicates average speed less than 0.0128 µm/s (non-motile DFs). Median ± IQR. Mixed-effects model; n = 248–397 total DF from 3–6 neurons per biological replicate, N = 3–5 biological replicates. **(E)** Scatterplot of average length reached during the duration of imaging for indicated conditions. Median ± IQR. Mixed-effects model; n = 248–397 total DF from 3–6 neurons per biological replicate, N = 3–5 biological replicates. **(F)** Scatterplot of median protrusion rates of DFs for indicated conditions. Median ± IQR. Mixed-effects model; n = 248–397 total DF from 3–6 neurons per biological replicate, N = 3–5 biological replicates. *P < 0.05, **P < 0.01, ***P < 0.001, ****P < 0.0001, n.s. is not significant. See also Video 5. Source data are available for this figure: SourceData F5.

Figure 6. **EVL tip localization is necessary and sufficient for DF motility. (A and B)** Schematic of optogenetic EVL-iLID system. **(A)** The SsrA peptide (A) is embedded in the Jα helix of *As*LOV2, and its binding partner protein, SspB (B), is tagged to EVL. LOV2-SsrA is additionally tagged by a plasma membrane

targeting motif (CAAX) or mitochondrial targeting sequence (MITO). Upon irradiation with 488 nm light, the Jα helix unwinds and reveals the occluded SsrA peptide, enabling dimerization with its partner SspB. **(B)** Each iLID component—EVL-iLID and either CAAX-iLID or MITO-iLID—are introduced to target cells by lentivirus. Exposure of the whole cell to 488 nm light recruits EVL to the plasma membrane or to the mitochondria, and influences actin-based protrusions including lamellipodia and filopodia (filmstrip, right). **(C–G)** Segment of dendrite from live primary cortical neurons derived from EVL knockout mice at D11 expressing iRFP670-LifeAct, and indicated iLID constructs. **(C)** Wild-type EVL-iLID with CAAX-iLID. **(D)** Wild-type EVL-iLID with MITO-iLID. **(E)** ΔEVH1-iLID with CAAX-iLID. **(F)** ΔActinPoly-iLID with CAAX-iLID. **(G)** ΔPFN-iLID with CAAX-iLID. Neurons were photostimulated with 488 nm light for 5 min, and imaged for 5 min before, during, and after photostimulation. Rows 1–2: Kymograph of representative DF position. 15 min duration, 5 s interval, vertical scale bar = 1 µm, horizontal scale bar = 1 min, dashed line indicates dendrite. Row 3: Line plot of normalized fluorescence intensity (cyan) and DF tip speed (gray) over time. Rows 4–5: Segment of dendrite showing localization of EVL-iLID and LifeAct at single timepoint during photostimulation experiment. Individual channels displayed with black subtraction for ease of morphological comparison. Scale bars = 10 µm. Row 6: Maximum intensity projection of temporally color-coded binary mask outline. Arrowheads indicate DFs which exhibited altered dynamics following photostimulation. Asterisk in D indicates a mitochondrion that moved into the kymograph. **(H)** Fold change in average tip speed of 2.5 min bins during or post-photostimulation for indicated iLID conditions, relative to average tip speed pre-photostimulation. Median ± interquartile range (IQR). Mixed-effects model; $n$ = 85–120 total DF from 2–3 neurons per biological replicate, $N$ = 3 biological replicates. I. Scatterplot of background-normalized fluorescence intensity of EVL-iLID pre-photostimulation, or in 2.5 min bins during or post-photostimulation, for indicated iLID conditions. Median ± IQR. Mixed-effects model; $n$ = 85–120 total DF from 2–3 neurons per biological replicate, $N$ = 3 biological replicates.*$P < 0.05$, **$P < 0.01$, ***$P < 0.001$, ****$P < 0.0001$, n.s. is not significant. See also Fig. S6 and Video 6.

To examine the requirement of MIM in regulating DF, we knocked-down MIM using shRNA (Fig. 7, I–L and Fig. S7 D). sh*Mtss1* in wild-type neurons increased DF length (Fig. 7 L), which is the opposite phenotype observed in MIM overexpressers (Fig. 7 H) and in agreement with previous findings (Kawabata Galbraith et al., 2018). However, in contrast to acute inhibition of MIM using the iLID system, sh*Mtss1* did not alter DF dynamics (Fig. 7, J and K). These results suggest that MIM is not required for the motility of DF, and in fact, it limits DF length. We examined the effects of MIM knockdown on DF in the absence of EVL. sh*Mtss1* in EVL knockout neurons resulted in a substantial reduction in the density of actin-rich protrusions along the dendrite, while in wild-type neurons, no differences in protrusion density were observed between MIM knockdown and control (Fig. 7, I, M, and N). These findings suggest that in the absence of both EVL and MIM, protrusion initiation is compromised.

We next investigated the relationship between MIM and EVL in DF initiation. During spontaneous DF initiation, both EVL and MIM exhibit tip localization during protrusion, and are strongly coincident preceding an initiation event. Visualization of proto-protrusion formation on the dendrite by kymography suggests that MIM enrichment is required for DF initiation, but membrane enrichment alone is not sufficient for productive initiation of a DF (Fig. 7 O, arrowheads). Further, co-expression of MIM and EVL promoted a stricking increase in de novo DF initiation compared to expression of either protein alone (Fig. 7 P). To further examine the cooperation between EVL and MIM, we examined initiation events elicited by photorecruitment of EVL-iLID to the membrane by CAAX-iLID. Neurons co-expressing EVL-iLID and MIM-iRFP exhibited extremely dynamic and unstable DF, with more initiation events at baseline before photostimulation, compared to EVL-iLID alone (Fig. 7, Q and R and Video 8). Following photostimulation, initiation was further increased, and subsided after stimulation was suspended (Fig. 7 R). Importantly, forward protrusion was not observed unless both EVL and MIM were present in the proto-protrusion, consistent with the spontaneous DF initiation results (Fig. 7 Q, arrowheads). We used MITO-iLID to sequester EVL-iLID away from the membrane; photostimulation in these experiments suppressed DF initiation, which robustly recovered after light

was withdrawn (Fig. 7 R and Video 8). These data suggest that while EVL promotes the actin elongation crucial for generating protrusive force, MIM is required to "license" the initiation of a nascent DF by signaling for actin nucleation.

### Arp2/3 and myosin-II differentially contribute to DF dynamics

I-BAR proteins, such as MIM, indirectly promote the activation of Arp2/3-mediated actin nucleation (Lin et al., 2005; Saarikangas et al., 2015), which builds branched actin architecture in DF (Korobova and Svitkina, 2010). Further, an important feature of DF is their myosin-II-dependent contractility, which promotes dendritic spine head maturation and remodeling following adhesion (Chazeau et al., 2015; Koskinen et al., 2014; Tatavarty et al., 2012). To explore how Arp2/3 and myosin-II activity interface with EVL during DF dynamics, we investigated the effects of their acute pharmacological inhibition using CK-666 (Nolen et al., 2009) and blebbistatin (Rizvi et al., 2009), respectively, in wild-type and EVL knockout neurons.

Following Arp2/3 inhibition, wild-type DF exhibited rapid elongation compared to DMSO controls (Fig. 8, A and C and Video 9), in agreement with previous studies (Kawabata Galbraith et al., 2018; Spence et al., 2016; Hotulainen et al., 2009). In EVL knockout neurons, protrusion motility was moderately but significantly increased after Arp2/3 inhibition with some delay compared to wild-type (Fig. 8, B and C and Video 9). CK-666 also reduced F-actin content and flaring protrusions in both wild-type and EVL knockout neurons, leaving DF as thin protrusions (Fig. 8, A, B, and D and Video 9). These results demonstrate that loss of EVL is not wholly sufficient to influence the change in DF dynamics caused by the acute inhibition of branching nucleation by Arp2/3, and likely suggests an eventual switch to formin-mediated nucleation and/or polymerization (Burke et al., 2014; Suarez et al., 2015). Inhibition of myosin-II contractility slightly decreased tip velocity in EVL knockout neurons (Fig. 8, A–D). On the other hand, DF length was significantly increased with blebbistatin in wild-type neurons; this increase was not observed in knockout neurons (Fig. 8, A–D and Video 9). These results demonstrate that DF motility dynamics that are enhanced by myosin inhibition are EVL-dependent.

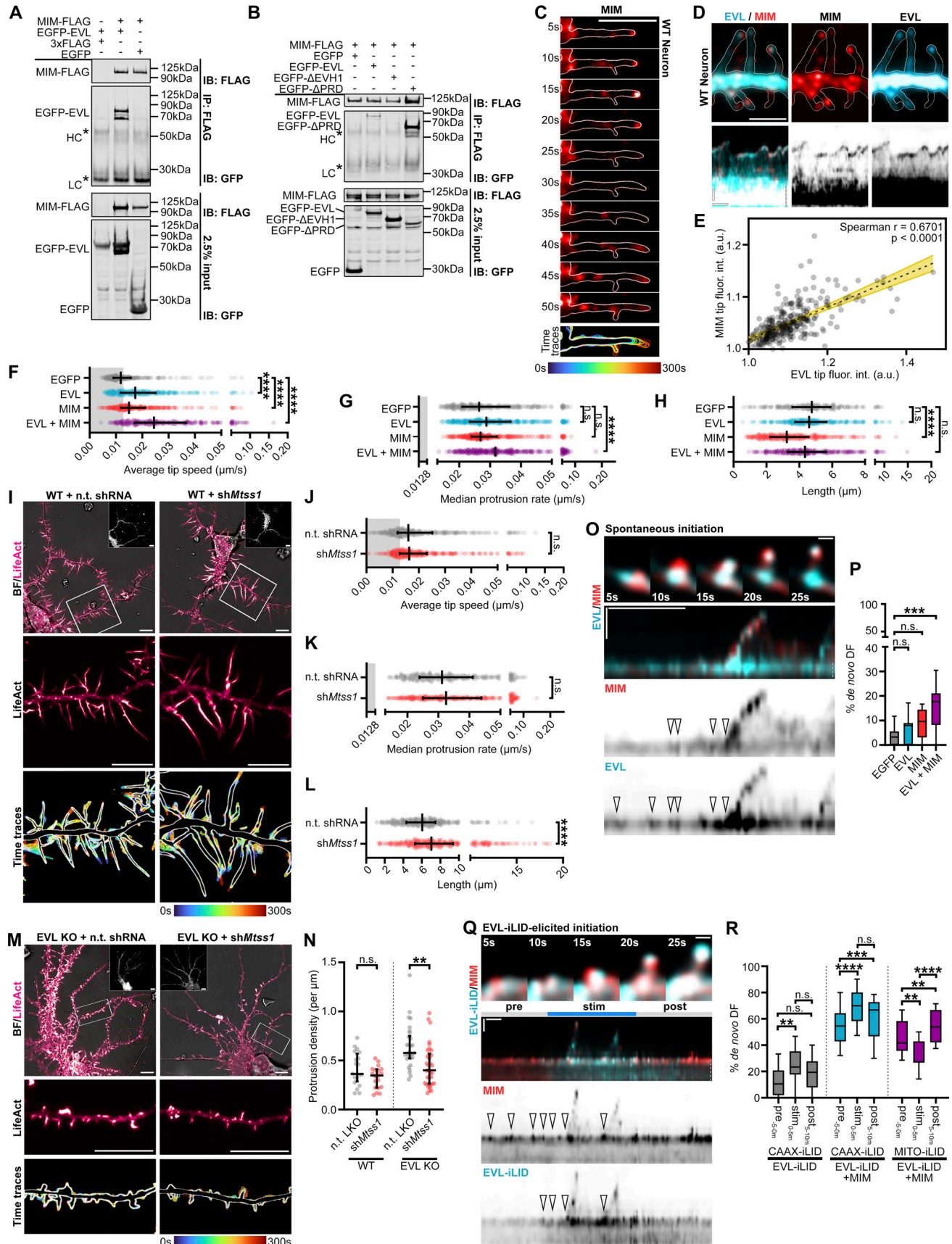

Figure 7. **MIM/MTSS1 cooperates with EVL to promote DF initiation and motility. (A and B)** Representative western blots of co-immunoprecipitation experiments confirming reciprocal interaction of MIM with EVL (A) and dependence of interaction on EVL's EVH1 domain (B). HEK293T overexpressing

indicated constructs were lysed and immunoprecipitated (IP) with FLAG antibody to pull down MIM-3xFLAG, resolved by SDS-PAGE, and immunoblotted (IB) with indicated antibodies. **(C)** Filmstrip showing localization dynamics of MIM-mRuby2 during protrusive motility in a wild-type cortical neuron DF at day in vitro 11 (D11; left), and maximum intensity projection of temporally color-coded binary mask outline (right; 5 s interval, 1 m duration). Scale bar = 5 μm. **(D)** Segment of dendrite from a wild-type cortical neuron at D11 co-expressing iRFP670-EVL and MIM-mRuby2 (top), and kymograph of DF indicated by arrowhead (bottom; 5 s interval, 5 min duration). Dendrite segment: Scale bar = 5 μm. Kymographs: Horizontal scale bar = 1 min, vertical scale bar = 1 μm, dashed line indicates dendrite. **(E)** Scatterplot of background-normalized EVL and MIM tip fluorescence at individual timepoints. Strong correlation observed by Spearman's correlation test. Individual timepoints presented from 5 total DF. **(F)** Scatterplot of average speed of wild-type DF tips for indicated conditions, calculated as the average absolute tip displacement between successive timepoints. Gray shaded region indicates average speed less than 0.0128 μm/s (non-motile DFs). Median ± interquartile range (IQR). Mixed-effects model; n = 273–345 total DF from 3–5 neurons per biological replicate, N = 3 biological replicates. **(G)** Scatterplot of median protrusion rates of wild-type DFs for indicated conditions (the median of values when instantaneous change in length was greater than +0.0128 μm/s [motile, protruding]). Median ± IQR. Mixed-effects model; n = 273–345 total DF from 3-5 neurons per biological replicate, N = 3 biological replicates. **(H)** Scatterplot of average length of wild-type DFs reached during the duration of imaging for indicated conditions. Median ± IQR. Mixed-effects model; n = 273–345 total DF from 3–5 neurons per biological replicate, N = 3 biological replicates. **(I)** Wild-type neurons expressing EGFP-LifeAct and pLKO-shRNA-TurboRFP targeting *Mtss1* or non-targeting as indicated. Cells were transduced with shRNA lentiviral particles on D7. Top: Full cell image. Inset: TurboRFP expression identifies shRNA-positive neurons. Middle: Indicated segment of dendrite. Bottom: Maximum intensity projection of temporally color-coded binary mask outline (5 s interval, 5 min duration). Scale bars = 10 μm. **(J)** Scatterplot of average speed of DF tips for indicated conditions. Median ± IQR. Mixed-effects model; n = 362–363 total DF from 2–4 neurons per biological replicate, N = 3 biological replicates. **(K)** Scatterplot of median protrusion rates of DFs for indicated conditions. Median ± IQR. Mixed-effects model; n = 362–363 total DF from 2–4 neurons per biological replicate, N = 3 biological replicates. **(L)** Scatterplot of average length of DFs reached during the duration of imaging for indicated conditions. Median ± IQR. Mixed-effects model; n = 362–363 total DF from 2–4 neurons per biological replicate, N = 3 biological replicates. **(M)** EVL knockout neurons expressing EGFP-LifeAct and pLKO-shRNA-TurboRFP targeting *Mtss1* or non-targeting as indicated. Top: Full cell image. Inset: TurboRFP expression identifies shRNA-positive neurons. Middle: Indicated segment of dendrite. Bottom: Maximum intensity projection of temporally color-coded binary mask outline (5 s interval, 5 min duration). Scale bars = 10 μm. **(N)** Scatterplot of protrusion density along dendrites from indicated conditions. Protrusions were defined as any actin-rich proturbences visibly extending beyond the dendrite. Median ± IQR. Mann-Whitney test; n = 19–35 dendrites from 3–5 neurons per biological replicate, N = 3 biological replicates. **(O)** Example of spontaneous DF initiation. Row 1: Filmstrip indicating localization of mEmerald-EVL and MIM-mRuby2 during DF initiation in a D11 wild-type neuron. Scale bar = 0.5 μm. Rows 2–4: Kymograph of mEmerald-EVL and MIM-mRuby2 during initiation. Arrowheads indicate foci of protein enrichment preceding initiation or motility events. 5 s interval, 5 min duration, horizontal scale bar = 1 min, vertical scale bar = 1 μm, dashed line indicates dendrite. **(P)** Box-and-whisker plot of newly initiated DFs as a percent of total DFs during the duration of imaging. Median ± IQR. One-way ANOVA corrected Holm-Sidak multiple comparisons; n = 9–14 total neurons, N = 3 biological replicates. **(Q)** Example of EVL-iLID-elicited DF initiation. Row 1: Filmstrip indicating localization of EVL-iLID and MIM-iRFP670 during photostimulation in a D11 knockout neuron. Scale bar = 0.5 μm. Rows 2–4: Kymograph of EVL-iLID and MIM-iRFP670 during initiation. Arrowheads indicate foci of protein enrichment preceding initiation or motility events. 5 s interval, 15 min duration, horizontal scale bar = 1 min, vertical scale bar = 1 μm, dashed line indicates dendrite. **(R)** Box-and-whisker plot of newly initiated DFs as a percent of total DFs during the duration of imaging before, during, and following iLID stimulation with indicated constructs. One-way ANOVA corrected Holm-Sidak multiple comparisons; n = 11–16 total neurons, N = 3 biological replicates *P < 0.05, **P < 0.01, ***P < 0.001, ****P < 0.0001, n.s. is not significant. See also Fig. S7, Video 7, and Video 8. Source data are available for this figure: SourceData F7.

## Discussion

Ena/VASP proteins are essential for neuronal morphogenesis. In this study, we identified a distinct role for EVL in the regulation of DF dynamics during early synaptogenesis in cortical neurons. EVL, through membrane-directed actin polymerization, is necessary and sufficient for generating DF morphology and protrusive motility. In contrast, MENA is not essential for DF motility, and VASP is not highly expressed. We determined that EVL is dynamically enriched to DF tips preceding protrusion, and loss of EVL profoundly impacts DF morphogenesis and dynamics. We identified an EVL:MIM complex, which promotes initiation of nascent DF and their motility. As an Inverse BAR (I-BAR) protein, MIM facilitates membrane deformation and promotes Arp2/3 activation to initiate DF protrusion (Saarikangas et al., 2015); however, the proteins responsible for promoting actin polymerization at these initiation sites had not been identified. We provided evidence that EVL is the primary Ena/VASP protein responsible for elongating actin at MIM-induced proto-protrusions. Our data support the following working model of DF morphogenesis (Fig. 8, E–G): (1) MIM initiates a proto-protrusion by facilitating membrane deformation; (2) Arp2/3 is activated at proto-protrusions and provides the initial burst in actin remodeling leading to initiation of DF protrusion; and (3) EVL is recruited by interaction with MIM leading to rapid actin polymerization and elongation of the protrusion to form a canonical DF, while myosin-II promotes rearward actin flow. In the absence of EVL, dendritic protrusions are unable to elongate.

### EVL is the major driver of DF

The critical role of Ena/VASP proteins in neuronal development is well established. MENA-VASP-EVL knockout mice exhibit profound neurodevelopmental defects resulting from a failure of filopodiagenesis (Dent et al., 2007; Kwiatkowski et al., 2007). MENA and VASP have been shown to regulate filopodia dynamics at the growth cone during axon pathfinding (Lanier et al., 1999; McConnell et al., 2016; Menon et al., 2015; Lebrand et al., 2004; Menzies et al., 2004). In contrast, we found that neurite and axon outgrowth were unaffected by EVL knockout, suggesting no growth cone defects occur with loss of EVL. We demonstrated that EVL is uniquely required for DF morphogenesis. Neurons derived from EVL knockout mice exhibit aberrant DF morphology and dynamics that manifest specifically during developmental periods in vitro when synaptic precursor DF are abundant and actively contributing to synapse formation. This indicates that DF are EVL dependent, while earlier developmental processes are likely mediated by different actin regulators.

Early morphogenic processes, such as neuritogenesis and dendrite branching, are regulated by neuronal filopodia

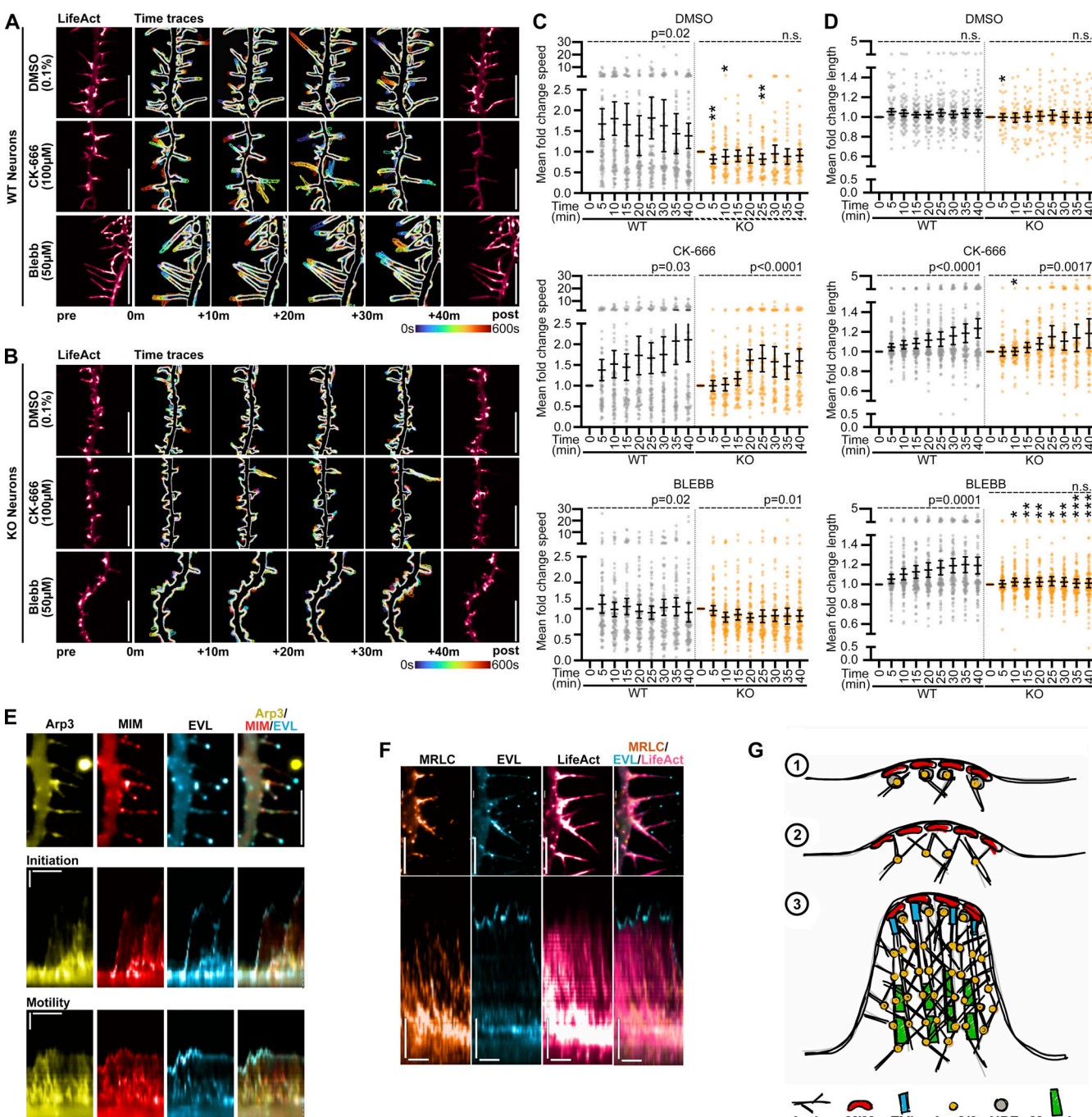

Figure 8. **Arp2/3 and myosin-II differentially contribute to DF initiation and motility. (A and B)** Segment of dendrite from wild-type (A) or EVL knockout (B) cortical neurons at D11 expressing mRuby2-LifeAct, and treated with DMSO (0.1%), Arp2/3 inhibitor CK-666 (100 μM), or myosin-II inhibitor blebbistatin (50 μM). Neurons were imaged 5 min prior to (left) and up to 40 min following (right) drug treatment. Maximum intensity projection of temporally color-coded binary mask outline of 10 min bins following treatment (middle; 15 s interval, 45 min duration). Scale bars = 10 μm. **(C)** Scatterplot of fold change of average tip speed following treatment with indicated pharmacological inhibitors, calculated as the average absolute tip displacement during successive 5 min bins, relative to average tip speed pre-treatment, for wild-type and EVL knockout DF. Mean ±95% confidence interval (CI). Mixed-effects model; DMSO n = 48–90 total DF, CK-666 n = 58–80 total DF, and Blebbistatin n = 88–96 total DF from 2–4 neurons per biological replicate, N = 3–5 biological replicates. **(D)** Scatterplot of fold change of average length reached between successive 5 min bins following treatment with indicated pharmacological inhibitors, relative to average length pre-treatment, for wild-type and EVL knockout DF. Mean ±95% CI. Mixed-effects model; DMSO n = 48–90 total DF, CK-666 n = 58–80 total DF, and Blebbistatin n = 88–96 total DF from 2–4 neurons per biological replicate, N = 3–5 biological replicates. **(E)** Segment of dendrite from an EVL knockout cortical neuron at D11 expressing mEmerald-Arp3, MIM-mRuby2, and iRFP670-EVL (top; scale bar = 5 μm), and kymographs of protein localization during DF initiation (middle) and motility (bottom). 5 s interval, 2 min duration, horizontal scale bar = 1 min, vertical scale bar = 1 μm, dashed line indicates dendrite. **(F)** Segment of dendrite from an EVL knockout cortical neuron at D11 expressing MRLC-mRuby2, mEmerald-EVL, and iRFP670-LifeAct (top; scale bar = 5 μm), and kymographs of protein localization during DF motility. 5 s interval, 2 min duration, horizontal scale bar = 1 min, vertical scale bar = 1 μm, dashed line indicates dendrite. **(G)** Model of spatiotemporal dynamics of MIM, Arp2/3, myosin-II, and EVL during DF initiation and protrusive motility. P-values indicated above each plot in C and D assess fold change over time within WT or KO. P values represented by asterisks on the plots assess difference between WT and KO fold change speed (C) or length (D); *P < 0.05, **P < 0.01, ***P < 0.001, ****P < 0.0001, only P < 0.05 values are shown on the plot (all P values are in Table S2). See also Video 9.

Parker et al.

EVL and MIM regulate dendritic filopodia dynamics

**Journal of Cell Biology**    15 of 26

(Portera-Cailliau et al., 2003; Dent et al., 2007; Kwiatkowski et al., 2007). Prior works implicated VASP in the genesis of neuronal filopodia (Lin et al., 2013, 2007). Due to the examination of early timepoints (D5-D7) in these studies, reliance upon overexpression, and the diversity in composition and function of neuronal filopodia throughout development, we are hesitant to conclude that VASP's role in synaptic precursor DF is well-supported. Importantly, we found that VASP expression is very low in cortical neurons compared to EVL and MENA. Interestingly, MENA knockdown, in contrast to EVL knockdown, resulted in elongated DF with higher motility, suggesting that MENA plays a role divergent from that of EVL in DF. Furthermore, MENA's function at the growth cone is well established (Lanier et al., 1999; Lebrand et al., 2004; McConnell et al., 2016), and in our work MENA overexpression promoted fan-like dendritic protrusions reminiscent of growth cones. We proposed that MENA and VASP are important for conventional filopodia in neurons and growth cones, while EVL is uniquely responsible for DF.

**EVL recruitment and actin polymerization activity are necessary and sufficient for promoting DF motility**
EVL's function in DF, much like the DF itself, is best examined through the lens of lamellipodia biogenesis. DF are protrusions constructed of branched actin (Korobova and Svitkina, 2010), and their dynamics, like lamellipodia, are governed by Arp2/3 actin assembly and base-directed myosin-II contractility (Korobova and Svitkina, 2010; Tatavarty et al., 2012; Burnette et al., 2011). Notwithstanding the important role of Ena/VASP proteins as critical actin regulators at the leading edge and filopodia tips (Applewhite et al., 2007; Bear et al., 2002; Damiano-Guercio et al., 2020), EVL's function in membrane protrusion is not well understood. Like its sister proteins, EVL's activity is defined by two stages: (1) recruitment, which constitutes the activation step of Ena/VASP proteins, and (2) processive actin polymerization. Recruitment, primarily through the EVH1 domain, targets EVL to polymerization sites where it binds actin and adds actin monomers, which are recruited as a profilin:G-actin complex to the growing barbed end (Bear and Gertler, 2009). To investigate the involvement of EVL in DF, we created a panel of functional mutants that target different components of these two stages.

We demonstrated that the EVH1 domain is required for EVL's tip localization and function in DF. The CAAX-iLID system allowed us to circumvent the requirement for the EVH1 domain by direct optogenetic recruitment of ΔEVH1-iLID to the membrane. This acute localization was sufficient to restore DF polymerization dynamics and length in EVL knockout neurons, albeit only transiently during the time of stimulation. Furthermore, we found that deletion of EVL's profilin binding site attenuated fast protrusion but did not eliminate dynamics. Disruption of both profilin- and G-actin binding (ΔActinPoly), or actin binding entirely (ΔGAB-FAB), fully inhibited motility. In our hands, the GAB domain was dispensable for EVL's function, in contrast to previous evidence that VASP-mediated actin polymerization and barbed end association requires the GAB domain (Hansen and Mullins, 2010; Breitsprecher et al., 2011; Applewhite et al.,

2007). However, high concentrations of profilin:G-actin bypass GAB's requirement (Hansen and Mullins, 2010), which may be the case in DF due to their small volume and high actin density. We have previously shown that EVL preferentially utilizes profilin-II to elongate actin filaments in breast cancer cells (Mouneimne et al., 2012). Intriguingly, profilin-II deficiency results in altered spine density and synaptic dysfunction (Ackermann and Matus, 2003; Michaelsen et al., 2010; Michaelsen-Preusse et al., 2016), phenotypes that are in line with compromised EVL function. Together, our findings support a model in which EVL is a central regulator of actin polymerization dynamics in DF.

**MIM plays a critical role in supporting EVL function in DF**
I-BAR family proteins, including MIM and IRSp53, promote outward membrane curvature at sites of PIP$_2$ enrichment, setting in motion a biophysical positive feedback loop driving PIP$_2$ clustering and further I-BAR oligomerization (Saarikangas et al., 2009, 2015; McCusker, 2020; Zhao et al., 2013; Mattila et al., 2007). This establishes an activity hub for PIP$_2$-responsive actin regulators to promote actin assembly and protrusion (Senju et al., 2017; Saarikangas et al., 2015). I-BAR proteins directly bind and recruit nucleation-promoting factors to activate Arp2/3-mediated actin nucleation (Lin et al., 2005; Mattila et al., 2003; Woodings et al., 2003). Recent papers identified MIM as an important regulator of DF in Purkinje cells of the cerebellum and cortical neurons (Kawabata Galbraith et al., 2018; Saarikangas et al., 2015). Our work advances these findings through identification of a tip-localized EVL:MIM complex in developing neurons that promotes DF initiation and elongation, and propose that MIM establishes hotspots of active Arp2/3 at nascent protrusions, as well as recruits and clusters EVL, to promote DF initiation and elongation.

Our findings demonstrate that MIM enrichment to foci on the dendrite licenses a proto-protrusion for elongation by EVL. During spontaneous or EVL iLID-elicited DF initiation, coincident enrichment of MIM and EVL immediately precedes protrusion, and MIM is a requisite first step in this process. When overexpressed in wild-type neurons, EVL predominantly elongates existing protrusions, while with MIM co-overexpression, it also substantially increases the initiation of de novo DF. This suggests that EVL alone is not sufficient for DF initiation, and is limited by the availability of suitable sites along the membrane. Although MIM knockdown did not reduce DF dynamics, acute inhibition of MIM using MITO-iLID significantly inhibited motility. This suggests that tip-localized MIM continues to promote actin polymerization in DF beyond initiation. EVL interacts with the poly-proline region of MIM through its EVH1 domain, and the two proteins are co-localized at the tips of DF during elongation. EVL knockout neurons co-expressing ΔEVH1-EVL and MIM exhibited a partial restoration of motility and morphology, while expression of ΔEVH1-EVL alone was incapable of rescue. Although knockdown of MIM in EVL knockout neurons strikingly reduced protrusion density in our study, it did not eliminate DF in wild-type neurons, in agreement with observations in MIM knockout mice (Kawabata Galbraith et al., 2018; Saarikangas et al., 2015; Minkeviciene et al., 2019), and suggests

functional redundancy with other I-BAR proteins. Collectively, these findings suggest that I-BAR proteins creates a permissive environment for EVL's actin polymerization activities, through the direct recruitment of EVL.

### Potential intersection between the role of EVL and formin in driving DF

Actin elongation factors, Arp2/3, and actin-binding proteins collaborate and compete in distinct spatiotemporal complexes to maintain actin homeostasis during cellular activities (Rotty et al., 2015; Suarez et al., 2015; Beli et al., 2008; Skruber et al., 2020; Suarez and Kovar, 2016; Barzik et al., 2014). Suppression of branched actin decreases the number of actively polymerizing barbed ends, resulting in a scenario where most of the actin monomers are channeled into linear actin polymerization (Bailly et al., 1999; Ressad et al., 1999; Burke et al., 2014; Rotty et al., 2015; Suarez and Kovar, 2016). This results in a sharp increase in the ratio of elongation factors to polymerizing barbed ends in DF, and hence longer DF with faster polymerization rates (Spence et al., 2016; Kawabata Galbraith et al., 2018). Therefore, it is not surprising that EVL knockout did not abolish DF motility following Arp2/3 inhibition, since very few elongation factors are needed after acute suppression of Arp2/3, and formins or MENA are potentially in enough abundance to supply those activities. Further, loss of Arp2/3 activity shifts actin nucleation activities to formins, and although Ena/VASP proteins can maintain actin dynamics in these conditions (Rotty et al., 2015), loss of both EVL and Arp2/3 activity leaves formins as the sole mediator of actin nucleation and subsequent elongation.

Formins are a class of actin elongation factor and nucleator that have been implicated in DF morphogenesis. The formin mDia2 has been shown to regulate dendritic protrusion density in cortical neurons (Hotulainen et al., 2009; Spence et al., 2016), and overexpressed constitutively active mDia2 localizes to DF tips (Hotulainen et al., 2009). However, whether formins are primarily acting as an actin nucleator, or elongator of Arp2/3-nucleated filaments in DF or proto-protrusions has not been established. Intriguingly, Galbraith et al. demonstrate that MIM binds the formin DAAM1 and suppresses its activity in Purkinje cell DF, while promoting activation of Arp2/3 (Kawabata Galbraith et al., 2018). It is tempting to hypothesize that MIM, a multi-domain scaffolding protein, could exhibit several conformational and activity states to dynamically regulate its interaction network. Could a MIM:formin complex be responsible for initiation of actin nucleation at the proto-protrusion, then disruption by EVL:MIM binding in response to local Arp2/3 activation promote rapid elongation of actin filaments? We suggest two possible, non-exclusive models: (1) nucleation by formins and Arp2/3 occurs in the initial protrusion and DF base, and EVL-mediated actin elongation and protrusive motility at the tip, and/or (2) formins and EVL elongate Arp2/3-nucleated filaments in parallel or under distinct regulatory mechanisms. Given the global formin inhibitor SMIFH2 demonstrates off-target inhibition of myosin isoforms (Nishimura et al., 2021), further work using specific targeting of formin paralogs is needed to clarify the relationship between formins and other actin regulators at DF.

### Myosin regulates DF length in an EVL-dependent manner

In DF, base-directed contractility by non-muscle myosin-II regulates retrograde actin flow in balance with tip-directed polymerization; this is critical to DF motility and length maintenance (Tatavarty et al., 2012; Chazeau et al., 2015; Marchenko et al., 2017). Although suppression of myosin-II slightly decreased velocity, it significantly increased DF length. It is well established that myosin-II inhibition increases lamellipodial protrusions (Vicente-Manzanares et al., 2007). Considering that DF are structurally similar to lamellipodial protrusions, increased DF length with myosin-II suppression is not surprising, and is in agreement with reports in neuronal filopodia (Marchenko et al., 2017). We posit that the increase in protrusion is a result of a decrease in actin retrograde flow when contractility is suppressed, as previously reported (Atakhani et al., 2019; Lin et al., 1996). This is consistent with our results showing that the increase in DF length by blebbistatin is not observed in EVL knockout neurons and further demonstrates that EVL is a significant promoter of actin elongation in DF.

### Regulation of DF by EVL has potential implications for neuronal plasticity

Triple knockout mice, where the expression of all three paralogs of Ena/VASP is lacking, exhibit profound defects in cortical development (Dent et al., 2007; Kwiatkowski et al., 2007). Notably, these defects were associated with compromised migration of cortical neurons, which is consistent with earlier work showing that the axon pathfinding is disrupted in MENA knockout and MENA/VASP double knockout mice (Lanier et al., 1999; Menzies et al., 2004). On the other hand, EVL knockout mice are viable and display no gross abnormalities (Kwiatkowski et al., 2007). Together, these findings suggest that MENA and VASP, distinct from EVL, are decidedly involved in mediating large-scale cellular remodeling events, while EVL is not. Accordingly, what is EVL doing during neuronal development in the brain? The studies presented here provide evidence that among the Ena/VASP proteins, EVL plays a unique role in promoting DF, and thereby regulating structural plasticity and remodeling.

The contribution of DF to synaptogenesis in early neurodevelopment is well established (Ziv and Smith, 1996; Fiala et al., 1998; Portera-Cailliau et al., 2003), and DF continue to shape neural connectivity later in life (Sala and Segal, 2014; Zuo et al., 2005). Experience-dependent plasticity and memory is made possible through the genesis, turnover, and structural remodeling of synapses. DF emanate from the dendrites in response to specific extracellular cues and intracellular signaling cascades (Jourdain et al., 2003; Petrak et al., 2005), and their tentative nature is thought to permit broad sampling of appropriate axon targets (Lohmann and Bonhoeffer, 2008; Ozcan, 2017). However, EVL knockout neurons lack DF in vitro yet exhibit enhanced synaptogenesis, and EVL knockout mice are overtly normal (Kwiatkowski et al., 2007; though to date, have not been characterized behaviorally). What then is the evolutionary rationale for a dedicated molecular pathway that uniquely regulates DF behavior, if not required for synapse formation? Fine-tuning of neural connectivity requires a cadre of actin regulators to promote

assembly and disassembly of synapses. It is intriguing to consider whether the dynamic instability of DF, and the synapses derived from them, provide an opportunity for rapid synapse formation and how this may influence learning. It will be incredibly exciting to determine where, when, and under what circumstances EVL influences synaptic plasticity in vivo.

## Materials and methods
### Animals
All animal procedures were performed in accordance with the regulations and protocols of the University of Arizona Institutional Animal Care and Use Committee. On embryonic day 17, pregnant mice were euthanized using $CO_2$ asphyxiation. EVL knockout mouse strain was a gift from Frank Gertler (Massachusetts Institute of Technology, Cambridge, MA, USA), generated as described (Kwiatkowski et al., 2007). In brief, a targeting vector disrupting exon 2 and 3 of *Evl* was electroporated into R1 embryonic stem cells for homologous recombination, and a germline clone was isolated (confirmed by Southern blot and Western blot [Kwiatkowski et al., 2007]). EVL knockout strains were backcrossed six times with C57B/6J to increase congenic status. Wild-type and EVL knockout parents for generating embryonic primary cortical neuron cultures were derived from littermates from heterozygous crossings.

### Cell culture
For primary cortical neuron cultures, the cortex was isolated from brains of embryos of either sex, dissociated, and cultured by methods described previously (Kaech and Banker, 2006). Cortical tissues were diced and digested for 20 min at 37°C in a solution of 0.125% Trypsin-EDTA (25-053-Cl; Corning) and 1.0% (wt/vol) DNase (9003-98-9; Bioline) in calcium- and magnesium-free 1X HBSS (14065-056; Gibco). Following three 5-min rinses in HBSS, digested tissue was triturated 20 times each with three P1000 pipette tips cut to progressively smaller gauges. The homogenate was passed through a 70 µm filter, counted, and cryostored as described (Parker et al., 2018). Isolated cells were resuspended in CryoStor CS10 (210102; BioLife Solutions) to 6 million cells per ml, aliquoted, and cryostored at –80°C for at least 2 d then transferred to liquid nitrogen. Cryostored cells were used in all assays except mass spectrometry.

After dissection or upon thawing, primary neuronal cells were plated on standard tissue culture dishes or #1.5 coverglass-bottom dishes (P35G-1.5-14-C; MatTek) that were coated overnight with 0.001% poly-L-lysine (P4707; MilliporeSigma, diluted in water 1:10) and washed three times for 10 min each with water. Cells were seeded at $7.8 \times 10^3/cm^2$ for low-density cultures for live-imaging and immunofluorescence to permit examination of freely motile DF that are not in contact with neighboring axons and $26.0 \times 10^3/cm^2$ for high-density cultures for protein and RNA samples. Cells were initially plated in Plating Media, containing 5% FBS (97068-085; VWR) and 0.6% (wt/vol) D-glucose (BP350-500; Thermo Fisher Scientific) in MEM with Earle's salts and L-glutamine (10-010CV; Corning). 2–4 h after plating, Plating Media was exchanged for Neuronal

Maintenance Media, containing 2% NeuroCult SM1 supplement (05711; StemCell Technologies), 1% L-glutamine (25-005CI; Corning), and 1% penicillin-streptomycin (30-002CI; Corning) in Neurobasal Media (21103049; Thermo Fisher Scientific). Primary neuron cultures are maintained at 37°C and 5% CO2. One-half volume media exchange was given on day four (D4), and one-third media exchanges every 3 to 4 d thereafter, using Neuronal Maintenance Media pre-conditioned on glial cells maintained in a separate culture, and supplemented with 2% NeuroCult and 5 µM cytosine arabinoside (C6645; AraC, MilliporeSigma) to curb glial cell proliferation. One-third media exchange was given prior to imaging, and all experiments were performed on D11 unless indicated otherwise.

For lentivirus production, HEK293T (a gift from Joan Brugge, Harvard Medical School, Cambridge, MA, USA) were cultured in 10% FBS (97068-085; VWR), 1% penicillin-streptomycin (30-002CI; Corning), and 1% L-glutamine (25-005CI; Corning) in high glucose DMEM with sodium pyruvate (10-013-CV; Corning). Plastic cell culture dishes were coated with poly-D-lysine (P0899; MilliporeSigma, 0.1 mg/ml) to promote cell adhesion during lentiviral production. Cells are maintained at 37°C and 5% CO2. 60% confluent HEK293T cells were transfected using PEI (23966; Polysciences) in OptiMEM (31985070; Thermo Fisher Scientific) as previously described (Yang et al., 2017) with transfer plasmid, pMD2.G, and psPAX2 (Didier Trono, École Polytechnique Fédérale de Lausanne, Lausanne, Switzerland; Addgene #12259 and #12260; RRID:Addgene_12259 and RRID: Addgene_12260) at a 1:0.25:0.75 µg plasmid ratio and 3:1 PEI: DNA ratio. Viral supernatant was collected 48–72 h posttransfection, filtered through 0.45 µm filters, and added directly to primary neuron cultures. All lentiviruses were added 4 d before experiments to minimize effects on other aspects of neuronal morphogenesis, with the exception of LifeAct viruses which were added at the time of plating. No antibiotic selection was used in transduced neuronal cultures.

Pharmacological Agents to induce expression of pCW57.1 mCherry-FP4/AP4-MITO, 1 µg/ml doxycycline (D9891; MilliporeSigma) was added to culture media 12 h before imaging experiments. Arp2/3 nucleation inhibitor CK-666 (100 µM in DMSO, SML0006; MilliporeSigma) or blebbistatin (50 µM in DMSO, 203390; MilliporeSigma) was acutely added during live-imaging.

### Plasmids and cloning
#### Sources
NEB Stables (C3040H; New England BioLabs) were used to propagate all lentiviral transfer plasmids to reduce recombination. NEB5α (C2987H; NEB) were used to propagate any non-lentiviral expression plasmids. Plasmids containing source cDNA sequences were gifts as follows: *Mus musculus* EVL, MENA, and ARP3 (Frank Gertler; Carl et al., 1999), *Mus musculus* GFP-ΔIBAR-MIM (Mineko Kengaku, Kyoto University, Kyoto, Japan; Kawabata Galbraith et al., 2018), *Homo sapiens* MRLC (Tom Egelhoff, Cleveland Clinic, Cleveland, OH; Beach et al., 2011; RRID:Addgene 35680), PSD95.FingR-eGFP-CCR5TC (Don Arnold, University of Southern California, Los Angeles, CA, USA; Gross et al., 2013; RRID:Addgene_46295). Plasmid containing

source cDNA sequence for *Mus musculus* MIM (NM_144800) with C-terminal myc- and FLAG-tag was purchased from Origene (MR210506). All plasmids were sequenced to confirm the correct coding sequence (Eton Bioscience).

## Cloning

All-in-one doxycycline-inducible lentiviral transfer plasmids (pLV-Dox mCherry-FP4/AP4-MITO) were generated by PCR and subcloning mCherry-FP4-MITO or mCherry-AP4-MITO (a gift from James Bear, University of North Carolina, Chapel Hill, NC, USA; Bear et al., 2000) in place of Cas9 in pCW-Cas9 (a gift from Eric Lander and David Sabatini (Massachusetts Institute of Technology, Cambridge, MA, USA; Wang et al., 2014; RRID: Addgene_50661). Briefly, Cas9 was cut out and replaced with a multiple cloning site by annealed oligo cloning, and AgeI-BamHI was used to insert FP4/AP4 sequences. TurboRFP-expressing pLKO transfer plasmid was generated by PCR and subcloning TurboRFP in place of the puromycin resistance gene at BamHI-KpnI in pLKO.1 - TRC cloning vector (a gift from David Root, The Broad Institute, Cambridge, MA, USA; Moffat et al., 2006]; RRID: Addgene_10878). Validated *Mus musculus* shRNA sequences were obtained from The RNAi Consortium (The Broad Institute via MilliporeSigma; Moffat et al., 2006) and oligos were ligated between the AgeI-EcoRI sites (replacing the 1.9 kb stuffer) of our pLKO.1 TurboRFP cloning vector using annealed oligo cloning. shRNA sequences and source MilliporeSigma product number are given in Table S1. Lentiviral LifeAct expression vectors were published previously (Padilla-Rodriguez et al., 2018; Parker et al., 2018): pLenti LifeAct-EGFP BlastR (RRID:Addgene_84383), pLenti-LifeAct-mRuby2 BlastR (RRID:Addgene_84384), pLenti LifeAct-iRFP670 BlastR (RRID:Addgene_84385). Lentiviral cDNA expression vectors were generated by PCR and subcloning cDNA of interest into the transfer plasmids pLenti CMVie-IRES-BlastR or pLenti CMVie-IRES-BlastR alt MCS (pCIB) published previously (Puleo et al., 2019; RRID:Addgene_119863 and RRID: Addgene_120862). To reduce expression of EVL, the CMVie promoter was replaced with the Ef1a short promoter EFS in some constructs. cDNAs were tagged with mEmerald (a gift from Michael Davidson, Florida State University, Tallahassee, FL, USA; RRID:Addgene_53975), mRuby2 (a gift from Michael Davidson; Lam et al., 2012; RRID:Addgene_54768), or piRFP670 (a gift from Vladislav Verkhusha, Albert Einstein College of Medicine, The Bronx, NY, USA; Shcherbakova and Verkhusha, 2013; RRID:Addgene_45457) on the N-terminus of EVL or Arp3, or C-terminus of MTSS1 or MRLC. To generate lentiviral expression vectors for PSD95-FingR-EGFP and -mRuby2, pCAGGs-PSD95-FingR-EGFP was cut with SpeI-AgeI and ligated into pCIB, and mRuby2 was subcloned into SmaI-AgeI sites. For EVL iLID and MIM iLID optogenetic systems, tgRFPt-SspB(R73Q) was PCRed and subcloned from pLL7.0 mTiam1(64–437)-tgRFPt-SSPB R73Q (a gift from Brian Kuhlman, University of North Carolina, Chapel Hill, NC, USA; Guntas et al., 2015; RRID:Addgene_60418) and added to the N-terminus of EVL or C-terminus of MTSS1. Lentiviral expression vectors for MIM-mRuby2, MIM-iRFP670, and MIM iLID were modified to remove the IRES-blasticidin resistance cassette to reduce plasmid size. Additional mammalian expression vectors for co-immunoprecipitation experiments were subcloned into pEF1-MCS-mychis6 B (V92120; Invitrogen) or pCMV 3xFLAG-MCS (Parker et al., 2013). Mutants of EVL and MIM were generated by PCR or inverse PCR and self-ligation cloning. All plasmids created for this paper will be made available through Addgene where possible. Complete list of plasmids generated and used in this paper can be found in Table S1.

## mRNA extraction and RT-qPCR

Total RNA content was isolated from primary neuron cultures using Trizol (15596026; Life Technologies) and Direct-zol RNA MiniPrep kit (R2050; Zymo Research), according to the manufacturer's instructions, including the optional DNase digestion step to eliminate genomic DNA in the sample. cDNA was synthesized using 1,000 ng of input RNA and qScript cDNA Supermix (84033; Quantabio) according to the manufacturer's instructions. RT-qPCR reactions were run in triplicate using an ABI 7500 Fast Real-Time PCR System (Applied Biosystems) and Apex qPCR 2X Master Mix Green, Low ROX (Apex BioResearch Products, 42-119 PG). Primer pairs were confirmed to have 85–110% efficiency, which was determined from the slope of the best fit curve for the CT of cDNA serial dilutions. CT values were normalized to the average of three control genes: Gapdh, Eef1a1, and Rpl29. Relative copy numbers were determined using comparative CT method ($2^{-\Delta CT}$ Schmittgen and Livak, 2008). Primer sequences are as follows: Mouse *Eef1a1* Fwd: 5′-CAACAT CGTCGTAATCGGACA-3′ Rev: 5′-GTCTAAGACCCAGGCGTACT T-3′ (Parker et al., 2018), Mouse *Gapdh* Fwd: 5′-AGGTCGGTG TGAACGGATTTG-3′ Rev: 5′-GGGGTCGTTGATGGCAACA-3′ (Parker et al., 2018), Mouse *Rpl29* Fwd: 5′-CAAGTCCAAGAACCACAC CAC-3′ Rev: 5′-GCAAAGCGCATGTTCCTCAG-3′ (Parker et al., 2018), Mouse *Enah* Fwd: 5′-CTGGTGGCTCAACTGGGTTC-3′ Rev: 5′-TGCCCACAACTCTGAATGTGT-3′ (PrimerBank, Harvard), Mouse *Evl* Fwd: 5′-TGAGAGCCAAACGGAAGACC Rev: TTCTGGACAGCAACGAGGAC-3′ (Puleo et al., 2019), Mouse *Vasp* Fwd: 5′-CGGGCTACTGTGATGCTTTATG-3′ Rev: 5′-TAG CAGTGGGGTTGTGGTAGA-3′ (PrimerBank, Harvard).

## Western blotting

Cells were lysed with ice cold lysis buffer buffer (25 mM HEPES, pH 7.4, 150 mM NaCl, 1% NP-40, 0.25% Na Deoxycholate, 10% glycerol) supplemented with protease and phosphatase inhibitor cocktails (539134; MilliporeSigma; BP-479; Boston BioProducts), and incubated on ice for 5 min. Cellular debris was pelleted out by centrifugation at 21,000 *g* for 15 min at 4°C. Protein concentration was determined using a Bradford assay compared to BSA protein standards. Supernatant was mixed with Laemmli buffer and boiled at 95°C for 5 min. To determine protein expression, 20–30 μg of protein lysate was loaded per well. For immunoprecipitation blots, 5% of input lysate and entire IP eluate was loaded. Samples were resolved using SDS-PAGE and 10% acrylamide gels. Resolved samples were transferred to nitrocellulose membranes (926-31090; LI-COR), and membranes were blocked in Intercept Blocking Buffer (927-70001; LI-COR) for 1 h rocking at room temperature. Membranes were probed overnight at 4°C with primary antibodies diluted in blocking

buffer and 0.2% Tween-20. After washing in TBS with 0.1% Tween-20, membranes were probed with near-infrared fluorescent secondary antibodies diluted in blocking buffer and 0.2% Tween-20 for 1 h at room temperature. Membranes were scanned using an Odyssey CLX imager (LI-COR) and quantified in Image Studio Lite (LI-COR). Antibodies were as follows: rabbit polyclonal anti-EVL (a gift from Frank Gertler, 1:1,000), rabbit polyclonal anti-ENAH (HPA028696; Sigma Prestige, RRID: AB_10611249; 1:250), rabbit monoclonal anti-VASP (#3132; Cell Signaling, RRID:AB_2213393; 1:500), rabbit polyclonal anti-MTSS1 (PA5-23200; Thermo Fisher Scientific, RRID: AB_2540726; 1:500) rabbit polyclonal anti-FLAG (20543-1; ProteinTech Group, RRID:AB_11232216; 1:1,000), rabbit polyclonal anti-GFP (66002-1; ProteinTech Group, RRID:AB_11182611; 1:1,000), mouse monoclonal anti-actin (66009-1; ProteinTech Group, RRID:AB_2687938; 1:5,000), goat anti-mouse Alexa Fluor 680 (#A-21057; Thermo Fisher Scientific, RRID:AB_2535723; 1:20,000) and goat anti-rabbit Alexa Fluor 790 secondary antibodies (#A-11367; Thermo Fisher Scientific, RRID:AB_2534141; 1:20,000).

## Immunoprecipitation and mass spectrometry

For immunoprecipitation-western blot experiments, 10 cm plates of 60% confluent HEK293T were transfected with 4 μg of each plasmid (total of 8 μg), and 24 μg PEI. 24 h post transfection, cells were lysed with ice cold IP buffer (10% glycerol, 1% NP-40, 50 mM Tris, pH 7.5, 200 mM NaCl, 2 mM MgCl2) supplemented with protease and phosphatase inhibitor cocktails (539134; MilliporeSigma; BP-479; Boston BioProducts) and incubated on ice for 5 min. Cellular debris was pelleted out by centrifugation at 21,000 $g$ for 15 min at 4°C. Protein concentration was determined using a Bradford assay compared to BSA protein standards, and 1–2 mg of protein was loaded into each IP reaction. Lysates were pre-cleared in 5 μl of magnetic Protein A/G bead slurry (88802; Thermo Fisher Scientific), and 2.5% of the volume of whole cell extract was set aside. Reactions were incubated overnight at 4°C with gentle rotation with mouse monoclonal anti-FLAG (66008-3; ProteinTech Group, RRID: AB_2749837; 5 μg antibody per mg protein). 25 μl magnetic Protein A + G bead slurry was added to each reaction, and incubated for 2 h at 4°C with gentle rotation. Beads were washed three times in IP buffer with 0.1% Tween-20, and proteins were eluted from beads by boiling at 95°C for 10 min in 2X Laemmli buffer diluted in IP buffer. Samples were resolved by SDS-PAGE and Western blotting as described above.

For affinity purification-mass spectrometry experiments, two poly-L-lysine-coated 15 cm plates were each plated with 10 million primary neuron cells. Cells were lysed on D11 and protein samples were handled as described above. Reactions were incubated overnight at 4°C with gentle rotation with mouse control IgG (sc-2025; Santa Cruz) or mouse monoclonal anti-EVL (a gift from Frank Gertler) at 5 μg antibody per mg protein. 25 μl magnetic Protein A/G bead slurry (88802; Thermo Fisher Scientific) was added to each reaction, and incubated for 2 h at 4°C with gentle rotation. Beads were washed three times in IP buffer, and proteins were eluted from beads by boiling at 95°C for 10 min in 2X laemmli buffer diluted in IP buffer. Entire

eluate was loaded onto Bolt 4–12% Bis-Tris pre-cast gels (Thermo Fisher Scientific, NW04120BOX), resolved by SDS-PAGE, and the gels were stained with Bio-Safe Coomassie G-250 Stain (#1610786; Bio-Rad).

In-gel digestion: in-gel tryptic digestion was performed as previously described (Kruse et al., 2017). In brief, each lane of the SDS-PAGE gel was cut into eight slices. Each gel slice was placed in a 0.6 ml LoBind polypropylene tube (Eppendorf), destained twice with 375 μl of 50% acetonitrile (ACN) in 40 mM NH$_4$HCO$_3$ and dehydrated with 100% ACN for 15 min. After removal of the ACN by aspiration, the gel pieces were dried in a vacuum centrifuge at 60°C for 30 min. Trypsin (250 ng; Sigma-Aldrich) in 20 μl of 40 mM NH$_4$HCO$_3$ was added, and the samples were maintained at 4°C for 15 min prior to the addition of 50–100 μl of 40 mM NH$_4$HCO$_3$. The digestion was allowed to proceed at 37°C overnight followed by termination with 10 μl of 5% formic acid (FA). After further incubation at 37°C for 30 min and centrifugation for 1 min, each supernatant was transferred to a clean LoBind polypropylene tube. The extraction procedure was repeated using 40 μl of 0.5% FA, and the two extracts were combined and dried down to ~5–10 μl followed by the addition of 10 μl 0.05% heptafluorobutyric acid/5% FA (vol/vol) and incubation at room temperature for 15 min. The resulting peptide mixtures were loaded on a solid phase C18 ZipTip (Millipore) and washed with 35 μl 0.005% heptafluorobutyric acid/5% FA (vol/vol) followed by elution first with 4 μl of 50% ACN/1% FA (vol/vol) and then a more stringent elution with 4 μl of 80% ACN/1% FA (vol/vol). The eluates were combined and dried completely by vacuum centrifugation and 6 μl of 0.1% FA (vol/vol) was added followed by sonication for 2 min. 2.5 μl of the final sample was then analyzed by mass spectrometry.

## Mass spectrometry and database search

HPLC-ESI-MS/MS was performed in positive ion mode on a Thermo Fisher Scientific Orbitrap Fusion Lumos tribrid mass spectrometer fitted with an EASY-Spray Source as previously described (Parker et al., 2019). NanoLC was performed using a Thermo Fisher Scientific UltiMate 3000 RSLCnano System with an EASY Spray C18 LC column (Thermo Fisher Scientific, 50 cm × 75 μm inner diameter, packed with PepMap RSLC C18 material, 2 μm, cat. # ES803); loading phase for 15 min; mobile phase, linear gradient of 1–47% ACN in 0.1% FA for 106 min, followed by a step to 95% ACN in 0.1% FA over 5 min, hold 10 min, and then a step to 1% ACN in 0.1% FA over 1 min and a final hold for 19 min (total run 156 min); Buffer A = 100% H2O in 0.1% FA; Buffer B = 80% ACN in 0.1% FA; flow rate, 250–300 nl/min. All solvents were liquid chromatography mass spectrometry grade. Spectra were acquired using XCalibur, version 2.1.0 (Thermo Fisher Scientific). A "top 15" data-dependent MS/MS analysis was performed (acquisition of a full scan spectrum followed by collision-induced dissociation mass spectra of the 15 most abundant ions in the survey scan). Dynamic exclusion was enabled with a repeat count of 1, a repeat duration of 30 s, an exclusion list size of 500, and an exclusion duration of 40 s. Tandem mass spectra were extracted from Xcalibur 'RAW' files and charge states were assigned using the ProteoWizard 2.1.x msConvert script using the default parameters. The fragment

mass spectra were searched against the 2016 *Mus musculus* SwissProt database (16838 entries) using Mascot (Matrix Science; version 2.4) using the default probability cut-off score. The search variables that were used were: 10 ppm mass tolerance for precursor ion masses and 0.5 Da for product ion masses; digestion with trypsin; a maximum of two missed tryptic cleavages; variable modifications of oxidation of methionine and phosphorylation of serine, threonine, and tyrosine. Cross-correlation of Mascot search results with X! Tandem was accomplished with Scaffold (version Scaffold_4.8.7; Proteome Software). Probability assessment of peptide assignments and protein identifications were made using Scaffold. Only peptides with ≥95% probability were considered.

### Single cell RNA-seq
Mouse and human brain single cell RNA-seq data was downloaded from the Allen Brain Map data portal (2010 Allen Institute for Brain Science. Allen Brain Map [Sunkin et al., 2013]; available from: https://portal.brain-map.org/atlases-and-data/rnaseq). In our analyses, we used read counts derived from the 2019 Smart-Seq method. The 2019 Smart-Seq data was selected due to having an overall greater read depth than the 2020 10x Genomics data. To directly compare the gene expression of *Evl*, *Enah*, and *Vasp* within mouse glutamatergic neurons, we first selected the top 1,000 glutamatergic neurons that had the greatest number of read counts. Next, we performed TPM normalization on the expression matrix using the R function "calculateTPM" from the R library "scater" (McCarthy et al., 2017). The gene lengths used in the calculateTPM function were calculated from the GTF2LengthGC.R script provided by the github user dpryan79. For the GTF and FASTA input files, gencode.vM25.annotation.gtf and GRCm38.p6.genome.fa files were downloaded from GENCODE (Frankish et al., 2019). The same pipeline was used to calculate gene expression of *EVL*, *ENAH*, and *VASP* in human glutamatergic cells, except the GENCODE files gencode.v34.annotation.gtf and GRCh38.p13.genome.fa were used to calculate gene lengths.

To compare the gene expression of *Evl*, *Enah*, and *Vasp* between glutamatergic neurons and non-neuronal cells in mice, we first selected the top 250 glutamatergic neurons and the top 250 non-neuronal cells that had the highest read counts. Next, the SCnorm pipeline was used to normalize gene expression across samples (Bacher et al., 2017). All default settings were used except the parameter "ditherCounts" was set to TRUE. The same pipeline was utilized for the comparison of *EVL*, *ENAH*, and *VASP* between human glutamatergic neurons and non-neuronal cells.

### Immunofluorescence
Neurons were seeded on poly-L-lysine-coated #1.5 coverslips (633029; Carolina Biological Supply). At indicated timepoints, coverslips were briefly washed in Dulbecco's PBS with calcium and magnesium (PBL02; Caisson Labs), and fixed for 15 min with 4% paraformaldehyde (diluted in PBS from 16%, Electron Microscopy Services, 15710) and 4% sucrose in PBS at 37°C. For immunolabeling of fascin, coverslips were washed and fixed for 10 min with ice-cold methanol at −20°C. Subsequent steps were performed at room temperature. Autofluorescence was

quenched by incubation with 50 mM NH$_4$Cl for 10 min at room temperature. Coverslips were blocked and permeabilized using a buffer of 10% goat serum and 0.1% Triton X-100 in PBS and incubating for 30 min. Antibodies were diluted in a buffer of 3% goat serum and 0.1% Triton X-100 in PBS. Coverslips were incubated with primary antibodies for 1 h, washed three times in PBS with 0.1% Tween-20, and incubated with secondary antibodies and Alexa Fluor 488 phalloidin (A12379; Thermo Fisher Scientific, 1:40) for 30 min. Coverslips were mounted using Prolong Gold Antifade (P36930; Thermo Fisher Scientific), and allowed to cure for at least 24 h prior to imaging. Antibodies used were as follows: mouse monoclonal anti-Arp2/3 complex (#MABT95; Millipore, RRID:AB_11205567; 1:250), mouse monoclonal anti-fascin (#54545; Cell Signaling Technology, RRID:AB_2799464; 1:50), rabbit monoclonal anti-β-III-tubulin (#5568; Cell Signaling, Clone D71G9, RRID:AB_10694505; 1:100), rabbit monoclonal anti-MAP2 (#8707; Cell Signaling, Clone D5G1, RRID:AB_2722660; 1:500), goat anti-rabbit Alexa Fluor 568 (#A-11011; Molecular Probes, RRID:AB_143157; 1:1,000), goat anti-rabbit Alexa Fluor 647 (#A-21245; Thermo Fisher Scientific, RRID:AB_2535813; 1:1,000).

### Microscopy
All images relating to DF motility and immunofluorescence were collected on a Nikon Ti-E Total Internal Reflection Fluorescence (TIRF) microscope (Nikon Instruments) equipped with an ORCA-Flash 4.0 V2 sCMOS camera (Hamamatsu), motorized stage, perfect focus system, and environmental chamber (InVivo Scientific) to maintain humidity, 37°C and 5% CO$_2$ ambient conditions. Neuron cultures were imaged in standard Neurobasal Media containing phenol red. For widefield fluorescence imaging, microscope is equipped with a SOLA solid-state LED white light source (Lumencor, 100% power), and a DAPI/FITC/CY3/CY5 excitation, emission, and dichroic filter set (89000 Sedat Quad ET, Chroma. Excitation Filters: D350/50x, ET402/15x, ET490/20x, ET555/25x, ET645/30x. Emission Filters: ET455/50 m, ET525/36 m, ET605/52 m, ET705/72 m). For TIRF imaging, microscope is equipped with a 405/488/561/640 nm laser launch (LUN4; Nikon, 15 mW, 100% power except for optogenetic activation at 10% power), Ti-TIRF-E Motorized Illuminator Unit, and utilized with C-FL TIRF Ultra Hi S/N 405/488/561/638 Quad Cube, Z Quad HC Cleanup, and HC TIRF Quad Dichroic. All images observing synapse formation and dendritic spine density were performed on a Nikon Ti2-E (Nikon) equipped with a CSU-W1 SoRa spinning disk confocal (Yokogawa), Kinetix sCMOS (Photometrics), motorized stage, perfect focus system, and an environmental chamber (Tokai). Microscope is further equipped with a 405/488/561/640 nm laser launch (Nikon, LUNF-XL, 50/60/50/40 mW respectively, 25–50% power), dichroics (100199, Chroma, ZT405/488/561/640RPCV2 MTD TIRF cube), and emission filters (ET455/50 m, ET535/30 m, ET605/52 m, ET705/72 m). Acquisition software was NIS Elements (Nikon).

### Live-cell imaging for DF motility
Primary neuron cultures were plated on poly-L-lysine-coated coverglass bottom dishes as described under Cell Culture.

Neurons were imaged using TIRF microscopy to minimize phototoxicity, improve resolution, and maximize acquisition speed. Images were acquired using a 100× Plan Apo TIRF 1.49NA objective (Nikon). Images were acquired on a 5 s interval for 5 min, except for pharmacological treatments in which three positions were acquired on a 15 s interval for 45 min, and optogenetic activation experiments which were acquired on a 5 s interval for 15 min. Images were acquired with 1 × 1 binning, or 2 × 2 binning for optogenetic activation experiments, and wherever more than two fluorescent channels were acquired (MIM + EVL + LifeAct experiments, MIM + EVL + Arp3/MRLC experiments) in order to decrease the light dose. Exposure times ranged between 70 and 200 ms at 100% laser power at TIR critical angle for each channel. For optogenetic activation experiments, whole cells were illuminated with vertical incident 488 nm laser light (non-TIRF) at 10% power for 400 ms exposure every 5 s during the stimulation phase.

### Live-cell imaging for synapse formation

Primary neuron cultures were plated on poly-L-lysine-coated coverglass bottom dishes as described under Cell Culture. Images were acquired on the CSU-W1 SoRa, using a 100X Plan Apo Lambda S NA 1.35 silicone-immersion objective (Nikon), five slice z-stacks (1.5 µm range) on a 30 s interval for 1 h.

### Fixed cells

Primary neuron cultures were plated on poly-L-lysine-coated coverslips and fixed as described under Cell Culture and Immunofluorescence. Cells were imaged in widefield fluorescence, using 40X Plan Fluor 1.3NA objective (Nikon) for neuron morphology. 5 × 5 fields of images were acquired and stitched to increase field of view.

### Software and image analysis
#### Processing

Time-lapse image stacks were processed using NIS Elements AR (Nikon). Pre-processing steps for analysis included the following workflow: correcting for drift in time using "Align ND Document," correcting for photobleaching using "Equalize Intensity in Time" to the first frame's histogram, and segmenting the neuron by merging denoised duplicates of fluorescent channels into a single binary mask, and subtracting the background. This eliminated any non-cell background from mean tip intensity measurements, which were measured from a 384 nm radius about the tracked tip. All fluorescence intensity analysis was performed on images without further processing. Image processing for figure preparation included the following additional workflow to improve signal to noise and presentation of live-cell imaging datasets: low-pass filtering (keep details larger than 2 pixels), either Noise2Void (Figs. 2, 3, and 4, using ZeroCostDL4Mic [von Chamier et al., 2021; Krull et al., 2019]) or "Advanced Denoising" (Figs. 5, 6, 7, and 8, NIS Elements), 2D Blind Deconvolution (1 iterations, low noise, calculated PSF, NIS Elements) on LifeAct channel, and 2D Richardson-Lucy Deconvolution (2 iterations, low noise, calculated PSF, NIS Elements) for all other fluorescent channels. Fig. 2 was additionally processed with Rolling Ball background subtraction (1 µm radius).

Advanced Denoising settings were adjusted as appropriate individually due to variation in construct expression levels.

### Analysis

Tip tracking was performed using Manual Tracking in NIH ImageJ (FIJI build [Schindelin et al., 2012]). DF were only tracked if they met the following conditions: no contact with axons, neighboring DF, or debris during the time course; emanated from dendrites at least 50 µm away from the center of the soma; clearly visible by brightfield during time course; if they initiated or retracted during imaging, non-existent timepoints were removed from further analysis; buckling and wagging DF were included in tracking. Using the manually tracked positions of the DF base and tip, the image files were then analyzed with a custom MATLAB script to determine the centerline path along each DF (Mendeley data hyperlink). This script used the fluorescent intensity in either the LifeAct or GFP space-filler channel in the vicinity of the tip and base coordinates to define the average tangent direction of the long axis of the DF by computing the tangent angle q at pixel i using

$$\theta_i = \frac{1}{2}\tan^{-1}\left(\frac{\langle y - y_i\rangle\langle x - x_i\rangle}{\langle x^2 - x_i^2\rangle\langle y^2 - y_i^2\rangle}\right),$$

where brackets denote the intensity-weighted average over a 15 × 15 pixel domain centered on the ith pixel. The centerline curve $(x(s), y(s))$ was then determined by solving

$$\frac{\partial^2 x}{\partial s^2} = -\sin\theta\frac{\partial\theta}{\partial s}; \frac{\partial^2 y}{\partial s^2} = \cos\theta\frac{\partial\theta}{\partial s}$$

subject to the constraint that the starting and ending positions were the tracked positions of the base and tip of the DF. Using the centerline curves for each DF at each time point, we then calculated the absolute tip displacement, DF length, and mean tip fluorescence intensity and were able to extract the following metrics: average filopodial tip speed calculated as the average of the instantaneous speeds (absolute tip displacement per 5 s interval) between successive timepoints; percent motile, percent of total DF population with average tip speeds greater than 0.0128 µm/s (motile; one pixel displacement or greater per 5 s interval) or less than 0.0128 µm/s (non-motile); percent time motile, the percent of time per DF in which instantaneous speed was greater than 0.0128 µm/s; average length, the distance from base to tip along the centerline curve, median protrusion or retraction rate, the positive or negative change in length between successive timepoints, when instantaneous change in length was greater than ±0.0128 µm/s (motile); mean fluorescence intensity for a circular area of 384 nm radius surrounding the distal DF tip with non-cell background omitted; fluorescence intensity variance, a measure of the spread of intensity values compared to the mean. Fluorescence intensity values were normalized for expression by the minimum local intensity during the duration of imaging. For defining motile versus non-motile filopodia, or substantiative protrusion/retraction rates, a threshold of 0.0128 µm/s was chosen as it represents one pixel (effective size at 100X = 0.064 µm) displacement per 5-s interval and undistinguishable from tracking error. Neurite morphology

was measured using the ImageJ plug-in Simple Neurite Tracer (Longair et al., 2011). Tracings were used to determine the number and length of primary and higher order neurites, and length of the axon (the longest Tau-positive process). Protrusion and spine density was determined by counting proturbences or dendritic spines along a length of dendrite. PSD95 foci analysis was performed by generating a binary mask of foci, and using the automated 2D tracking module in NIS-Elements (Nikon) to follow their trajectories.

## Statistical analysis

Exact P values and statistical methods per figure are listed in Table S2. All data were tested for normality by the D'Agostino-Pearson method. Outliers were not removed for any statistical comparisons. The following are the different methods used in our analysis.

### Filopodyan

We utilized the R Studio package of Filopodyan (Urbančič et al., 2017), a filopodia tracking package to determine the correlation of MENA or EVL tip fluorescence intensity to various DF metrics and the cross-correlation function and time offset. Data output from our custom MatLab script was reformatted for compatibility with Filopodyan inputs. R Studio scripts were edited for pixel size, interval, and to permit compatibility and data export. Filopodyan_Masterscript.R comprising Modules 1, 2, and 3 was used to calculate and configure input data for FilopodyanR_CCF.R, CCF_subcluster-analysis.R, and CCF_Randomisations.R. FilopodyanR_CCF.R calculates the cross-correlation function (CCF) of Tip Fluorescence (FTIP; mean fluorescence intensity at tip region) and Direction-Corrected Tip Movement (DCTM; protrusion and retraction as calculated by changing length), calculated from a moving average of three adjacent timepoints. Using unbiased hierarchical clustering, high CCF DF clusters and low CCF DF clusters were designated as the top-correlating subcluster (TCS) and non-TCS, respectively. CCF_subcluster-analysis.R was used to extract and analyze DF metrics for the TCS and non-TCS populations. CCF_Randomisations.R generated 1,000 randomized datasets by shuffling eight-timepoint-blocks of motility data with respect to tip fluorescence data for each DF, and performing identical cross-correlation analysis on datasets with similar subcluster size. Specific statistical analyses are given in the figure legends.

### Mixed-effects models

For large datasets, as indicated in Table S2, mixed models were fit, controlling for repeated measurements by cell nested within conditions. For observations in a single DF over time, data were normalized over pre-photostimulation or pre-treatment with drugs and paired for statistical analysis. All outcomes were log transformed to normal distributions.

### Non-parametric comparisons

Unpaired non-parametric comparisons were made using the Mann-Whitney test or Kruskal-Wallis test in Prism v9.1.0 (GraphPad).

### Parametric comparisons

Unpaired parametric comparisons were made using the two-sided Student's t-test or Welch's t-test in Prism v9.1.0 (GraphPad).

### Materials availability

All unique plasmids generated in this study are available upon request.

### Online supplemental material

Fig. S1 shows characterization of synapse precursor DF in cultured mouse primary cortical neurons. Fig. S2 shows that EVL is the predominant Ena/VASP paralog regulating DF. Fig. S3 shows that MENA and EVL overexpression enhance DF motility. Fig. S4 shows tip enrichment of EVL precedes DF protrusion. Fig. S5 shows that EVL is required for DF morphogenesis, and influences dendritic spine plasticity. Fig. S6 shows that EVL tip localization is necessary and sufficient for DF motility. Fig. S7 shows that MIM/MTSS1 cooperates with EVL to promote DF initiation and motility.

### Data availability

Proteomics datasets and custom MatLab script are available at the public repository Mendeley Data: "Parker et al.," Mendeley DOI:10.17632/yjbczc6twt.1. https://data.mendeley.com/datasets/yjbczc6twt/1.

## Acknowledgments

We thank Seth Zimmerman for assistance with iLID and Vasja Urbančič for assistance with Filopodyan, Jean Wilson who provided feedback on the manuscript, Konrad E Zinsmaier for his guidance, Marco Padilla-Rodriguez who provided illustrations, Gillian Paine-Murrieta and the Experimental Mouse Shared Resource (EMSR) for maintenance of mouse colonies, and Christopher Miller for assistance and expertise with the microscopes.

Research was supported by the National Cancer Institute (NCI) of the National Institutes of Health under award number P30 CA023074 (to University of Arizona Cancer Center Biostatistics and EMSR), a fellowship from Science Foundation Arizona (SSP), and NCI grant R01 CA196885-01 (GM). Open Access funding provided by the University of Arizona.

Author contributions: Conceptualization of the project was performed by S.S. Parker and G. Mouneimne. Investigation was performed by S.S. Parker, A.D. Grant, PR Langlais, and K.T. Lee. Methodology was developed by SS Parker, A.D. Grant, P.R. Langlais, and G. Mouneimne. Resources were developed by S.S. Parker. Software was developed by C.W. Wolgemuth. Analysis was performed by S.S. Parker, K.T. Lee, A.D. Grant, J. Sweetland, A.M. Wang, J.D. Parker, M.R. Roman, K. Saboda and D.J. Roe. Project was supervised by G. Mouneimne, M. Padi, P.R. Langlais, C.W. Wolgemuth, and G. Mouneimne. Visualization of data was performed by S.S. Parker, P.R. Langlais, and G. Mouneimne. Writing and revision of the manuscript was performed by S.S. Parker, K.T. Lee, A.D. Grant, K. Saboda, D.J. Roe, P.R. Langlais, C.W. Wolgemuth, and G. Mouneimne.

Disclosures: The authors declare no competing interests exist.

Submitted: 15 June 2021

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

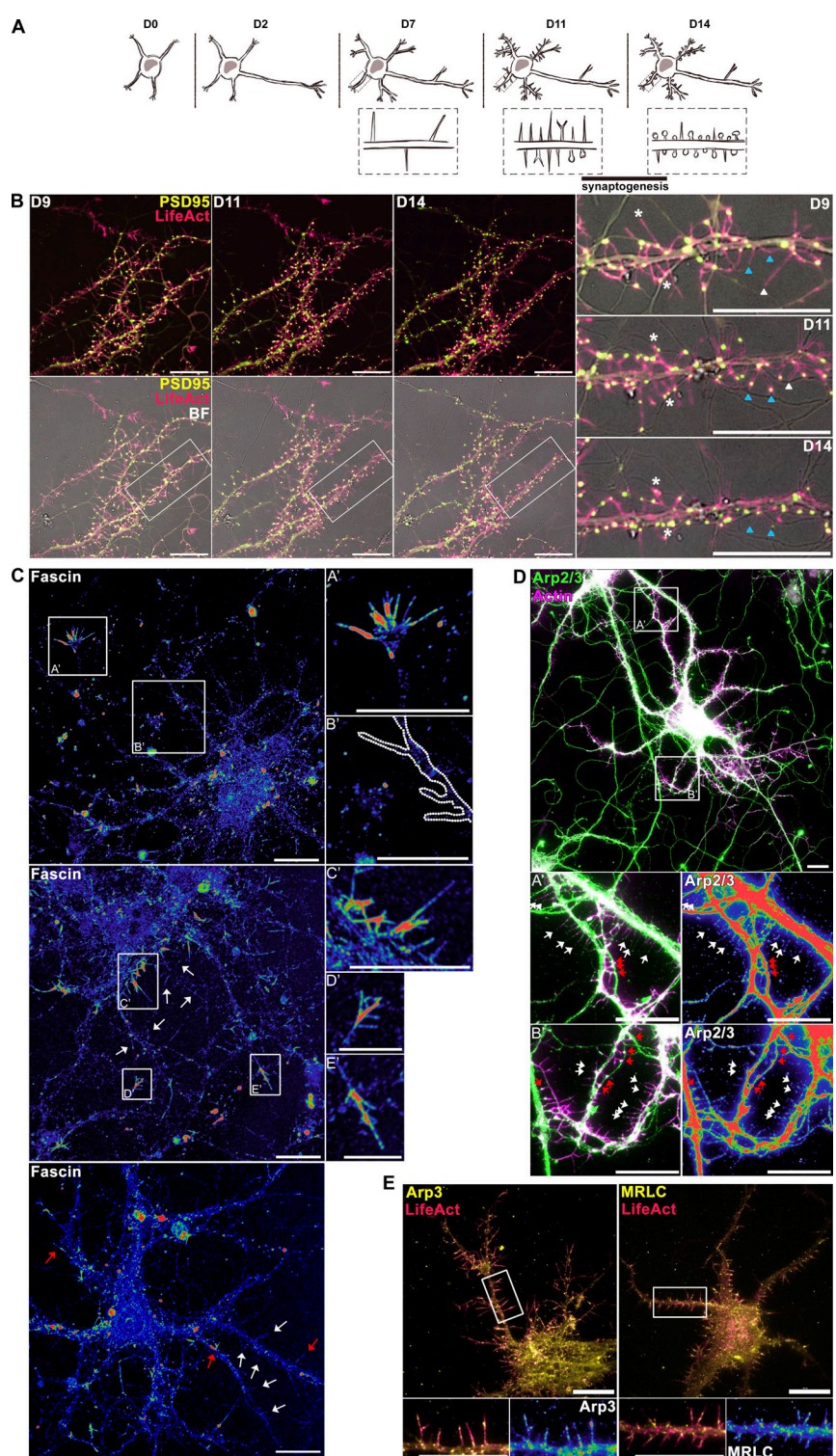

Figure S1. **Characterization of synapse precursor dendritic filopodia in cultured mouse primary cortical neurons. (A)** Schematic of in vitro cortical neuron development and synaptogenesis. **(B)** Live primary mouse cortical neurons at indicated days in vitro expressing mRuby2-LifeAct and synapse reporter PSD95-FingR-EGFP, demonstrating synapse formation through dendritic filopodia and synapse maturation (asterisks). Left: full cell image. Right: indicated segment of dendrite. **(C)** Immunofluorescence labeling of primary cortical neurons for the conventional filopodia marker fascin at D11 in vitro. Left: Intensity-coded LUT of fascin labeling in full cell image. Insets: Indicated regions showing fascin labeling at the axonal growth cone (A'), DF (B'), soma (C'), and dendritic growth cones (D' and E'). White arrows indicate fascin-negative DF, red arrows indicate fascin-positive filopodia. **(D)** Immunofluorescence labeling of primary cortical neurons for the branched actin marker Arp2/3 and phalloidin at D11 in vitro. Top: Full cell image. Insets: Indicated segments of dendrite (left) and intensity-coded LUT of Arp2/3 labeling (right). White arrows indicate dendritic filopodia, red arrows indicate synapses. **(E)** Live D11 neurons expressing iRFP670-LifeAct and mRuby2-Arp3 or MRLC-mRuby2 showing localization of Arp2/3 complex and myosin-II in dendritic filopodia. Top: Full cell image. Inset: Indicated segment of dendrite (left) and intensity-coded LUT of localization (right). Scale bars = 10 µm.

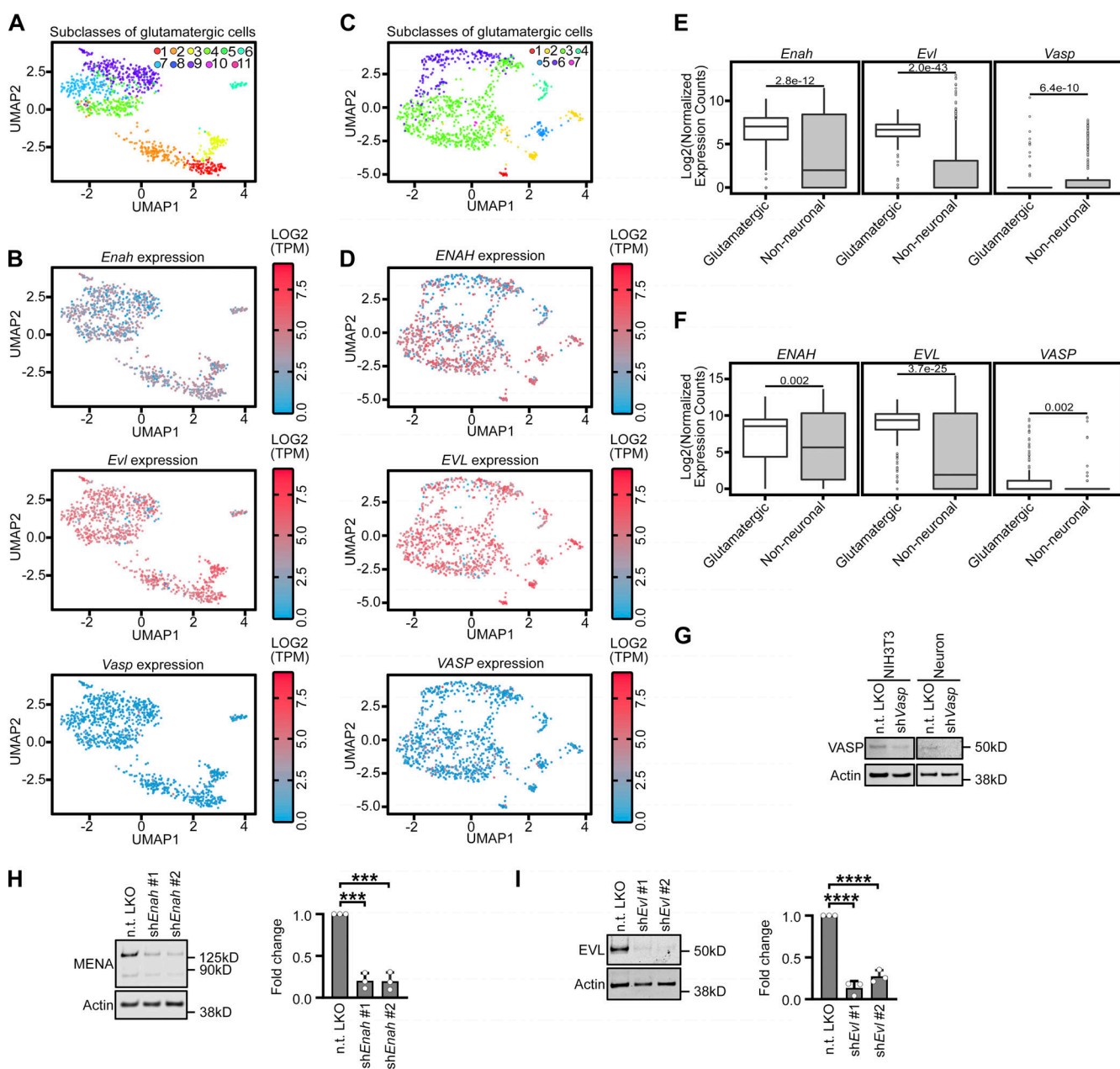

Figure S2. **EVL is the predominant Ena/VASP paralog regulating dendritic filopodia. (A)** Uniform manifold approximation and projection (UMAP) subcluster categories of single-cell RNA-seq expression profiles from mouse glutamatergic neurons. 1: L6b cortex (CTX), 2: L6 corticothalamic CTX, 3: L5 nearprojecting CTX, 4: L6 intratelencephalic (IT) CTX, 5: L5 IT CTX, 6: L5 pyramidal tract CTX, 7: L2/3 IT CTX-1, 8: Car3, 9: L4/5 IT CTX, 10: L2/3 IT CTX Ppp1r18, 11: L2/3 IT CTX-2. **(B)** Log$_2$(TPM(transcripts per million)) expression in mouse of *Enah*, *Evl*, and *Vasp* across single-cell RNA-seq UMAP subcluster categories. **(C)** UMAP subcluster categories of single-cell RNAseq expression profiles from human glutamatergic neurons. 1: L5/6 near-projecting cortex (CTX), 2: L6 corticothalamic CTX, 3: intratelencephalic (IT) CTX, 4: L5/6 IT CTX Car3, 5: L6b CTX, 6: L4 IT CTX, 7: L5 IT CTX. **(D)** Log$_2$(TPM) expression in human of *ENAH*, *EVL*, and *VASP* across single-cell RNA-seq UMAP subcluster categories. **(E and F)** Comparison of *Enah*, *Evl*, and *Vasp* expression in mouse (E) and human (F) across single-cell RNA-seq grouped by glutamatergic neurons versus non-neuronal cell types. Statistical comparisons of RNA abundance in cell type categories made by Welch's *t* test, *n* = 250 cells. **(G)** Representative western blot confirming specificity of VASP antibody in NIH3T3 (left) and primary neuron cultures (right) with pLKO-shRNA-TurboRFP targeting *Vasp* mRNA. Mean ± SD. *N* = 3 biological replicates. **(H and I)** Representative western blots of protein lysates from primary neurons at D11, transduced on D7 with indicated pLKO-shRNA-TurboRFP lentiviral particles targeting *Enah* (H; left) or *Evl* mRNA (I; left). Fold change quantification of knockdown compared to non-targeting shRNA control (right). Mean ± SD. *N* = 3 biological replicates. Statistical comparisons of shRNA knockdown efficiency were made by unpaired *t* test. *P < 0.05, **P < 0.01, ***P < 0.001, ****P < 0.0001, n.s. is not significant. See also Fig. 1 and Video 1. Source data are available for this figure: SourceData FS2.

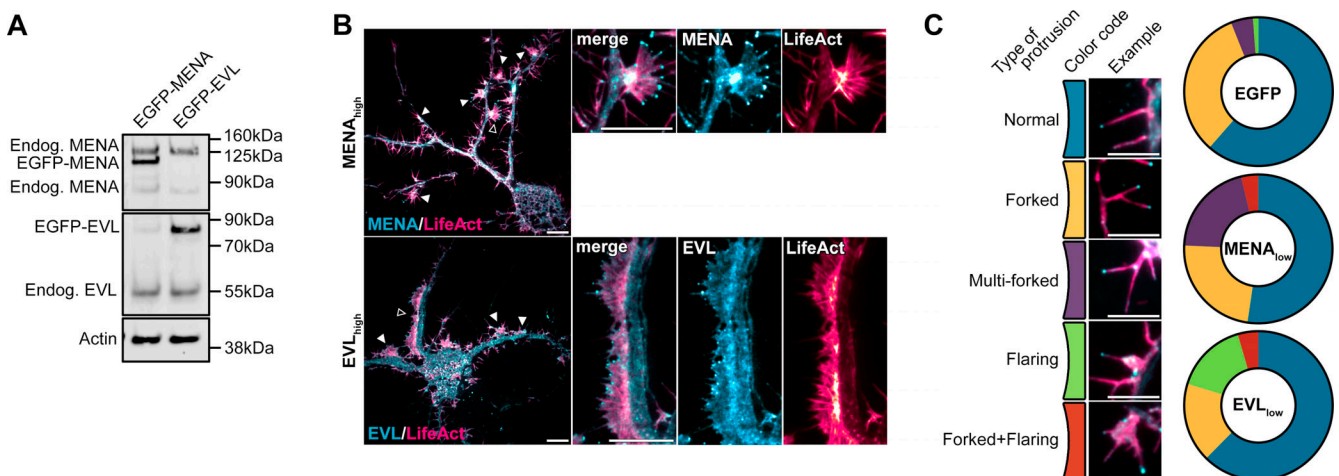

Figure S3.  **MENA and EVL overexpression enhance DF motility. (A)** Representative Western blot of protein lysates from D11 mouse cortical neuron cultures expressing EGFP-MENA or EGFP-EVL, and probed with antibodies targeting MENA or EVL for detection of endogenous and overexpressed species. **(B)** Examples of extreme phenotypes observed in live primary mouse cortical neurons at day in vitro 11 (D11) expressing mRuby2-LifeAct with high expression (SNR > 1.5) of EGFP-MENA or EGFP-EVL. Left: full cell image. Closed arrowheads indicate abnormal structures. Open arrowhead indicates magnified region (right). **(C)** Examples and quantification of DF morphological phenotypes observed in D11 neurons expressing mRuby2-LifeAct with low-expression (SNR < 1.5) of EGFP-MENA or EGFP-EVL. n = 83–133 DF per condition, N = 3 neurons. Scale bars = 10 µm. See also Fig. 2 and Video 2. Source data are available for this figure: SourceData FS3.

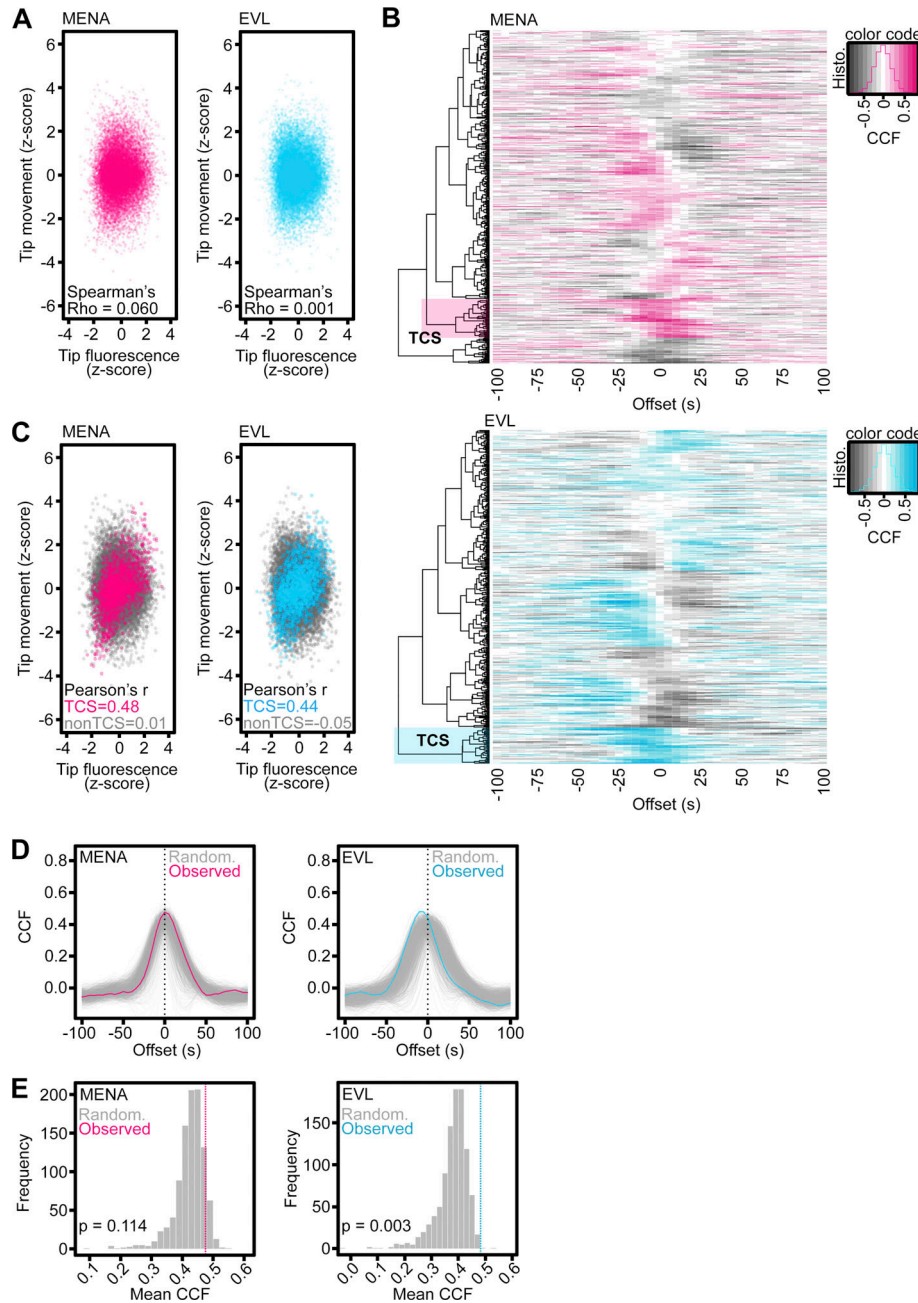

Figure S4. **Tip enrichment of EVL precedes DF protrusion. (A)** Scatterplots of tip fluorescence and tip motility z-scores at individual timepoints across entire population of analyzed DFs from D11 neurons expressing EGFP-MENA or EGFP-EVL. Poor correlation of tip fluorescence and motility observed by Spearman's correlation test across total population of DFs. Individual timepoints presented, $n = 368–389$ total DF from 4–6 neurons per biological replicate, $N = 3$ biological replicates. **(B)** Heatmaps of cross-correlation function (CCF) of normalized tip fluorescence intensity and tip motility per individual DF (rows) as a function of time offset (columns), of DFs from D11 neurons of indicated conditions. Positive cross-correlations indicated in color, negative cross-correlations indicated in gray. High cross-correlation occurring with negative offset values indicates that fluorescence enrichment precedes motility; high cross-correlation with positive offset values indicates fluorescence enrichment follows motility. Color shaded region denotes top-correlating subcluster (TCS) of DFs within each condition. This subpopulation was used to compare metrics for DFs in which fluorescence intensity and motility are highly correlated (TCS), versus DFs in which this correlation is weak or negatively correlated (non-TCS). Histogram (overlaid on color code scale bar) displays DF population distribution across cross-correlation values. **(C)** Scatterplots of tip fluorescence and tip motility z-scores at individual timepoints for top-correlating subcluster (color) overlaid on non-top-correlating subcluster (gray) DFs of indicated conditions. Strong correlation observed by Pearson's correlation test for TCS compared to non-TCS of DFs. Individual timepoints presented, TCS $n = 42$ (MENA) and 39 (EVL); non-TCS $n = 320$ (MENA) and 321 (EVL) total DF from 4–6 neurons per biological replicate, $N = 3$ biological replicates. **(D)** Line plots of cross-correlation values of normalized tip fluorescence intensity and tip motility as a function of time offset for observed top-correlating subcluster (color lines) and the TCS from randomized datasets (gray lines). Randomization was performed by shuffling eight-timepoint blocks of motility data with respect to tip fluorescence per DF (block bootstrap). $N = 1,000$ randomized datasets. **(E)** Frequency histogram displaying distribution of TCS cross-correlation values for randomized subcluster datasets (gray bars) compared to observed TCS cross-correlation values (color dashed line). Frequency of randomized mean peak cross-correlation value exceeding observed mean peak: EGFP-MENA 114/1,000 (bootstrap P = 0.114); EGFP-EVL 3/1,000 (bootstrap P = 003). See also Fig. 3.

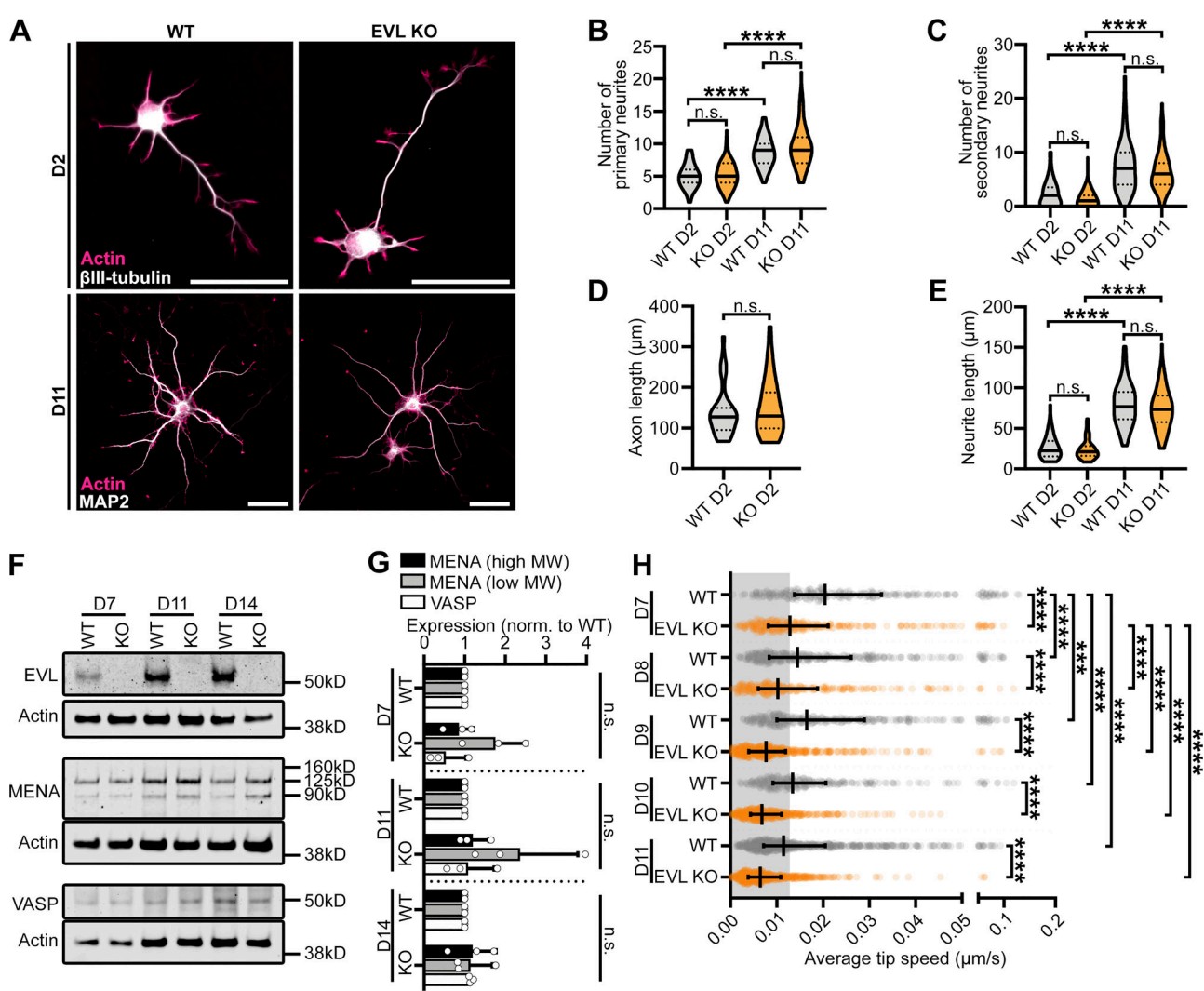

Figure S5. **EVL is required for DF morphogenesis, and influences dendritic spine plasticity. (A)** Immunofluorescence labeling of primary cortical neurons derived from wild-type (WT) or EVL knockout (KO) mice at indicated days in vitro. β-III-tubulin labeling shows overall neurite and early axon morphology, while MAP2 specifically labels dendrites. Scale bar = 50 μm. **(B–E)** Quantification of neurite morphogenesis in cortical neurons derived from wild-type or EVL knockout mice at indicated days in vitro. **(B)** Number of primary neurites originating from the soma. **(C)** Number of higher-order branches (secondary or higher). **(D)** Early axon length. The presumptive axon was defined as a primary neurite with a length greater than three times longer than the minor neurites. **(E)** Average length of primary neurites (excluding presumptive axons). Central line = median, dashed lines = interquartile range (IQR). **(B, C, and E)** Kruskal-Wallis test corrected for multiple comparisons; n = 77–99 total neurons, N = 3 biological replicates, and (D) Mann-Whitney test; n = 32–35 total neurons, N = 3 biological replicates. **(F and G)** Representative Western blot (F) and quantification of (G) protein lysates from cortical neurons derived from wild-type or EVL knockout mice at indicated days in vitro, demonstrating complete loss of EVL, and no compensatory upregulation of VASP or MENA high and low molecular weight isoforms. Mean ± SD. Unpaired t test; N = 3 biological replicates. **(H)**. Scatterplot of average speed of DF tips, calculated as the average absolute tip displacement between successive timepoints. Gray shaded region indicates average speed less than 0.0128 μm/s (non-motile DFs). Median ± IQR. Mixed-effects model; n = 280–641 total DF from 4–6 neurons per biological replicate, N = 3 biological replicates. *P < 0.05, **P < 0.01, ***P < 0.001, ****P < 0.0001, n.s. is not significant. See also Fig. 4, Video 3, and Video 4. Source data are available for this figure: SourceData FS5.

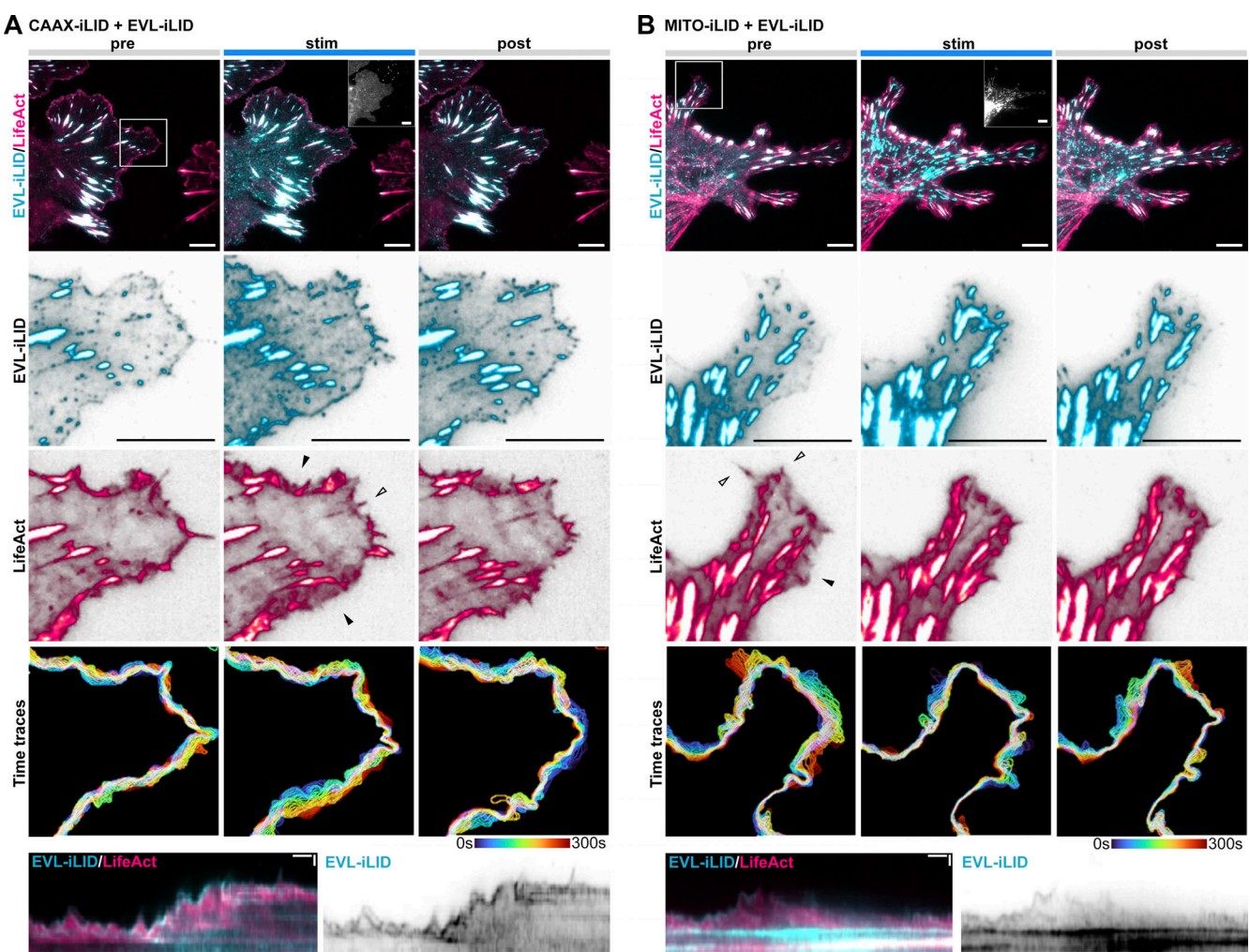

Figure S6. **EVL tip localization is necessary and sufficient for DF motility. (A and B)** Glial cells expressing iRFP670-LifeAct, EVL-iLID, and either CAAX-iLID (A) or MITO-iLID (B). Cells were photostimulated with 488 nm light for 5 min, and imaged for 5 min before, during, and after photostimulation. Row 1: Localization of LifeAct and EVL-iLID during indicated phases of photostimulation. Inset: mVenus-CAAX-iLID or mVenus-MITO-iLID localization. Scale bars = 10 µm. Rows 2–3: Localization of EVL-iLID and LifeAct in magnified region of lamellipodia indicated in Row 1. Individual channels displayed with black subtraction for ease of morphological comparison. Arrowheads indicate regions which exhibited altered dynamics following photostimulation (lamellipodia by empty arrowheads, filopodia by filled arrowheads). Row 4: Maximum intensity projection of temporally color-coded binary mask outline. Row 5: Kymograph of cell's edge. 15 min duration, 5 s interval, vertical scale bar = 1 µm, horizontal scale bar = 1 min, dashed line indicates dendrite. See also Fig. 6 and Video 6.

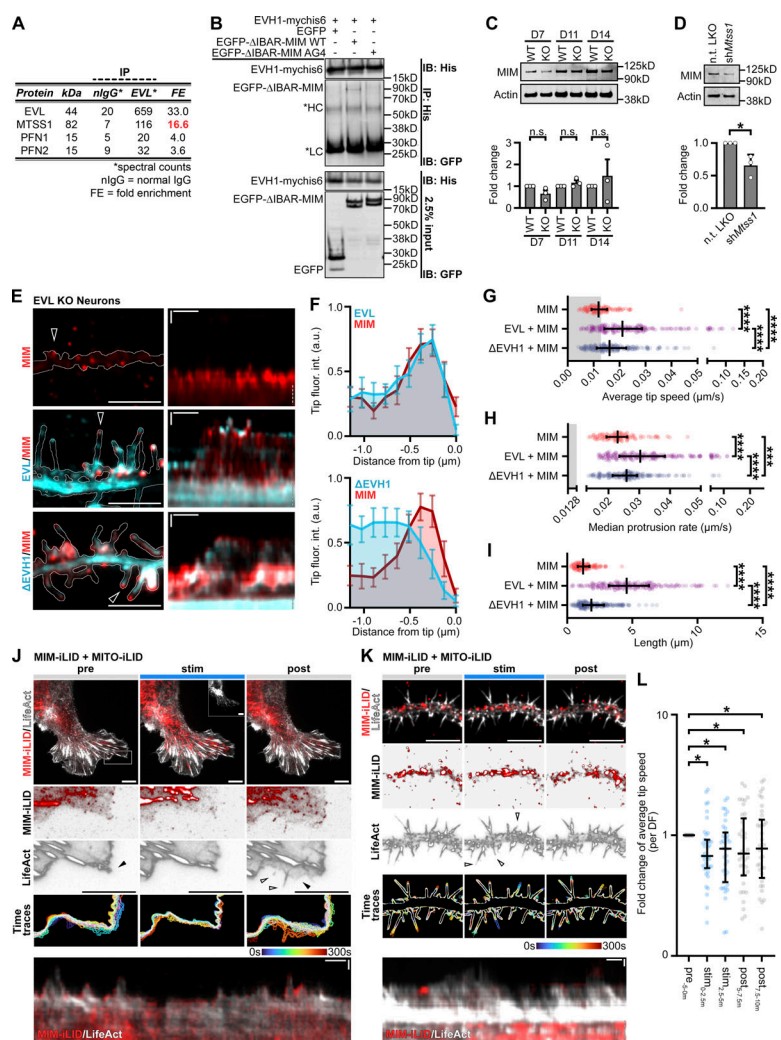

Figure S7. **MIM/MTSS1 cooperates with EVL to promote DF initiation and motility. (A)** Table of select actin-regulating proteins identified by affinity purification-mass spectrometry (AP-MS). **(B)** Representative western blot of co-immunoprecipitation experiments demonstrating requirement of MIM's $_{634}$LPSPP$_{638}$ motif for interaction with EVL's EVH1 domain. HEK293T overexpressing indicated constructs were lysed and immunoprecipitated (IP) with anti-His antibody to pull down EVH1-mychis6, resolved by SDS-PAGE, and immunoblotted (IB) with indicated antibodies. **(C)** Representative western blot and quantification of protein lysates from cortical neurons derived from wild-type or EVL knockout mice at indicated days in vitro, and probed with an antibody targeting MIM. Mean ± SD. Unpaired *t* test; *N* = 3 biological replicates. **(D)** Representative western blot of protein lysates from primary neurons at D11, transduced on D7 with indicated pLKO-shRNA-TurboRFP lentiviral particles targeting *Mtss1* or non-targeting (n.t.) control. Fold change quantification of knockdown compared to non-targeting shRNA control. Mean ± SD. Unpaired *t* test; *N* = 3 biological replicates. **(E)** Segment of dendrite from live EVL knockout cortical neurons at D11 expressing mRuby2-LifeAct, MIM-iRFP670, or MIM together with mEmerald-tagged wild-type EVL or ΔEVH1 as indicated (left). Right: kymograph of DF position indicated by arrowhead in left column. 5 s interval, 5 min duration, dendrite segment: scale bar = 10 µm. Kymograph: Horizontal scale bar = 1 min, vertical scale bar = 1 µm. **(F)** Distribution of MIM and EVL or ΔEVH1 localization along the distal length of DF. Mean ±95% confidence interval, *n* = 25 total DF. **(G)** Scatterplot of average speed of EVL knockout DF tips for indicated expression conditions, calculated as the average absolute tip displacement between successive timepoints. Gray shaded region indicates average speed less than 0.0128 µm/s (non-motile DFs). Median ± interquartile range (IQR). Mixed-effects model; *n* = 101–195 total DF from 2–4 neurons per biological replicate, *N* = 3 biological replicates. **(H)** Scatterplot of median protrusion rates of EVL knockout DFs for indicated expression conditions (the median of values when instantaneous change in length was greater than +0.0128 µm/s [motile, protruding]). Median ± IQR. Mixed-effects model; total DF from 2–4 neurons per biological replicate, *N* = 3 biological replicates. **(I)** Scatterplot of average length of EVL knockout DFs reached during the duration of imaging for indicated expression conditions. Median ± IQR. Mixed-effects model; total DF from 2–4 neurons per biological replicate, *N* = 3 biological replicates. **(J and K)** Glial cells (J) and cortical neurons (K) derived from wild-type mice expressing iRFP670-LifeAct, MIM-iLID, and MITO-iLID. Cells were photostimulated with 488 nm light for 5 min, and imaged for 5 min before, during, and after photostimulation. Row 1: Localization of LifeAct and MIM-iLID during indicated phases of photostimulation. Inset: mVenus-MITO-iLID localization (J only). Rows 2–3: Localization of MIM-iLID and LifeAct in magnified region of lamellipodia indicated in Row 1 (J) or the full dendrite segment (K). Individual channels displayed with black subtraction for ease of morphological comparison. Arrowheads indicate regions which exhibited altered dynamics following photostimulation (lamellipodia = filled arrowheads, filopodia = empty arrowheads). Row 4: Maximum intensity projection of temporally color-coded binary mask outline. Row 5: Kymograph of cell's edge (J) or representative DF position (K). Scale bar = 10 µm. 5 s interval, 15 min duration. **(L)** Scatterplot of fold change of average tip speed, calculated as the average absolute tip displacement between successive timepoints relative to average tip speed pre-photostimulation, for 2.5 min bins during or post-photostimulation, for wild-type neurons expressing MIM-iLID with MITO-iLID. Median ± IQR. Mixed-effects model; *n* = 39 total DF from 2 neurons per biological replicate, *N* = 2 biological replicates. *P <0.05, **P <0.01, ***P <0.001, ****P <0.0001, n.s. is not significant. See also Fig. 6, Video 7, and Video 8. Source data are available for this figure: SourceData FS7.

Video 1.    **Live primary mouse cortical neurons at day in vitro 11 (D11) expressing EGFP-LifeAct and pLKO-shRNA-TurboRFP targeting *Enah*, *Evl*, or non-targeting (n.t.) as indicated.** Left: segment of dendrite with overlay of brightfield and intensity-coded LUT of EGFP-LifeAct. Right: EGFP-LifeAct alone. Scale bar = 10 µm. 5 s interval, 5 min duration, 10 fps. See also Fig. 1 and Fig. S2.

Video 2.    **Live primary mouse cortical neurons at day in vitro 11 (D11) expressing mRuby2-LifeAct and EGFP (top row), EGFP-MENA (middle row), or EGFP-EVL (bottom row).** Left: segment of dendrite with merge of EGFP (cyan) and LifeAct (magenta). Right: rainbow intensity-coded LUT of EGFP with outline of LifeAct-positive borders. Red = highest intensity, purple = lowest intensity. Scale bar = 10 µm. 5 s interval, 5 min duration, 10 fps. See also Fig. 2 and Fig. S3.

Video 3.    **Live primary cortical neurons derived from wild-type (WT) or EVL knockout (KO) mice at indicated days in vitro expressing mRuby2-LifeAct.** Left: segment of dendrite with overlay of brightfield and intensity-coded LUT of mRuby2-LifeAct. Right: mRuby2-LifeAct alone. Scale bar = 10 µm. 5 s interval, 5 min duration, 10 fps. See also Fig. 4 and Fig. S5.

Video 4.    **Live wild-type and EVL knockout (KO) cortical neurons at day in vitro 12 (D12) expressing EGFP-LifeAct and PSD95-FingR-mRuby2.** Row 1: Segment of dendrite with merge of PSD95 reporter (green) and LifeAct (magenta). Row 2: LifeAct fluorescence intensity presented with an inverted LUT to highlight actin dynamics. Row 3: PSD95 reporter fluorescence intensity presented with an inverted LUT. Scale bar = 10 µm. 30 s interval, 50 min duration, 10 fps. See also Fig. 4.

Video 5.    **Live EVL knockout (KO) cortical neurons at day in vitro 11 (D11) expressing mRuby2-LifeAct and mEmerald alone, or mEmerald-tagged wild-type EVL or mutants of EVL as indicated.** Phase 1: Segment of dendrite with merge of mEmerald (cyan) and LifeAct (magenta). Phase 2: Intensity-coded LUT of mRuby2-LifeAct. Phase 3: Rainbow intensity-coded LUT of mEmerald with outline of LifeAct-positive borders. Red = highest intensity, purple = lowest intensity. Scale bar = 10 µm. 5 s interval, 5 min duration, 10 fps. See also Fig. 5.

Video 6.    **Live primary EVL knockout (KO) mouse cortical neurons at day in vitro 11 (D11) expressing iRFP670-LifeAct and indicated iLID constructs.** Neurons were photostimulated with 488 nm laser light for 5 min, and imaged for 5 min before, during, and after photostimulation (stimulation period indicated by blue circle). Row 1: Segment of dendrite with merge of indicated EVL-iLID constructs (cyan) and LifeAct (magenta). Row 2: iLID localization constructs CAAX-iLID or MITO-iLID (images acquired only during stimulation). Row 3: Rainbow intensity-coded LUT of indicated EVL-iLID constructs with outline of LifeAct-positive borders. Red = highest intensity, purple = lowest intensity. Row 4: Intensity-coded LUT of iRFP670-LifeAct. Scale bar = 10 µm. 5 s interval, 15 min duration, 10 fps. See also Fig. 6 and Fig. S6.

Video 7.    **Live primary wild-type mouse cortical neurons at day in vitro 11 (D11) expressing mRuby2-LifeAct, together with mEmerald-EVL and/or MIM-iRFP670 as indicated.** Left: Segment of dendrite with overlay of brightfield and intensity-coded LUT of mRuby2-LifeAct. Right, bottom: Rainbow intensity-coded LUT of indicated constructs with outline of LifeAct-positive borders. Red = highest intensity, purple = lowest intensity. Right (MIM + EVL only): Merge of mEmerald-EVL (cyan) and MIM-iRFP670 (red). Scale bar = 10 µm. 5 s interval, 5 min duration, 10 fps. See also Fig. 7 and Fig. S7.

Video 8.    **Live primary wild-type mouse cortical neurons at day in vitro 11 (D11) expressing EVL-iLID and MIM-iRFP670, and CAAX-iLID or MITO-iLID as indicated.** Neurons were photostimulated with 488 nm laser light for 5 min, and imaged for 5 min before, during, and after photostimulation (stimulation period indicated by blue circle in row 2). Row 1: Segment of dendrite with merge of EVL-iLID (cyan), MIM-iRFP670 (red), and brightfield to better display DF dynamics. Row 2: Merge of EVL-iLID (cyan), MIM-iRFP670 (red) only. Rows 3 and 4: Rainbow intensity-coded LUT of EVL-iLID (row 3) or MIM-iRFP670 (row 4). Red = highest intensity, purple = lowest intensity. Scale bar = 10 µm. 5 s interval, 15 min duration, 10 fps. See also Fig. 7.

Video 9. **Segments of dendrite (left, middle) or dendrite growth cone (right) from live primary cortical neurons derived from wild-type (WT) or EVL knockout (KO) mice at day in vitro 11 (D11) expressing mRuby2-LifeAct.** Cultures were treated with 0.1% DMSO, 100 µM CK-666, or 50 µM Blebbistatin at 5 min, and imaged for 40 min following treatment (treatment period indicated by yellow square). Growth cone dynamics were monitored as a read-out for approximating the time to maximal inhibition of Arp2/3 (∼20 min) and myosin activity (∼15 min) in neurons. Scale bar = 10 µm. 15 s interval, 45 min duration, 10 fps. See also Fig. 8.

**Provided online are Table S1 and Table S2. Table S1 shows plasmids generated and used in this study. Table S2 shows exact P values and statistical methods for all figures.**

