## [Peer Review File · The Journal of Cell Biology]

EVL and MIM/MTSS1 regulate actin cytoskeletal remodeling to promote dendritic filopodia in neurons

Sara Parker, Kenneth Ly, Adam Grant, Jillian Sweetland, Ashley Wang, James Parker, Mackenzie Roman, Kathylynn Saboda, Denise Roe, Megha Padi, Charles Wolgemuth, Paul Langlais, and Ghassan Mouneimne

Corresponding Author(s): Ghassan Mouneimne, University of Arizona

Review Timeline:

Submission Date:	2021-06-15
Editorial Decision:	2021-08-31
Revision Received:	2022-11-22
Editorial Decision:	2023-01-08
Revision Received:	2023-01-17

Monitoring Editor: Louis Reichardt

Scientific Editor: Dan Simon

Transaction Report:

DOI: <https://doi.org/10.1083/jcb.202106081>

August 31, 2021

Re: JCB manuscript #202106081

Dr. Ghassan Mouneimne
University of Arizona
Cellular and Molecular Medicine
1515 N Campbell
Room 0983
Tucson 85724

Dear Dr. Mouneimne,

I am attaching to this letter the evaluations of three reviewers of the manuscript entitled "EVL and MIM/MTSS1 regulate actin cytoskeletal remodeling to promote dendritic filopodia in neurons" that you submitted recently to the Journal of Cell Biology. As will be evident from the reviews, each of the reviewers is quite interested in your results, but also has major scientific concerns that would have to be addressed convincingly in a revision for this journal.

The first reviewer expresses the opinion that you are not studying filopodia relevant for later spine development, but instead filopodia more relevant for understanding their development and function in fibroblasts and perhaps also other non-neuronal cells. I do not agree with the assessment of this reviewer, obviously an expert in neuronal filopodia and spine development, that simple textual changes would suffice. The manuscript needs stronger evidence that the subject of your study is relevant for development of neuronal spines. This could be done by comparing morphological development in your low-density cultures to similar development in higher density cultures with at least some inclusion of time-lapse imaging results.

The second reviewer has two major scientific points that I view as important. First, can you include a better characterization of the domains involved in the MIM-EVL interaction; second, this reviewer has the major concern that data you are interpreting as demonstrating formin dependence have another explanation due to the cross-specificity of SMIFH2 with certain myosins. This reviewer has also several more minor issues that I believe you can clarify in a response and with some modest textual or experimental revisions (items 3, 4, 5, 6). I would join this reviewer in urging you to minimize use of acronyms, restricting their use to a small number.

The third reviewer has some significant statistical recommendations that I believe will enhance your own and readers' confidence in the durability of the differences revealed through your data. Please also clarify the discrepancies identified by this reviewer in Figure 6. Finally, of course, this journal needs some response to the final sentence in this review (related or the same manuscript?).

It was interesting to me that each reviewer identified one or two major concerns that were not identified by the other two. Nonetheless, after reading your manuscript and their evaluations, I think a final positive editorial decision will require experiments and/or statistical revisions that comprehensively address each of them. I think you can do this, but it will require some additional work by you and your lab members. I think your responses must be convincing to each of them. Additionally, it seems important that you demonstrate that the filopodia under study are relevant for neuronal development through some comparative studies. As an editor, I feel it important to emphasize this point.

To summarize, the overall positive views by each of reviewer of the importance and novelty of your work is a strong plus, but the serious scientific concerns must be addressed. I encourage you and your colleagues to submit a revision, but only after you have done this. The revision will almost certainly be returned to each of the reviewers and I am also looking forward to seeing it. Please do not do it prematurely, since the JCB has the policy of only considering a single round of revision.

If you would like to resubmit this work to JCB, please contact the journal office to discuss an appeal of this decision or you may submit an appeal directly through our manuscript submission system. Please note that priority and novelty would be reassessed at resubmission.

Regardless of how you choose to proceed, we hope that the comments below will prove constructive as your work progresses. We would be happy to discuss the reviewer comments further once you've had a chance to consider the points raised in this letter. You can contact the journal office with any questions, cellbio@rockefeller.edu or call (212) 327-8588.

Thank you for thinking of JCB as an appropriate place to publish your work.

Sincerely,

Louis Reichardt, PhD
Monitoring Editor
Journal of Cell Biology

Dan Simon, PhD
Scientific Editor
Journal of Cell Biology

Reviewer #1 (Comments to the Authors (Required)):

This study has nice cell biological approaches but the problem is that it does not study dendritic filopodia, which are precursors of the dendritic spines, but other filopodia structures which are typical for young dissociated neurons cultured in low density. I propose that authors re-write the article. All problems / confusion in discussion reflects to mis-named objects of the study. Nobody knows what exactly are these low-density culture filopodia but I assume that they resemble developmental filopodia structures which are seen appearing from soma during neuron development. The difference between these studied low-density-culture and real dendritic filopodia (high-density-cultures or in vivo) can be seen with filamentous actin. These low-density-culture filopodia resemble fibroblast filopodia and growth cone filopodia, which have clear F-actin bundles (actin staining is clearly visible). In dendritic filopodia, actin is hardly visible and it is still partially unclear whether filaments are branched (Arp2/3) or straight (formins), bundled, uniparallel or contractile or even in periodic actin rings. To summarize, actin organization in dendritic filopodia is not so well organized as we are used to see in fibroblast filopodia. In filopodia studied in this study, actin looks nice and bright, which also means that they are not exactly dendritic filopodia which can be seen in dense neuron cultures or in vivo tissue. The dynamics of studied filopodia and dendritic filopodia are also different. High-density cultured neurons have thin spines (=small spine heads) at DIV11, in this study, there are no spine heads at DIV11. I assume one reason to use low-density-cultures was TIRF-microscopy. TIRF images look nice indeed but TIRF microscopy compromises the used model.

Anyway, I think studying these low-density-culture dendritic filopodia is interesting, especially from the angle that how cell can regulate the formation of different types of protrusions depending on the environment and contacts to other cells. The actin structure of filopodia depends on whether dendrites and forming filopodia are in contact with other neurons. We were trying to produce dendritic spines in iPSC derived neurons. What we managed to produce, were these low-density-culture filopodia. I am not sure what is the physiological relevance of these filopodia, as they might be important mainly in low-density cultures but obviously there might be some function also in vivo. What is very good is that this study does not try to show dendritic spines or anything which would not be normal for low-density-cultures. Figures and experiments look very honest. Thus, I think that by changing the wording, introduction and discussion, this can become a nice article. Main question is, what is the physiological relevance of these filopodia. But even with that possible weakness, study is interesting for actin biologists as a study describing a mechanism of one type filopodium formation. And study uses nice cell biology techniques. Overall, this study looks like cell biologist study done in neurons, which is highly appreciated, however, I am not sure that studied objects are important for neuron functionality.

MIM-EVL interaction is an interesting finding and according to my knowledge, not reported earlier. And proposed MIM-Arp2/3 - EVL model is interesting and possible and maybe the same mechanism is used in high-density-culture dendritic filopodia.

In general, experiments look well done.

Some minor notes, questions:

1. I did some fun experiments with VASP in past and I saw the same star phenotype which is shown here. I was wondering at that time, and I am still wondering: If Ena/VASP (here MENA in S2B) is the main actin polymerizing factor here and it is on the tips of star filopodia, why then F-actin seems to concentrate in the middle of star shape? By quick look, it looks like F-actin is polymerizing in the middle but I don't know if this is true or is F-actin just for example bundled together in the middle resulting in bright staining.
2. I could not find information for used MIM antibody. It is good to show whole Western blots at least in supplementary figures.

Reviewer #2 (Comments to the Authors (Required)):

This manuscript presents convincing evidence that EVL is the primary ENA/VASP family actin polymerase involved in promoting outgrowth of the dendritic filopodia that eventually become dendritic spines. In addition, they describe an interaction between EVL and the I-BAR domain containing protein MIM/MTSS1 that appears to play a role in initiation and outgrowth of the dendritic filopodia. While the evidence that EVL and MIM are important to this process is solid, many of the more mechanistic conclusions

that the authors attempt to draw are not supported by the evidence. The paper will require significant revision before it would be acceptable for publication in the JCB.

Specific concerns:

1. How does MIM/MTSS1 interact with the EVH1 domain of EVL? On p. 5 the authors claim that MIM interacts with the EVH1 domain of EVL but they do not test whether the proline-rich-domain of EVL (which mediates many of its interactions) might also play a role. They also do not discuss how the EVH1 domain might interact with MIM. The canonical binding motif is an FPPPP sequence (or rarely an LPPPP motif), but MIM does not have any such sequence (the closest match is residues 630-634 which are LPAPP). If the authors think that this interaction does not happen via a canonical binding motif this would be big news (e.g. Boëda et al., 2007).

Tes, a specific Mena interacting partner, breaks the rules for EVH1 binding. *Mol Cell*. 28(6):1071-82). The authors should determine whether the proline rich domain contributes to the MIM/EVL interaction and which region of MIM interacts with the EVH1 domain of EVL.

2. On p. 7 the authors state that "I-BAR proteins such as MIM, promote the activation of Arp2/3-dependent actin nucleation. The two papers they cite for this fact, however, do not support the statement. Lin et al. (2005) demonstrate that cortactin and MIM co-localize and that MIM weakly potentiates the ability of cortactin to stimulate the Arp2/3 complex. This stimulation is parabolic, and likely reflects MIM-dependent dimerization of cortactin, an effect known to enhance the activity of Arp2/3 activators (Padrick et al., 2011. *PNAS*. 108(33):E472-9). And Saarikangas et al. (2015) state clearly that their data suggest that Arp2/3 activation in their experiments is stimulated by PIP2, not MIM: "We show that I-BAR protein MIM nucleates spine formation through its membrane deforming activity, which occurs independently and prior to actin assembly. MIM is recruited to the plasma membrane by PIPs that provide a bimodal signal for protrusion formation by activating both the membrane bending activity of MIM and by stimulating actin polymerization." I appreciate that the authors would like to provide a molecular-level explanation for their results, but they should avoid over-interpreting both the literature and their own results.

3. I do not understand the authors interpretation of the data in Figure 7. For example, on p. 8 the authors note that "CK-666-treated EVL KO neurons exhibited comparatively minor increases in DF length and motility [...], demonstrating that EVL is required for the observed motility phenotypes with Arp2/3 inhibition." There is no quantification of these 'comparatively minor increases' and a minor increase would be inconsistent with the claim that EVL is required for the observed effect of Arp2/3 inhibition.

4. Also on p. 8: "CK-666 also reduced F-actin content and flaring in both WT and EVL KO neurons, leaving DF as thin protrusions..." The authors cite Figure 7D as evidence for this statement, but to my eye 7D shows exactly the opposite. CK-666 appears to induce a big drop in spread area of the EVL knockout cells but not the untreated cells. Why does the description not match the figure?

5. Continuing on p. 8 "Collectively, these experiments demonstrate that EVL is the primary AEF responsible for elongation of Arp2/3-dependent actin networks..." That is not the conclusion that I draw from Figure 7. The fact that spread area and total actin drop when EVL is knocked out and the Arp2/3 complex is inhibited but not when Arp2/3 alone is inhibited suggests that the two factors might be operating in parallel and that EVL can partially compensate for the acute loss of Arp2/3 activity.

6. In Figure 7E I am not convinced by the effect of SMIFH2 on wild type cells (right panel). The reduction in protrusion density caused by addition of SMIFH2 is very similar to that caused by addition of DMSO (control). It appears that the only thing that makes the SMIFH2 result 'significant' is the small error bars. In fact the error bars on this post are smaller than any of the others points in the plot. Also, the reduction caused by CK-666 is almost identical to that caused by SMIFH2.

7. Finally there is a big problem with the use of the "formin inhibitor" SMIFH2. The Sellers laboratory recently showed that SMIFH2 turns out to be a more potent inhibitor of myosin-family motor proteins than of formin-family nucleation factors (Nishimura et al., 2021. *J Cell Sci*. 134:jcs253708). Since the velocity and extent of filopodial protrusion are known to be regulated by myosin activity (Mallavarapu and Mitchison, 1999. *J Cell Biol*. 146:1097) interpreting the SMIFH2 data in this paper is impossible. The authors should remove these data from the paper and base any statements about the role of formins in dendritic filopodia formation and growth on specific knockouts or dominant negative constructs.

Minor points and technical issues:

1. The paper is very confusing to read. One of the biggest problems is the proliferation of acronyms. If I put the paper down for a minute and then tried to take up where I left off, I found myself constantly flipping back in the text to figure out what all the acronyms stood for. At the very least the acronyms DF, AEF, CCF, TCS, WT, and KO should ALL be written out everywhere in the main body of the manuscript. It will dramatically improve readability of the paper.

2. On a related note, I never figured out what TPM (in the legend of Figure 1H) means. Transcripts per million? This should be written out in the figure caption at least.

3. On p. 3 and in Figure 1 the authors establish that EVL is the dominant ENA/VASP protein in primary cortical neurons. Most of

the methods rely on counting transcript numbers except for Figure 1F, which is an immunoblot. The authors should make some attempt to quantify the relative amounts of MENA, VASP, and EVL protein in their cell preparations. Even though these preparations are contaminated with glial cells, some quantitative sense of the relative protein levels would be useful.

Reviewer #3 (Comments to the Authors (Required)):

Review of Parker et al

This is an intriguing and well analyzed manuscript on the importance of the Ena/VASP family member and actin polymerase EVL, and the I-BAR protein MIM, in dendritic filopodia formation and elongation. Dendritic filopodia formation and elongation are important because they are the precursors of dendritic spines, which form excitatory synapses in the brain. Although Ena/VASP family proteins (specifically Mena and VASP) are well known to function in filopodia formation in early developing neurons and non-neuronal cells, the function of the third family member EVL (Ena/VASP-like) is more of an enigma. Here the authors conduct a number interesting experiments with both shRNA and the EVL KO mouse to show convincingly that EVL is necessary for dendritic filopodia elongation and motility in maturing (DIV 11) mouse cortical neurons, while Mena is not (VASP was shown to be expressed at low levels in the maturing brain and was not studied in these processes). Additionally, they conduct experiments with iLID constructs to show the EVL is sufficient to elongate dendritic filopodia. However, it is clear EVL is not acting alone. They conduct a mass spec analysis of EVL-associated proteins and find that the plasma membrane deforming I-BAR protein MIM is also an EVL binding partner and required for dendritic filopodia. Because both the actin branching factor Arp2/3 and the actin nucleator/polymerase formin have been shown to be critical for dendritic filopodia they conclude their study showing pharmacological manipulation of these proteins also affects dendritic filopodia formation and motility in the early formation of filopodia, but EVL is essential for dendritic filopodia elongation and motility. Overall, this is a very well written and constructed manuscript with excellent data that implicates a heretofore unknown regulator of actin polymerization in dendritic spines. One important caveat and a few minor comments follow.

- The authors are very quantitative in their assays. However, they consider the "n" for each graph as individual filopodia analyzed, or the like. This results in an "n" of several hundred for most of the graphs presented. With this high of an "n" it is extremely likely that type 1 errors are made (suggesting two comparisons are significantly different, when they are not). These type of errors are well described in a recent JCB publication (SuperPlots: Communicating reproducibility and variability in cell biology. Lord SJ, Velle KB, Mullins RD, Fritz-Laylin LK. J Cell Biol. 2020 Jun 1;219(6):e202001064. doi: 10.1083/jcb.202001064 (see Table S1 specifically). I have no doubt that much of the data the authors are showing here are significant, but using an "n" as the biological replicate (3-5), showing each experimental mean/median (with individual filopodial points shown in the background, i.e. a SuperPlot) and doing a paired t-test or ANOVA would be much more convincing, because some of the differences in the graphs that are shown are quite small, but would be statistically significant if the "n" was in the hundreds. One could argue that even conducting their statistical methodology they did find non-significant results. Yes, but these values are almost exactly the same and detected quite infrequently. One is left with wondering if several of the values shown might be statistically significant, but biologically insignificant. Nevertheless, one could also argue that using such a small n=3-5 would result in a type 2 error (missing a significant difference). Potentially, using dendrites or cells, rather than filopodia as the "n" might make the data more convincing. However, it is unclear what each "n" is in each graph.

- Fig. 1G,H and I could go in the supplemental data, since these are data from adults, whereas this study is focused on maturing (DIV 7-11).

- In Fig. 6N it does not make sense that the protrusion density is higher in the EVL KO than in control and KD of MIM in the KO results WT levels. The images in 6I and 6M, which are examples of the data graphed in 6N, appear very different.

- The related manuscript appears to be the same manuscript.

Response to reviewers:

Reviewer #1:

This study has nice cell biological approaches but the problem is that it does not study dendritic filopodia, which are precursors of the dendritic spines, but other filopodia structures which are typical for young dissociated neurons cultured in low density. I propose that authors re-write the article. All problems / confusion in discussion reflects to mis-named objects of the study. Nobody knows what exactly are these low-density culture filopodia but I assume that they resemble developmental filopodia structures which are seen appearing from soma during neuron development.

We agree that the nature of “dendritic filopodia” is controversial and complex. The actin architecture and dynamics of filopodia-like protrusions emanating from the dendrite indeed change throughout development, and there are likely different subpopulations of filopodia-like protrusions at any given time point that contribute to different functions in the neuron (i.e. filopodia involved in dendrite branching versus synaptogenesis; (Portera-Cailliau *et al* 2003 J Neurosci, Leondaritis & Eickholt 2015 PLOS Biol)). Further, given the diversity of filopodia form and function, the literature is muddled due to differences in experimental conditions between studies, including examination of different days of *in vitro* development, differences in neuronal morphogenesis between model organisms, and variation in culture conditions and methods of actin visualization. These developmental and experimental complexities are in part what attracted us to the problem, and the rationale behind our careful and explicit characterization of these structures in culture.

Importantly, we used low density cultures in our studies to ensure that we readily find clean dendrite segments that are not enveloped in axons, providing ample examples of freely motile dendritic filopodia that are not in contact with other structures. At high culture densities, we find that the incredibly dense meshwork of traversing axons precludes examination of unbound dendritic filopodia and hence, would have prevented us from accomplishing the work we performed in this study.

In this study, the filopodia that we show are EVL-dependent dendritic protrusions are developmentally coincident with synaptogenesis, and distinct from growth cone filopodia, filopodia emanating from the soma, and certain other filopodia emanating from dendrites. Cultures from EVL KO mice retain these latter structures and demonstrate pronounced morphological defects beginning around day *in vitro* 9 (Fig. 4) – a time point, which we and others found that it immediately precedes early synaptogenesis (Basarsky *et al* 1994 J Neurosci). This suggests that the EVL dependent dendritic filopodia-like protrusions that are associated with synaptogenesis, while the other types of filopodia could be mediated by other actin regulators, such as ENAH or formins.

Furthermore, in our revised manuscript, we provide evidence that the dendritic filopodia under study here are physiologically relevant, and capable of forming synapses not only in low ($7.8 \times 10^3/\text{cm}^2$) density, but also in high ($26.0 \times 10^3/\text{cm}^2$)

density culture conditions (SFig. 1A, Fig. 4DF, Video 4). We utilized a lentiviral construct containing fluorescently-tagged PSD95 Fibronectin intrabodies generated with mRNA display (FingR; Gross *et al* 2013 Neuron). This tool, unlike tagged PSD95 overexpression, is non-perturbing, and does not affect synapse density, morphology, or function. Further, it enables observation of synapse formation in the same neuron over time. Our newly added Supplementary Figure 1 provides an expanded validation and characterization of dendritic filopodia to address the concerns and questions brought by Reviewer 1, including a developmental time course of synaptogenesis in wildtype, low density neuron cultures (SFig. 1A). In this panel, synapse formation can be observed forming via dendritic filopodia, and maturing throughout the observation period at days *in vitro* 9-14 in the same neuron. Our revised manuscript also provides validating examples of synaptogenesis occurring via dendritic filopodia during live-cell imaging experiments in high density ($26.0 \times 10^3/\text{cm}^2$) neuron cultures (Fig. 4DF, Video 4) – a cell density chosen to increase the consistency of synapse density for quantification purposes. Together, these points and additional perturbation experiments (Fig. 4) demonstrate the physiological relevance of our study, and that the dendritic filopodia examined throughout our study are indeed *bona fide* synaptic precursors.

The difference between these studied low-density-culture and real dendritic filopodia (high-density-cultures or in vivo) can be seen with filamentous actin. These low-density-culture filopodia resemble fibroblast filopodia and growth cone filopodia, which have clear F-actin bundles (actin staining is clearly visible). In dendritic filopodia, actin is hardly visible and it is still partially unclear whether filaments are branched (Arp2/3) or straight (formins), bundled, uniparallel or contractile or even in periodic actin rings. To summarize, actin organization in dendritic filopodia is not so well organized as we are used to see in fibroblast filopodia. In filopodia studied in this study, actin looks nice and bright, which also means that they are not exactly dendritic filopodia which can be seen in dense neuron cultures or in vivo tissue.

To date, previous work strongly supports a model of dendritic filopodia architecture features Arp2/3-mediated branched actin networks and base-directed myosin-II-dependent contractility of mixed-polarity actin filaments, with mostly tip-directed polymerization (Chazeau *et al* 2014 EMBO, Hotulainen *et al* 2009 JCB, Korobova & Svitkina 2010 MBoC, Tatavarty *et al* 2012 MBoC. Therefore, the architecture of the actin cytoskeleton in dendritic filopodia is structurally more reminiscent of lamellipodia than conventional filopodia, such as those found in growth cones and fibroblasts, which have F-actin typically bundled with fascin. Our work advances this model by identifying and characterizing a unique actin elongation factor that is necessary and sufficient for dendritic filopodia morphogenesis and their defining dynamics. Certainly, we are not claiming that the structure of the actin cytoskeleton in dendritic filopodia is uniformly made of one architecture; and we acknowledge, as the reviewer stated, that the actin cytoskeleton in dendritic filopodia is probably not as uniformly structured as conventional filopodia.

To demonstrate that the molecular architecture of the dendritic filopodia in our study is in agreement with the currently accepted model, we included in our revised manuscript characterization of conventional and dendritic filopodial markers in low-density DIV11 cultures (SFig. 1). We show that although growth cone and soma filopodia are strongly enriched with fascin (and are thus categorized as “conventional filopodia”), the dendritic filopodia-like protrusions are fascin-negative (SFig. 1C). Further, Arp2/3 is localized at the tip and along the length of the dendritic filopodium (SFig. 1D). Lastly, we used an overexpression strategy using fluorescently-tagged Arp3 and myosin-II regulatory light chain (MRLC) to demarcate branched and contractile antiparallel actin architectures in the dendritic filopodium, respectively. We found Arp3 throughout the dendritic filopodia and enriched dynamically at the tip (SFig. 1E, Fig. 8E); while MRLC is predominantly found near the base and exhibits rearward flow (SFig. 1E, Fig. 8F). We hope these newly provided data will assuage concerns about the identity of the dendritic filopodia under study.

Regarding the reviewer’s comment about the intensity of F-actin, we and others found that dendritic filopodia labeled with phalloidin in fixed cultures, or with LifeAct or fluorescently tagged-actin in live neurons, are strongly positive for F-actin (Hotulainen *et al* 2009 JCB, Saarikangas *et al* 2015 Dev Cell, Tatavarty *et al* 2012 MBoC). Additionally, other works demonstrate that dendritic filopodia are actin-rich structures by cryoEM, as well as STED microscopy in living brains (Fiala *et al* 1998 J Neurosci, Willig *et al* 2014 Biophys J, Wegner *et al* 2017 Sci Rep). It is possible that due to the small volume of dendritic filopodia, which would concentrate fluorophores and give a very strong fluorescence intensity that although reflective of the actin biology in the filopodia, might exaggerate the intensity of the actin therein. Additionally, the intensity of LifeAct at dendritic filopodia in our studies may appear stronger relative to the parent dendrite due to our use of TIRF microscopy. We understand that bundled actin in conventional filopodia might appear brighter than branched actin; nonetheless, there are multiple reasons for potential differences in actin intensity not related to the actin architecture/biology.

The dynamics of studied filopodia and dendritic filopodia are also different. High-density cultured neurons have thin spines (=small spine heads) at DIV11, in this study, there are no spine heads at DIV11. I assume one reason to use low-density-cultures was TIRF-microscopy. TIRF images look nice indeed but TIRF microscopy compromises the used model. Anyway, I think studying these low-density-culture dendritic filopodia is interesting, especially from the angle that how cell can regulate the formation of different types of protrusions depending on the environment and contacts to other cells. The actin structure of filopodia depends on whether dendrites and forming filopodia are in contact with other neurons.

We have attempted to navigate the complexities of cultured neurons by consistently examining dendritic filopodia in low density cultures at day 11. In our hands, these low density cultures can indeed form synapses both before and immediately following day 11, and was purposefully chosen as the latest developmental day we could examine before rampant synaptogenesis. As described above, low density

cultures allow us to confidently examine the dynamics of unbound dendritic filopodia, those not in contact with an axon. Immature dendritic spines with a filopodia-like morphology (often conflated with dendritic filopodia) exhibit reduced dynamics and altered actin architecture, as they are bound to a synaptic partner (Tatavarty *et al* 2012 MBoC). Engagement of transsynaptic cell adhesion molecules initiates the molecular clutch of spine morphogenesis, in which protrusive force and myosin contractility drives actin reorganization and the expansion of the spine head (Chazeau *et al* 2015 MBoC, Kastian *et al* 2021 Cell Reports).

To preserve filopodial dynamics in living neuronal cultures, TIRF microscopy was chosen as the fastest and least damaging imaging method available to us. We do acknowledge that the drawbacks of TIRF include the inaccessibility of distal DF relative to the TIRF imaging plane, resulting in missing structures that may have been otherwise captured using a different mode of microscopy, such as light sheet or spinning disk confocal. We tested both of these imaging technologies and found that they did not preserve neuronal health as well as TIRF, and exhibited pronounced photobleaching which would preclude us from investigating dynamics with sufficient temporal resolution or duration. Thus, to maintain the integrity of our dynamic study, we chose TIRF as our imaging methodology.

We were trying to produce dendritic spines in iPSC derived neurons. What we managed to produce, were these low-density-culture filopodia. I am not sure what is the physiological relevance of these filopodia, as they might be important mainly in low-density cultures but obviously there might be some function also in vivo. What is very good is that this study does not try to show dendritic spines or anything which would not be normal for low-density-cultures. Figures and experiments look very honest. Thus, I think that by changing the wording, introduction and discussion, this can become a nice article. Main question is, what is the physiological relevance of these filopodia. But even with that possible weakness, study is interesting for actin biologists as a study describing a mechanism of one type filopodium formation. And study uses nice cell biology techniques. Overall, this study looks like cell biologist study done in neurons, which is highly appreciated, however, I am not sure that studied objects are important for neuron functionality.

We thank the reviewer for their appreciation of the significance of our work. We believe that the experiments included in our revised manuscript show the synapse-forming capabilities of the dendritic filopodia under study, and clarify the appropriateness of our in vitro model.

MIM-EVL interaction is an interesting finding and according to my knowledge, not reported earlier. And proposed MIM-Arp2/3 -EVL model is interesting and possible and maybe the same mechanism is used in high-density-culture dendritic filopodia.

In general, experiments look well done.

Some minor notes, questions:

1. I did some fun experiments with VASP in past and I saw the same star phenotype which is shown here. I was wondering at that time, and I am still wondering: If Ena/VASP (here MENA in S2B) is the main actin polymerizing factor here and it is on the tips of star filopodia, why then F-actin seems to concentrate in the middle of star shape? By quick look, it looks like F-actin is polymerizing in the middle but I don't know if this is true or is F-actin just for example bundled together in the middle resulting in bright staining.

We also found it very interesting that overexpression of MENA promotes the generation of filopodia with very different morphology and dynamics compared to EVL overexpression, as well as compared to wildtype neurons. We felt the phenotypes detailed in SFig.2 (now SFig.3) demonstrate that MENA and EVL are not interchangeable and carry out different functions in neurons. We have preliminary data with co-overexpression of mRuby2-EVL and GFP-MENA in neurons that indicates that EVL and MENA can occasionally be found together at dendritic protrusions but are often localized to distinct filopodial structures. These data were not included in our manuscript, as Ena/VASP proteins are known to heterotetramerize (Riquelme *et al* 2015 Biosci Rep), which may confound interpretation of these results.

2. I could not find information for used MIM antibody. It is good to show whole Western blots at least in supplementary figures.

The materials and methods have been revised to include the information on the MIM antibody source, and we apologize for that oversight. We have included all uncropped Western blots in the revised submission (and will deposit them on publicly accessible Mendeley data).

Reviewer #2:

This manuscript presents convincing evidence that EVL is the primary ENA/VASP family actin polymerase involved in promoting outgrowth of the dendritic filopodia that eventually become dendritic spines. In addition, they describe an interaction between EVL and the I-BAR domain containing protein MIM/MTSS1 that appears to play a role in initiation and outgrowth of the dendritic filopodia. While the evidence that EVL and MIM are important to this process is solid, many of the more mechanistic conclusions that the authors attempt to draw are not supported by the evidence. The paper will require significant revision before it would be acceptable for publication in the JCB.

We appreciate the reviewer's thoughtful assessment of our work. We believe that we addressed the reviewer's concerns in our revised manuscript, with additional mechanistic interrogation of the EVL-MIM interaction and re-thinking of our pharmacological inhibition experiments.

Specific concerns:

1. How does MIM/MTSS1 interact with the EVH1 domain of EVL? On p. 5 the authors claim that MIM interacts with the EVH1 domain of EVL but they do not test whether the proline-rich-domain of EVL (which mediates many of its interactions) might also play a role. They also do not discuss how the EVH1 domain might interact with MIM. The canonical binding motif is an FPPPP sequence (or rarely an LPPPP motif), but MIM does not have any such sequence (the closest match is residues 630-634 which are LPAPP). If the authors think that this interaction does not happen via a canonical binding motif this would be big news (e.g. Boëda et al., 2007. Tes, a specific Mena interacting partner, breaks the rules for EVH1 binding. Mol Cell. 28(6):1071-82). The authors should determine whether the proline rich domain contributes to the MIM/EVL interaction and which region of MIM interacts with the EVH1 domain of EVL.

In our original submission, we presented co-immunoprecipitation experiments of full-length MIM with full-length EVL and EVH1-deletion EVL (Fig. 6B, now Fig. 7B), demonstrating that EVL's EVH1 domain is required for MIM interaction. We had also performed co-immunoprecipitation experiments with a proline-rich region deletion of EVL (previously not shown) - the proline-rich deletion mutant co-immunoprecipitates strongly with MIM, in fact, with higher observed affinity than full-length EVL. We did not include these results in our first submission because we were unable to further characterize the proline-rich deletion mutant, as it would not express in neurons (despite strong expression in HEK293T). This may be due to an unknown regulatory mechanism that targets this mutant protein for degradation, however these questions are beyond the scope of our study. We have revised the figure to include this immunoprecipitation data, and postulate that native binding partners to EVL's proline-rich region – such as SH3 domain-containing proteins, could render EVL sterically or spatially unavailable to interact with MIM via its EVH1 domain.

We also identified the LPSPP motif at amino acids 634-638 in MIM earlier in our investigation, while querying our mass spectroscopy hit for putative EVH1-interacting partners. We have since performed co-immunoprecipitation experiments between the C-terminus of MIM containing its proline-rich region and EVL's EVH1 domain. Mutation of LPSPP to AGGGG abrogated EVH1 interaction, suggesting that this sequence is critical for EVL-MIM binding. These new experiments have been included in the revised manuscript (SFig.7B)

2. On p. 7 the authors state that "I-BAR proteins such as MIM, promote the activation of Arp2/3-dependent actin nucleation. The two papers they cite for this fact, however, do not support the statement. Lin et al. (2005) demonstrate that cortactin and MIM co-localize and that MIM weakly potentiates the ability of cortactin to stimulate the Arp2/3 complex. This stimulation is parabolic, and likely reflects MIM-dependent dimerization of cortactin, an effect known to enhance the activity of Arp2/3 activators (Padrick et al., 2011. PNAS. 108(33):E472-9). And Saarikangas et al. (2015) state clearly that their data suggest that Arp2/3 activation in their experiments is stimulated by PIP2, not MIM:

"We show that I-BAR protein MIM nucleates spine formation through its membrane deforming activity, which occurs independently and prior to actin assembly. MIM is recruited to the plasma membrane by PIPs that provide a bimodal signal for protrusion formation by activating both the membrane bending activity of MIM and by stimulating actin polymerization." I appreciate that the authors would like to provide a molecular-level explanation for their results, but they should avoid over-interpreting both the literature and their own results.

We apologize for our oversight and overinterpretation in our reference to these manuscripts. Our intention was to connect MIM's role in membrane deformation and PIP₂ enrichment to the resulting increased concentration of nucleation-promoting factors that promote protrusion. With regards to this matter, the text has been corrected throughout in the revised manuscript.

3. I do not understand the authors interpretation of the data in Figure 7. For example, on p. 8 the authors note that "CK-666-treated EVL KO neurons exhibited comparatively minor increases in DF length and motility [...], demonstrating that EVL is required for the observed motility phenotypes with Arp2/3 inhibition." There is no quantification of these 'comparatively minor increases' and a minor increase would be inconsistent with the claim that EVL is required for the observed effect of Arp2/3 inhibition.

4. Also on p. 8: "CK-666 also reduced F-actin content and flaring in both WT and EVL KO neurons, leaving DF as thin protrusions..." The authors cite Figure 7D as evidence for this statement, but to my eye 7D shows exactly the opposite. CK-666 appears to induce a big drop in spread area of the EVL knockout cells but not the untreated cells. Why does the description not match the figure?

5. Continuing on p. 8 "Collectively, these experiments demonstrate that EVL is the primary AEF responsible for elongation of Arp2/3-dependent actin networks..." That is not the conclusion that I draw from Figure 7. The fact that spread area and total actin drop when EVL is knocked out and the Arp2/3 complex is inhibited but not when Arp2/3 alone is inhibited suggests that the two factors might be operating in parallel and that EVL can partially compensate for the acute loss of Arp2/3 activity.

The pharmacological inhibition experiments were quantified by examining fold change in cell area (formerly Fig. 7D) and providing examples of individual filopodia behavior during the course of treatment (formerly Fig. 7C). Fold change in area was used as a collective measure of changes in tip motility, elongation, flaring, and the maintenance of nascent protrusions. We see now that this approach lacked the granularity needed to draw specific conclusions on protrusive motility, which is a central rationale for these experiments. We feel that the above reviewer's comment, and several that follow, are addressed by re-analyzing and expanding our imaging experiments using drug treatments to include our conventional approach of tip tracking (Fig. 8 in revised manuscript). In our new analyses, we present data that demonstrate a significant increase in DF fold-change tip speed in both WT and EVL KO neurons following treatment with the Arp2/3 nucleation inhibitor CK-666 compared to pre-treatment speeds. Additionally, the length of DF is significantly increased following CK-666 treatment compared to pre-treatment lengths. In the

revised Discussion, we posit that the observed elongation and enhanced motility observed in when Arp2/3 is suppressed is due to a shift towards linear actin polymerization. In EVL KO cells, we posit that other linear polymerization factors are compensating for the lack of EVL, such as formins and MENA.

6. In Figure 7E I am not convinced by the effect of SMIFH2 on wild type cells (right panel). The reduction in protrusion density caused by addition of SMIFH2 is very similar to that caused by addition of DMSO (control). It appears that the only thing that makes the SMIFH2 result 'significant' is the small error bars. In fact the error bars on this post are smaller than any of the others points in the plot. Also, the reduction caused by CK-666 is almost identical to that caused by SMIFH2.

We thank the reviewer for their thoughtful observations, as well as their later critique in the following point of SMIFH2's usage as a pan-formin inhibitor. Given the recent findings from the Sellers lab showing SMIFH2's off-target effects on the myosin superfamily, we have questioned whether our data with this inhibitor is due to targeting myosin-II activity. We shifted our focus on myosin-II instead of formins (discussed further in the next point). As pointed out by the reviewer, myosins (including myosin-II) have been shown to be important for dendritic filopodia and spine morphogenesis (Ryu *et al* 2006 Neuron, Korobova & Svitkina 2010 MBoC). Treatment of mature (day 17) cultured neurons with blebbistatin resulted in the dissolution of dendritic spines into elongated filopodial protrusions with enhanced motility and increased protrusion density (Ryu *et al* 2006 Neuron) – similar elongation phenotypes were observed in dendritic filopodia with blebbistatin treatment in immature (day 8) cultured neurons (Marchenko *et al* 2017 MBoC). Therefore myosin-II could be a potential partner of EVL in promoting these protrusions. In our new experiments, inhibition of myosin using blebbistatin shows an increase in elongation of DF in WT neurons. Intriguingly, we observed that EVL KO neurons exhibit no increase in length following treatment with blebbistatin. Collectively, as we discuss in the revised manuscript, these new data support a model in which, under the suppression of contractility, when actin elongation is expected to be the main driver of protrusion, EVL is required for promoting DF.

*7. Finally there is a big problem with the use of the "formin inhibitor" SMIFH2. The Sellers laboratory recently showed that SMIFH2 turns out to be a more potent inhibitor of myosin-family motor proteins than of formin-family nucleation factors (Nishimura *et al.*, 2021. J Cell Sci. 134:jcs253708). Since the velocity and extent of filopodial protrusion are known to be regulated by myosin activity (Mallavarapu and Mitchison, 1999. J Cell Biol. 146:1097) interpreting the SMIFH2 data in this paper is impossible. The authors should remove these data from the paper and base any statements about the role of formins in dendritic filopodia formation and growth on specific knockouts or dominant negative constructs.*

We thank the reviewer for bringing this recent report to our attention - this is indeed an important caveat to the use of SMIFH2, and any claims of formin specificity. We decided to omit all experiments and interpretations relying upon SMIFH2-derived

results from the final manuscript. To investigate individual formins in the regulation of DF, we identified, in our unpublished work, the following formins to be significantly expressed during developmental periods spanning synaptogenesis: FMN1, FMN2, DAAM1, DAMM2, FHDC1, and FMNL1. We have attempted to manipulate these individual paralogs using the shRNA and acute optogenetic inhibition approach as we employed for EVL; however, at this point, we have decided to exclude the investigation of formins from this manuscript at the JCB due to the magnitude of the work we feel is required to adequately interrogate the paralog-specific roles of individual formins. Importantly, we do not believe that omitting the investigation of formins in the context of this study alters any of our conclusions. We have included a discussion of the potential involvement of formin in DF and we will follow up on these experiments in a future manuscript.

Minor points and technical issues:

- 1. The paper is very confusing to read. One of the biggest problems is the proliferation of acronyms. If I put the paper down for a minute and then tried to take up where I left off, I found myself constantly flipping back in the text to figure out what all the acronyms stood for. At the very least the acronyms DF, AEF, CCF, TCS, WT, and KO should ALL be written out everywhere in the main body of the manuscript. It will dramatically improve readability of the paper.*
- 2. On a related note, I never figured out what TPM (in the legend of Figure 1H) means. Transcripts per million? This should be written out in the figure caption at least.*

We apologize that our overuse of acronyms reduced the accessibility of the writing. This has been corrected in the revised manuscript, with the exception of DF (dendritic filopodia), since it is repeated very often that it disrupts the flow of the text to always spell it out.

- 3. On p. 3 and in Figure 1 the authors establish that EVL is the dominant ENA/VASP protein in primary cortical neurons. Most of the methods rely on counting transcript numbers except for Figure 1F, which is an immunoblot. The authors should make some attempt to quantify the relative amounts of MENA, VASP, and EVL protein in their cell preparations. Even though these preparations are contaminated with glial cells, some quantitative sense of the relative protein levels would be useful.*

In our hands, expression of purified protein for EVL is a challenge. Comparing the relative expression of the proteins would require an onerous amount of work: all three proteins would need to be purified and used as standards against which the endogenous protein levels are measured. We believe that such analysis is beyond the scope of this work and that it will not add to our story since we are not associating such roles with EVL relative expression.

Reviewer #3:

Review of Parker et al

This is an intriguing and well analyzed manuscript on the importance of the Ena/VASP family member and actin polymerase EVL, and the I-BAR protein MIM, in dendritic filopodia formation and elongation. Dendritic filopodia formation and elongation are important because they are the precursors of dendritic spines, which form excitatory synapses in the brain. Although Ena/VASP family proteins (specifically Mena and VASP) are well known to function in filopodia formation in early developing neurons and non-neuronal cells, the function of the third family member EVL (Ena/VASP-like) is more of an enigma. Here the authors conduct a number interesting experiments with both shRNA and the EVL KO mouse to show convincingly that EVL is necessary for dendritic filopodia elongation and motility in maturing (DIV 11) mouse cortical neurons, while Mena is not (VASP was shown to be expressed at low levels in the maturing brain and was not studied in these processes). Additionally, they conduct experiments with iLID constructs to show the EVL is sufficient to elongate dendritic filopodia. However, it is clear EVL is not acting alone. They conduct a mass spec analysis of EVL-associated proteins and find that the plasma membrane deforming I-BAR protein MIM is also an EVL binding partner and required for dendritic filopodia. Because both the actin branching factor Arp2/3 and the actin nucleator/polymerase formin have been shown to be critical for dendritic filopodia they conclude their study showing pharmacological manipulation of these proteins also affects dendritic filopodia formation and motility in the early formation of filopodia, but EVL is essential for dendritic filopodia elongation and motility.

We thank the reviewer for their thoughtful comments, and appreciate their enthusiasm for the findings.

Overall, this is a very well written and constructed manuscript with excellent data that implicates a heretofore unknown regulator of actin polymerization in dendritic spines. One important caveat and a few minor comments follow.

- The authors are very quantitative in their assays. However, they consider the "n" for each graph as individual filopodia analyzed, or the like. This results in an "n" of several hundred for most of the graphs presented. With this high of an "n" it is extremely likely that type 1 errors are made (suggesting two comparisons are significantly different, when they are not). These type of errors are well described in a recent JCB publication (SuperPlots: Communicating reproducibility and variability in cell biology. Lord SJ, Velle KB, Mullins RD, Fritz-Laylin LK. J Cell Biol. 2020 Jun 1;219(6):e202001064. doi: 10.1083/jcb.202001064 (see Table S1 specifically). I have no doubt that much of the data the authors are showing here are significant, but using an "n" as the biological replicate (3-5), showing each experimental mean/median (with individual filopodial points shown in the background, i.e. a SuperPlot) and doing a paired t-test or ANOVA would be much more convincing, because some of the differences in the graphs that

are shown are quite small, but would be statistically significant if the "n" was in the hundreds. One could argue that even conducting their statistical methodology they did find non-significant results. Yes, but these values are almost exactly the same and detected quite infrequently. One is left with wondering if several of the values shown might be statistically significant, but biologically insignificant. Nevertheless, one could also argue that using such a small n=3-5 would result in a type 2 error (missing a significant difference). Potentially, using dendrites or cells, rather than filopodia as the "n" might make the data more convincing. However, it is unclear what each "n" is in each graph.

In this manuscript, we attempt to draw statistical conclusions regarding the changing distribution in behavior of a large population of dendritic filopodia. Many dendritic filopodia are non-motile – whether this is due to adhesion of some filopodia to the substrate, or a biologically-relevant lack of dynamics during the course of imaging, cannot be conclusively demonstrated. We chose to represent our data, and perform statistical analysis, on the total individual datapoints from all biological replicates, rather than the median values on a per cell basis, because we were concerned that biologically-relevant differences in the motile subset of filopodia would be obscured by the “weight” of non-motile filopodia

We have re-done all the statistical analyses in the study in collaboration with the Biostatistics Shared Resources at UACC. As a general approach, we used a linear mixed effects model to appropriately account for the potential correlations among the measurements within a cell. The linear mixed effects model is an extension of an ANOVA model that allows estimation of effect sizes. The new analyses did indeed change a few of our original p values. However, our main conclusions did not change. We report all the p values and the methods used for each analysis in the supplementary data.

Importantly, the type I error is controlled using the linear mixed effects model since it appropriately accounts for the correlation among the measurements (they are not treated as independent observations). The type II error is controlled because all the measurements were appropriately used in the analysis, rather than summary values at the cell level.

- Fig. 1G,H and I could go in the supplemental data, since these are data from adults, whereas this study is focused on maturing (DIV 7-11).

This is an important point, due to the relevance of developmental stage on expression. We have moved the expression analysis of adult mouse to the supplement, to avoid any confusion or misrepresentation.

- In Fig. 6N it does not make sense that the protrusion density is higher in the EVL KO than in control and KD of MIM in the KO results WT levels. The images in 6I and 6M, which are examples of the data graphed in 6N, appear very different.

We thank the reviewer for bringing this to our attention. The quantification of protrusion density includes not only full-fledged filopodia, but also proto-protrusions and proturbences along the dendrite. Factoring these features in, EVL KO neurons have more overall protrusions compared to WT neurons, however develop few protrusions that morphologically resemble DF. We believe the reason for this is two-fold: (1) EVL serves to coalesce and elongate proto-protrusions, and when present, actin architectures are tilted towards productive filopodia and (2) EVL plays a role in cortical actin along the length of the dendrite, and in its absence, small lamellipodia are more abundant. EVL has been demonstrated to play a significant role in fortifying cortical actin in non-neuronal cell types (Padilla-Rodriguez *et al* 2018 Nat Comm, Yu-Kemp *et al* 2017 JCB). Additionally, we have observed that neurons derived from EVL KO mice tend to have dendrites with a narrow diameter, which we hypothesize may be due to EVL's role in the regulation of cortical actin. We are pursuing these questions separately in a different manuscript.

- *The related manuscript appears to be the same manuscript.*

During submission, there were instructions to include any other iterations of the manuscript as a related manuscript. We therefore included the pre-print version that is available on Biorxiv as the related manuscript. We apologize for the confusion that this has caused the reviewers or editorial team.

January 8, 2023

RE: JCB Manuscript #202106081R-A

Dr. Ghassan Mouneimne
University of Arizona
Cellular and Molecular Medicine
1515 N Campbell
Room 0983
Tucson 85724

Dear Dr. Mouneimne,

Thank you for submitting your revised manuscript entitled "EVL and MIM/MTSS1 regulate actin cytoskeletal remodeling to promote dendritic filopodia in neurons". Thank you also for your patience with the review process. We would be happy to publish your paper in JCB pending final revisions necessary to meet our formatting guidelines (see details below). Please also consider the points raised by Reviewer #3 and make any edits that may be necessary.

A. MANUSCRIPT ORGANIZATION AND FORMATTING:

1) Text limits: Character count for Articles is < 40,000, not including spaces. Count includes title page, abstract, introduction, results, discussion, and acknowledgments. Count does not include materials and methods, figure legends, references, tables, or supplemental legends.

2) Figure formatting: Articles may have up to 10 main text figures. Scale bars must be present on all microscopy images, including inset magnifications. Molecular weight or nucleic acid size markers must be included on all gel electrophoresis. Please add scale bars to Figures 6B, 7C, S1 D'/E', and the insets in 1A, 1G, 4D, 7I/M, & S7J. Please also add MW markers in Figures 7A/B, S7B and the actin blots in 1F, S2G/H/I, S3A, S5F, S7C/D.

Please avoid pairing red and green for images and graphs to ensure legibility for color-blind readers. If red and green are paired for images, please ensure that the particular red and green hues used in micrographs are distinctive with any of the colorblind types. If not, please modify colors accordingly or provide separate images of the individual channels.

3) Statistical analysis: Error bars on graphic representations of numerical data must be clearly described in the figure legend. The number of independent data points (n) represented in a graph must be indicated in the legend. Please, indicate whether 'n' refers to technical or biological replicates (i.e. number of analyzed cells, samples or animals, number of independent experiments). If independent experiments with multiple biological replicates have been performed, we recommend using distribution-reproducibility SuperPlots (please see Lord et al., JCB 2020) to better display the distribution of the entire dataset, and report statistics (such as means, error bars, and P values) that address the reproducibility of the findings.

Statistical methods should be explained in full in the materials and methods. For figures presenting pooled data the statistical measure should be defined in the figure legends. Please also be sure to indicate the statistical tests used in each of your experiments (both in the figure legend itself and in a separate methods section) as well as the parameters of the test (for example, if you ran a t-test, please indicate if it was one- or two-sided, etc.). Also, if you used parametric tests, please indicate if the data distribution was tested for normality (and if so, how). If not, you must state something to the effect that "Data distribution was assumed to be normal but this was not formally tested."

4) Materials and methods: Should be comprehensive and not simply reference a previous publication for details on how an experiment was performed. Please provide full descriptions (at least in brief) in the text for readers who may not have access to referenced manuscripts. The text should not refer to methods "...as previously described." Please also indicate the acquisition and quantification methods for immunoblotting/western blots.

5) For all cell lines, vectors, constructs/cDNAs, etc. - all genetic material: please include database / vendor ID (e.g., Addgene, ATCC, etc.) or if unavailable, please briefly describe their basic genetic features, even if described in other published work or gifted to you by other investigators (and provide references where appropriate). Please be sure to provide the sequences for all of your oligos: primers, si/shRNA, RNAi, gRNAs, etc. in the materials and methods. You must also indicate in the methods the

source, species, and catalog numbers/vendor identifiers (where appropriate) for all of your antibodies, including secondary. If antibodies are not commercial, please add a reference citation if possible.

6) Microscope image acquisition: The following information must be provided about the acquisition and processing of images:

- Make and model of microscope
- Type, magnification, and numerical aperture of the objective lenses
- Temperature
- Imaging medium
- Fluorochromes
- Camera make and model
- Acquisition software
- Any software used for image processing subsequent to data acquisition. Please include details and types of operations involved (e.g., type of deconvolution, 3D reconstitutions, surface or volume rendering, gamma adjustments, etc.).

7) References: There is no limit to the number of references cited in a manuscript. References should be cited parenthetically in the text by author and year of publication. Abbreviate the names of journals according to PubMed. JCB formatting does not allow for supplementary references, please remove these from the Table 1 file and add any non-duplicate references to the main reference list.

8) Supplemental materials: Articles generally may have up to 5 supplemental figures and 10 videos. You currently exceed this limit but, in this case, we will be able to give you the extra space. Please also note that tables, like figures, should be provided as individual, editable files. A summary of all supplemental material should appear at the end of the Materials and methods section. Please include one brief sentence per item.

9) Video legends: Should describe what is being shown, the cell type or tissue being viewed (including relevant cell treatments, concentration and duration, or transfection), the imaging method (e.g., time-lapse epifluorescence microscopy), what each color represents, how often frames were collected, the frames/second display rate, and the number of any figure that has related video stills or images.

10) eTOC summary: A ~40-50 word summary that describes the context and significance of the findings for a general readership should be included on the title page. The statement should be written in the present tense and refer to the work in the third person. It should begin with "First author name(s) et al..." to match our preferred style.

11) Conflict of interest statement: JCB requires inclusion of a statement in the acknowledgements regarding competing financial interests. If no competing financial interests exist, please include the following statement: "The authors declare no competing financial interests." If competing interests are declared, please follow your statement of these competing interests with the following statement: "The authors declare no further competing financial interests."

12) A separate author contribution section is required following the Acknowledgments in all research manuscripts. All authors should be mentioned and designated by their first and middle initials and full surnames. We encourage use of the CRediT nomenclature (<https://casrai.org/credit/>).

13) ORCID IDs: ORCID IDs are unique identifiers allowing researchers to create a record of their various scholarly contributions in a single place. At resubmission of your final files, please consider providing an ORCID ID for as many contributing authors as possible.

14) Materials and data sharing: All animal and human studies must be conducted in compliance with relevant local guidelines, such as the US Department of Health and Human Services Guide for the Care and Use of Laboratory Animals or MRC guidelines, and must be approved by the authors' Institutional Review Board(s). A statement to this effect with the name of the approving IRB(s) must be included in the Materials and Methods section.

As a condition of publication, authors must make protocols and unique materials (including, but not limited to, cloned DNAs; antibodies; bacterial, animal, or plant cells; and viruses) described in our published articles freely available upon request by researchers, who may use them in their own laboratory only. All materials must be made available on request and without undue delay. We strongly encourage to deposit all the cell lines/strains and reagents generated in this study in public repositories.

All datasets included in the manuscript must be available from the date of online publication, and the source code for all custom computational methods, apart from commercial software programs, must be made available either in a publicly available database or as supplemental materials hosted on the journal website. Numerous resources exist for data storage and sharing (see Data Deposition: <https://rupress.org/jcb/pages/data-deposition>), and you should choose the most appropriate venue based on your data type and/or community standard. If no appropriate specific database exists, please deposit your data to an appropriate publicly available database. Please, deposit your mass spectrometry data in appropriate public databases.

15) JCB now requires authors to submit Source Data used to generate figures containing gels and Western blots with all revised

manuscripts. This Source Data consists of fully uncropped and unprocessed images for each gel/blot displayed in the main and supplemental figures. Since your paper includes cropped gel and/or blot images, please be sure to provide one Source Data file for each figure that contains gels and/or blots along with your revised manuscript files. File names for Source Data figures should be alphanumeric without any spaces or special characters (i.e., SourceDataF#, where F# refers to the associated main figure number or SourceDataFS# for those associated with Supplementary figures). The lanes of the gels/blots should be labeled as they are in the associated figure, the place where cropping was applied should be marked (with a box), and molecular weight/size standards should be labeled wherever possible. Source Data files will be directly linked to specific figures in the published article.

B. FINAL FILES:

Thank you for this interesting contribution, we look forward to publishing your paper in Journal of Cell Biology.

Sincerely,

Louis Reichardt, PhD
Monitoring Editor
Journal of Cell Biology

Dan Simon, PhD
Scientific Editor
Journal of Cell Biology

Reviewer #1 (Comments to the Authors (Required)):

Again, all experiments very nicely done, very beautiful videos. Lot of data.

My concern was that studied dendritic filopodia are not those filopodia, which are precursors of the dendritic spines, but other filopodia structures which are typical for young dissociated neurons cultured in low density. Authors have now provided more data and I think we can reach here an agreement. I still see the difference between filopodia of neurons cultured in low density (filopodia are more straight and longer) compared to high density cultures. Figure 4D shows a neuron which I am used to watch. I think low density DIV11 looks much younger than what DIV11 look in high density culture. This can be seen also from overall morphology (shorter dendrites). I am quite sure that authors understand what I mean. On the other hand then, does it matter? A long straight filopodium can clearly change during development to less straight and shorter filopodia. Either there is more F-actin in younger filopodia or then there is a contrast problem. When spine heads form, F-actin looks there very bright and in comparison, other structures look dark. Without spine head actin filopodia look then brighter because there are no brighter structures.

I guess I need to open my thinking, so far I have been focused on neurons around DIV14 high density culture but obviously there are filopodia also before and I guess there is no limit to say which is considered as a spine precursor. For sure a long straight filopodia can also get a synapse if it finds presynaptic partner. S1B nicely shows that when axon grows there close by, a synapse can form.

It is very evident especially from videos that EVL is needed for elongation of these filopodia. Funnily, MIM looked very inactive alone in example videos. No idea why is it so, again, I have focused on older neurons and I don't know how MIM behaves in these younger neurons. It is expressed throughout the development so one would expect that it functions similarly throughout the early development. But this is a minor issue, maybe it was just a not the best video. Clearly EVL is not helping to initiate spines, it is clearly important for the elongation of filopodia (based on filopodia dynamics and spine density phenotype of OE and KD Evl). In higher density cultures with EVL knockdown, filopodia are lost, phenotype look very similar to mDia2 knockdown in Hotulainen et al., 2009.

So taken together, I will accept the manuscript for publication as it is.

Reviewer #2 (Comments to the Authors (Required)):

The authors addressed all of my points satisfactorily. I take Reviewer 1's comments about the identification of the protrusions as bona fide dendritic filopodia serious, so as long as this reviewer is satisfied by the arguments in the author's response, I think the paper looks good and will make a nice contribution to the field.

Reviewer #3 (Comments to the Authors (Required)):

Re-Review of Parker et al.

The authors have included new data and analysis and have strengthened the paper. In addressing my primary concern, they have used a different statistical test to compare most of the data (linear mixed effects model). However, they continue to use individual filopodia as an "n" value because they argue that many of the filopodia are non-motile and considering an "n" as a dendrite or cell would obscure the potential biologically-relevant differences. This is a reasonable argument for the dynamic data they show (avg. tip speed, displacement rate, etc.), but does not pertain to other measurements that are made (length, time motile, etc.). It seems like the "n" for these data could be plotted and analyzed as averages per dendrite or per specific length of dendrite.

It is not clear to what the p values in the new Fig. 8C and D refer? Also, these data seem far too complicated. Wouldn't line graphs be better with comparisons between WT and KO on the same graph (like what they had in the previous version). In all these data it only looks like blebbistatin is having any differential effect between WT and KO.

Ghassan Mouneimne, PhD
Associate Professor, Cellular and Molecular Medicine
520-626-4616
gmouneimne@arizona.edu

1515 N. Campbell Ave
Tucson AZ 85724-5024

January 18th, 2023

Dear Dr. Simon,

We submitted the revised manuscript "**EVL and MIM/MTSS1 regulate actin cytoskeletal remodeling to promote dendritic filopodia in neurons**", by Parker et al. We thank all the reviewers for their critique. Below is our response to the remaining points raised by reviewer 3:

The authors have included new data and analysis and have strengthened the paper. In addressing my primary concern, they have used a different statistical test to compare most of the data (linear mixed effects model). However, they continue to use individual filopodia as an "n" value because they argue that many of the filopodia are non-motile and considering an "n" as a dendrite or cell would obscure the potential biologically-relevant differences. This is a reasonable argument for the dynamic data they show (avg. tip speed, displacement rate, etc.), but does not pertain to other measurements that are made (length, time motile, etc.). It seems like the "n" for these data could be plotted and analyzed as averages per dendrite or per specific length of dendrite.

The linear mix effects model that we have applied in our analyses of all the larger datasets quantifying individual filopodial motility estimates and accounts for the potential effects of collecting data from different *dendrites/cells* from different *experiments*. In our opinion, reducing the data to a smaller "n" is not beneficial since it leads to the loss of potential differences at the level of filopodial variability (which is the most crucial variable, as discuss in our last response, since it represents motile and non-motile filopodia).

It is not clear to what the p values in the new Fig. 8C and D refer? Also, these data seem far too complicated. Wouldn't line graphs be better with comparisons between WT and KO on the same graph (like what they had in the previous version). In all these data it only looks like blebbistatin is having any differential effect between WT and KO.

We appreciate the reviewer's critique and agree that it was not clear as to what the p values in these panels in Fig. 8 are assessing. These p values represent the differences in the fold change over time within each condition (i.e., assessing the time effect); we have indicated that in the figure legend now. Additionally, we included another statistical analysis that we did not have before: assessing the difference in fold change between WT and KO at each time point (for each treatment condition); this analysis illustrates the differences between WT and KO (the blebbistatin is most significantly different at most time points as mentioned by the reviewer).

We are very grateful for your constructive review process and we are looking forward to having our study published at the Journal of Cell Biology.

Sincerely,

Ghassan Mouneimne, PhD